# FedGMR: Federated Learning with Gradual Model Restoration under Asynchrony and Model Heterogeneity

## Abstract

Federated learning (FL) holds strong potential for distributed machine learning, but in heterogeneous environments, Bandwidth-Constrained Clients (BCCs) often struggle to participate effectively due to limited communication capacity. Their small sub-models learn quickly at first but become under-parameterized in later stages, leading to slow convergence and degraded generalization. We propose FedGMR—Federated Learning with Gradual Model Restoration under Asynchrony and Model Heterogeneity. FedGMR progressively increases each client's sub-model density during training, enabling BCCs to remain effective contributors throughout the process. In addition, we develop a mask-aware aggregation (MA) rule tailored for asynchronous MHFL and provide convergence guarantees showing that aggregated error scales with the average sub-model density across clients and rounds, while GMR provably shrinks this gap toward full-model FL. Extensive experiments on FEMNIST, CIFAR-10, and ImageNet-100 demonstrate that FedGMR achieves faster convergence and higher accuracy, especially under high heterogeneity and non-IID settings.

## 1 Introduction

Federated learning (FL) enables multiple clients to collaboratively train a global model without sharing local data (Pei et al., 2024; Hu et al., 2024). In practical deployments, clients frequently exhibit substantial variability in computation and communication capabilities (Zhang et al., 2022). This motivates model-heterogeneous FL (MHFL) (Diao et al., 2021; Wu et al., 2024; Liu et al., 2025; Kim et al., 2023) , where resource-limited clients are assigned compact sub-models to reduce their training and transmission cost.

However, a fundamental limitation of MHFL has been largely overlooked: **small sub-models contribute effectively in the early stage of training but gradually lose learning ability as the global model converges toward a higher-capacity solution.** As optimization deepens, these compact models become under-parameterized for the increasingly complex decision boundary, causing their updates to become noisy or even uninformative. This phenomenon leads to a form of *client bias* similar to that observed in asynchronous FL (Xie et al., 2019; Lan et al., 2024; Chen et al., 2025; Zhou et al., 2024): in AFL, bandwidth-constrained clients (BCCs) are marginalized because they upload fewer updates, while in MHFL, BCCs are marginalized because their models lack sufficient capacity to continue learning. Consequently, **MHFL sub-models converge quickly in early rounds but fail to "train deeper" in later stages, whereas full-capacity models on abundant clients continue improving. This creates a form of client imbalance similar to AFL, though arising from a fundamentally different cause**.

This dilemma leads to a natural question: *Can we design an MHFL framework in which small sub-models are used when they are beneficial—i.e., fast in early rounds—but are gradually replaced by larger ones once their limited capacity prevents further progress?* In other words, the sub-model density assigned to BCCs should not remain fixed: small models accelerate the early stage, but deeper convergence requires progressively restoring pruned capacity as training advances.

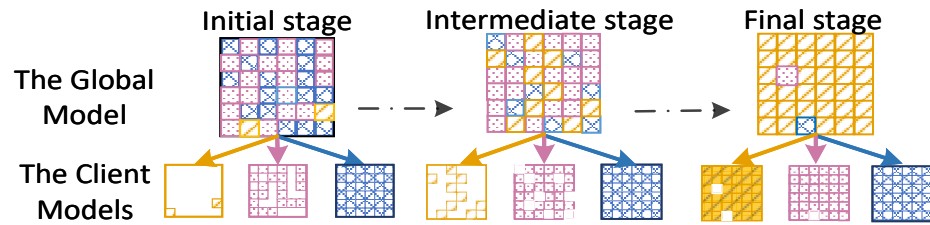

Figure 1: **GMR**: client models are heterogeneous but are gradually restored during training.

**Gradual Model Restoration.** To address this challenge, we propose **FedGMR**, an asynchronous MHFL framework built upon **Gradual Model Restoration (GMR)**. GMR initializes BCCs with compact sub-models to maximize early-round efficiency and *progressively restores their structures* as optimization proceeds, dynamically increasing model capacity when it becomes beneficial. This mechanism unifies the advantages of low-density and high-density models: rapid optimization in early rounds and strong representational power in later rounds.

To ensure stable training under asynchronous updates and evolving sub-model structures, we adopt a buffering mask-aware aggregation mechanism. Our convergence analysis shows that the global optimization bias decreases monotonically with increasing sub-model density, providing theoretical support for gradual restoration.

**Contributions.** This work makes four contributions. (1) We identify a core limitation of model-heterogeneous FL: static and highly sparse sub-models accelerate early training but inevitably stagnate in later stages due to capacity-induced marginalization. (2) We introduce *Gradual Model Restoration* (GMR), a principled mechanism that progressively increases sub-model density along the optimization trajectory, allowing bandwidth-constrained clients to break late-stage capacity bottlenecks while preserving early-round efficiency. (3) We develop a practical asynchronous system for dynamic sub-model training. Through buffering mask-aware aggregation and server-side pruning with incremental transmission, FedGMR ensures consistent global updates despite evolving and heterogeneous model structures, with minimal communication overhead. (4) We conduct extensive experiments on image benchmarks, demonstrating faster and more stable convergence, significant gains attributable to GMR, and superior robustness under strong structural and data heterogeneity.

## 2 RELATED WORKS

**Asynchronous Federated Learning (AFL).** AFL reduces client idle time by aggregating updates as soon as they arrive, accelerating training in heterogeneous systems (Dai et al., 2018; Xie et al., 2019). To mitigate the negative effects of stale updates, prior works propose staleness weighting (You et al., 2022), distance-aware adjustments (Wang et al., 2022), or explicit drift-correction mechanisms such as FedFix (Fraboni et al., 2023). Semi-asynchronous and hybrid methods (Wu et al., 2020; 2023) further balance stability and responsiveness. However, these approaches assume *homogeneous model structures* across clients. They do not address the weak and increasingly noisy contributions from bandwidth-constrained clients (BCCs), a key bottleneck identified in heterogeneous environments (Ma et al., 2021).

**Model-Heterogeneous Federated Learning (MHFL).** MHFL assigns sub-models of different sizes to match client-side computational or communication constraints. Typical strategies include layer scaling and width reduction (Diao et al., 2021; Kim et al., 2023; Ilhan et al., 2023), distillation-based aggregation (Lin et al., 2020), and pruning-based extraction (Horvath et al., 2021; Vahidian et al., 2021). More recent advances—FedRolex (Alam et al., 2022), Every Parameter Matters (Zhou et al., 2023), and Fiarse (Wu et al., 2024)—highlight the importance of preserving salient neurons or important parameters during sub-model construction. Yet despite these improvements, **static or highly sparse sub-models still suffer from intrinsic capacity limitations**: they capture useful patterns in early rounds but eventually fail to fit the increasingly complex global decision boundary. This leads to degraded late-stage accuracy and unstable aggregation when model structures differ across clients.

**Summary.** AFL reduces idle time but does not strengthen the diminishing contributions of BCCs. MHFL improves early-stage participation but cannot sustain late-stage learning due to limited model capacity. These complementary limitations expose an *efficiency–capacity trade-off* unresolved by existing work. This motivates our dynamic restoration approach, which allows sub-models to remain efficient initially yet regain capacity as training deepens.

## 3 PRELIMINARY: MODEL-HETEROGENEOUS FEDERATED LEARNING

We consider a federated learning system with a central server and $C$ heterogeneous clients, each holding a local dataset $D = \{D_1, \ldots, D_C\}$ and possessing different computation and communication budgets. To keep training speeds comparable across devices, model-heterogeneous FL assigns *smaller sub-models* to bandwidth-constrained clients (BCCs), while bandwidth-abundant clients (BACs) train larger ones.

**Sub-model representation.** Let the global model be $w \in \mathbb{R}^N$, where each coordinate corresponds to a parameter or neuron index. At round $k$, client $i$ trains a masked sub-model defined by a binary mask $m_{i,k} \in \{0, 1\}^N$, where $m_{i,k}^{(n)} = 1$ indicates that coordinate $n$ is retained. The set of clients covering coordinate $n$ is $\Gamma_k^{(n)} = \{ i \mid m_{i,k}^{(n)} = 1 \}$, where $1 \leq |\Gamma_k^{(n)}| < C$. The density of client $i$ at round $k$ is $\rho_{i,k} \in (0, 1]$, and the collection $\mathbf{P}_k = \{\rho_{i,k}\}_{i=1}^C$ forms the client-density vector describing structural heterogeneity at round $k$.

**Local objective and updates.** Each client extracts its local sub-model from the global model using $w_{i,k,0} = w_k \odot m_{i,k}$. The local objective is $F_i(w) = \mathbb{E}_{\xi \sim \mathcal{D}_i}[\ell(w; \xi)]$, and the client performs $\mathcal{T}$ steps of SGD with learning rate $\gamma > 0$. Local updates follow $w_{i,k,\tau} = w_{i,k,\tau-1} - \gamma \nabla\ell(w_{i,k,\tau-1}; \xi_{i,\tau-1}) \odot m_{i,k}$ for $\tau = 1, \ldots, \mathcal{T}$, ensuring that gradients on pruned coordinates remain zero.

**MaskFedAvg.** Because clients train sub-models with different masks, their local updates contain zeros on pruned coordinates. Naively averaging such masked updates (as in FedAvg (McMahan et al., 2017)) would treat these zeros as valid contributions, causing parameters trained only by BACs to be diluted toward zero. To avoid this issue, MaskFedAvg aggregates each coordinate solely over the clients that retain it: the server computes $w^{(n)} = \left( \sum_{i \in \Gamma_k^{(n)}} w_i^{(n)} \right) / |\Gamma_k^{(n)}|$ for $n = 1, \ldots, N$, ensuring that each coordinate is updated consistently under structural heterogeneity.

## 4 FEDGMR

We now present the full FedGMR framework. Built on top of the model-heterogeneous FL paradigm, FedGMR is driven by a single core mechanism: *dynamically increasing model density during training*. This mechanism, GMR, operates in two stages. The first stage assigns sub-models of appropriate densities to heterogeneous clients to balance round completion time. The second stage progressively restores these densities, specifying both *when* restoration should occur and *how* density should be increased, enabling sub-models to gradually approach full-model capacity.

### 4.1 STAGE 1: INITIAL DENSITY ASSIGNMENT

As in conventional MHFL, the server extracts sub-models and assigns them to clients so that their effective training speeds remain comparable. The detailed extraction procedure is given in Appendix E. Below we describe the design unique to FedGMR.

**Model density gradient descent.** Compared with asynchronous FL methods that train full models, MHFL aims to equalize training speed by assigning smaller sub-models to slower clients. However, prior MHFL methods typically rely on *fixed* densities and synchronous updates, which neither adapt to heterogeneous client runtimes nor scale well in asynchronous settings.

To address this problem, we use the time difference between clients as the gradient to dynamically adjust the client density to make all clients maintain a similar training speed to finish one round of FL training. Since communication delay is the main bottleneck, assigning small sub-models to

**Algorithm 1:** FedGMR

**Init**: global model $W_0$, client-density vector $\mathbf{P}_0$, buffer $\mathcal{B} \leftarrow \emptyset$, aggregation times $T_1 = 0$, $T_k = T_{k-1} + \Delta T$ for $k > 1$.

**while** $T < T_{\max}$ **do**
  // Client $i$ (independent)
  **if** *client $i$ is free* **then**
    $w_{i,k'+1} \leftarrow \text{LocalTrain}(w_{i,k'}, \mathcal{D}_i)$;
  // Server (semi-async)
  **if** $w_{i,k'+1}$ *arrives* **then**
    $\mathcal{B} \leftarrow \mathcal{B} \cup \{w_{i,k'+1}\}$;
  **if** $T = T_k$ **then**
    $W_{k+1} \leftarrow \text{BuffMaskFedAvg}(\mathcal{B}, W_k)$;
    $\mathbf{P}_{k+1} \leftarrow \text{GMR}(\mathbf{P}_k, \mathcal{B}, D_{\text{test}})$;
    $\{w_{i,k+1}\}_{i=1}^C \leftarrow$
      $\text{Extract}(W_{k+1}, \mathbf{P}_{k+1})$;

**Algorithm 2:** Gradual Model Restoration (GMR)

**Input:** client densities $\mathbf{P}_k = \{\rho_{i,k}\}$, buffer $\mathbf{B}$, test set $D_{\text{test}}$;
**Output:** updated densities $\mathbf{P}_{k+1}$;
**def** GMR ($\mathbf{P}_k, \mathbf{B}, D_{\text{test}}$) :
  **for** *each client $i$* **do**
    $\mathbf{w}_i \leftarrow \mathbf{B}[i]$;
    **if** *Initial stage* **then**
      update $\rho_{i,k}$ using Eq. 2;
    **if** *Second stage and* $\text{ESM}(\mathbf{w}_i, D_{\text{test}})$
    **then**
      $\rho^{\text{next}} \leftarrow \min\{\rho \in \mathbf{P}_k \mid \rho > \rho_{i,k}\}$;
      **if** $\rho^{next}$ *exists* **then**
        $\rho_{i,k} \leftarrow \rho^{\text{next}}$;
  **return** $\mathbf{P}_{k+1} = \{\rho_{i,k}\}$

BCCs reduces upload cost and accelerates the client speed. We formulate this goal as the following optimization problem:

$$\min_{\{\rho_i\}} \ \mathcal{L}_{\text{time}}(\rho) = \frac{1}{2} \sum_{i=1}^{N} \left( \frac{t_i(\rho_i) - t_s}{t(\rho_i)} \right)^2, \tag{1}$$

where $t_i(\rho_i)$ denotes the one-round training time of client $i$ under density $\rho_i$, and $t_s$ is the round time of the most bandwidth-abundant (reference) client.

When the round completion time difference $(t_{i,k+1} - t_{i,k})$ is sufficiently large, it can be approximated as the gradient of $t_i$ with respect to the model density $\rho_i$ (see Appendix B). Thus, we update each client's model density using gradient descent on $\mathcal{L}_{\text{time}}$:

$$\rho_{i,k+1} = \Pi_{[\rho_{\min}, \, 1]} \left( \rho_{i,k} - \lambda \, \nabla_{\rho_i} \mathcal{L}_{\text{time}}(\rho_k) \right), \tag{2}$$

where $\lambda$ is a learning rate and $\Pi_{[\rho_{\min}, 1]}(\cdot)$ is the projection onto interval $[\rho_{\min}, 1]$. Using the monotonic relation between density and runtime, the gradient in equation 2 can be approximated by $\frac{t_s - t_{i,k}}{t_{i,k}} \rho_{i,k}$. This adaptive initialization balances client-side training time without requiring prior knowledge of client bandwidth, and mitigates early-stage training imbalance.

### 4.2 The Second Stage: Density Restoration

In the early stage of MHFL, assigning small sub-models with low density $\rho_i$ allows clients to update frequently. However, such compact sub-models eventually become under-parameterized as the global optimization progresses, limiting their ability to fit increasingly complex decision boundaries. Increasing the density $\rho_i$ is a natural way to improve model capacity $g(\rho_i)$, but doing so also increases latency $t(\rho_i)$ and reduces update frequency $f(\rho_i)$. Thus, restoring density introduces an inherent trade-off between update frequency and representational capacity.

FedGMR resolves this tension by restoring density only when necessary—namely, when limited capacity becomes the primary bottleneck. In other words, the optimal density should evolve dynamically and progressively increase as training deepens, an observation empirically validated in Sec. 6.1.

**When to restore the sub-models?** Directly quantifying the relationship among capacity, convergence speed, and generalization is difficult. Instead, consider an idealized scenario: if a sub-model is capacity-limited, it may converge to a sub-optimal point where its updates plateau, even though additional capacity would enable further progress. We formally analyze this condition in Appendix C.

Motivated by this insight, FedGMR triggers density restoration when the sub-model enters a *plateau phase*. Saturation is detected via an **Early Stopping Mechanism (ESM)** that monitors validation accuracy across a fixed window. If the accuracy shows no improvement for a patience interval $k_{\text{rest}}$, the client is deemed saturated and restoration is activated.

**How density should be increased?** Upon restoration, a client upgrades from its current density $\rho_i$ to the next available level. For instance, if the density ladder is $\{0.05, 0.1, 0.2, 0.5, 1.0\}$ as defined by Eq. equation 2, a client at density $0.1$ will restore to $0.2$ once the ESM condition is satisfied. This *multi-level schedule* is reminiscent of Once-for-All training (Cai et al., 2020), where nested sub-models are sequentially optimized across clients with diverse capabilities, reinforcing shared structures and improving convergence stability.

As densities increase, client training speeds diverge, reintroducing system heterogeneity. To avoid idle waiting, FedGMR integrates density restoration with a semi-asynchronous aggregation scheme, ensuring steady progress throughout training.

**Practical Enhancements.** FedGMR can optionally reduce server–to–client communication through *incremental model splitting* (IMS), which transmits only the newly added parameters required for the next density level. IMS provides practical speedup by avoiding redundant transmission, yet it is conceptually independent of the GMR mechanism; implementation details are provided in Appendix F. Algorithm 1 summarizes the full training pipeline of FedGMR, and Algorithm 2 specifies the density restoration rule.

## 5 MASK-AWARE AGGREGATION AND ITS CONVERGENCE PROPERTIES

Following the mask-aware aggregation paradigm, FedGMR incorporates mask information directly into the aggregation rule to ensure consistent updates under heterogeneous sub-model structures. Under asynchronous training, we further extend this rule with *staleness-aware weighting* and a lightweight *server-side buffer* to accommodate out-of-order arrivals and maintain update stability.

Based on this aggregation formulation, we analyze how heterogeneous sub-model densities influence the optimization dynamics. Our convergence characterization shows that higher sub-model density reduces the aggregation discrepancy and leads to smaller optimization error. The analysis further demonstrates that mask-aware aggregation yields more stable gradient and weight behaviors than naive averaging, thus enabling reliable optimization under heterogeneous sparsity.

### 5.1 BUFFMASKFEDAVG: AGGREGATING VARYING-SIZE MODELS.

We introduce **BuffMaskFedAvg**, a buffer-based, mask-aware aggregator for asynchronous MHFL. The server keeps the latest received models $\{\mathbf{w}_{i,k_i'}\}$, preventing forgetting of slower clients and ensuring all available information is used in aggregation.

To handle asynchronous arrivals, we adopt staleness-aware weighting Xie et al. (2019). For neuron $n$,

$$s_{i,k}^{(n)} = \left(1 + (k - k_i')\right)^{-\alpha}, \qquad \beta_{i,k}^{(n)} = \frac{s_{i,k}^{(n)}}{\sum_{j \in \Gamma_k^{(n)}} s_{j,k}^{(n)}}, \tag{3}$$

where $\Gamma_k^{(n)}$ is the set of clients whose sub-models retain neuron $n$.

The global aggregation is

$$W_k^{(n)} = \begin{cases} \dfrac{\sum_{i \in \Gamma_k^{(n)}} \beta_{i,k}^{(n)} w_{i,k}^{(n)}}{\sum_{i \in \Gamma_k^{(n)}} \beta_{i,k}^{(n)}}, & \text{if } \sum_{i \in \Gamma_k^{(n)}} \beta_{i,k}^{(n)} > 0, \\ W_{k-1}^{(n)}, & \text{otherwise,} \end{cases} \tag{4}$$

which ensures neuron-wise consistency.

The gradient form is

$$W_{k+1} = W_k - \gamma \sum_{i=1}^{C} (\beta_{i,k} \odot m_{i,k}) \odot \left( \sum_{\tau=1}^{\mathcal{T}} \nabla F_i(w_{i,k,\tau-1}, \xi_{i,\tau-1}) \right), \tag{5}$$

naturally ignoring coordinates with $\Gamma_k^{(n)} = \emptyset$. Pseudocode is provided in Appendix D.

## 5.2 Convergence Analysis

We analyze how sub-model sparsity shapes the convergence behavior of model-heterogeneous FL, aiming to characterize the optimization gap relative to full-model FL and clarify why increasing density (as in FedGMR) narrows this gap. Sparsity impacts aggregation in two fundamental ways: (1) each client supplies only partial gradients, shrinking the effective optimization subspace; and (2) heterogeneous masks induce uneven coverage across clients, creating aggregation discrepancy. These effects accumulate over rounds, producing a density-dependent gap that monotonically decreases as model density increases—providing the intuition behind GMR.

To isolate the effect of sparsity, we analyze the synchronous (zero-staleness) setting and reduce BuffMaskFedAvg to its core *mask-aware aggregation* (MA) operator. This abstraction allows us to focus on how masked gradients are aggregated and how coverage across density groups determines the resulting convergence gap. We now state the assumptions and present the formal results. Following standard FL practice, each pruned sub-model is treated as the full model composed with a binary masking operator; for client $i$ at round $k$, the density $p_{i,k}$ extends classical smoothness, variance, and heterogeneity assumptions to masked structures.

**Assumption 1.** (*Smoothness*). All client cost functions $F_i$ are $L$-smooth. That is, for any $w, \phi \in \mathbb{R}^N$ and any client $i$, there exists a constant $L > 0$ such that:

$$\|\nabla F_i(w) - \nabla F_i(\phi)\| \leq L\|w - \phi\|,$$
$$\|\nabla F_i(w \odot m_1) - \nabla F_i(\phi \odot m_2)\| \leq L\|w \odot m_1 - \phi \odot m_2\|. \tag{6}$$

**Assumption 2.** (*Bounded Gradient*). Define the stochastic full gradient $g_{i,k,\tau} := \nabla F_i(W_{i,k,\tau}, \xi_{i,k,\tau})$. Then for any client $i$, round $k$, and local step $\tau$, it holds that

$$\mathbb{E}\|g_{i,k,\tau}\|^2 \leq G^2, \quad \mathbb{E}\|g_{i,k,\tau} \odot m_{i,k}\|^2 \leq f_1^2(p_{i,k})G^2. \tag{7}$$

**Assumption 3.** (*Gradient Noise*). Define the gradient noise $n_{i,k,\tau} := \nabla F_i(W_{i,k,\tau}, \xi_{i,k,\tau}) - \nabla F_i(W_{i,k,\tau})$. Then for any $i, k, \tau$, it holds that

$$\mathbb{E}\|n_{i,k,\tau}\|^2 \leq \sigma^2, \quad \mathbb{E}\|n_{i,k,\tau} \odot m_{i,k}\|^2 \leq f_2^2(p_{i,k})\sigma^2. \tag{8}$$

**Assumption 4.** (*Bounded Non-IID Bias*). Define the Non-IID bias $b_i(W) := \nabla F_i(W) - \nabla F(W)$. Then it holds that

$$\mathbb{E}\|b_i(W)\|^2 \leq \zeta^2, \quad \mathbb{E}\|b_i(W) \odot m_{i,k}\|^2 \leq f_3^2(p_{i,k})\zeta^2. \tag{9}$$

where $f_i(\cdot) \in [0, 1]$ for $i \in \{1, 2, 3\}$ are non-decreasing sparsity–scaling functions that capture how density affects the drift, variance, and non-IID bias.

We adopt the notation introduced in Appendix G.1, where the two-stage aggregation view is formalized. Here, $S_{g,k}$ denotes the retained coordinates of density group $g$, and $\Gamma^*_{g,k}$ represents its worst-case coverage factor. This decomposition enables tighter and more structured convergence bounds by separating intra-group averaging from inter-group mask-aware aggregation. All proofs of the following lemmas and theorems are provided in Appendix G, with a brief outline presented below.

**Theorem 1.** Under Assumptions 1–3, suppose all client masks satisfy $\Gamma_k^{(n)} \geq 1$. Then the expected one-round descent bound $\mathbb{E}\big[F(W_{k+1})\big] - \mathbb{E}\big[F(W_k)\big]$ satisfies:

$$\mathbb{E}\big[F(W_{k+1})\big] - \mathbb{E}\big[F(W_k)\big] \leq -\frac{G_0}{2}\mathbb{E}\|\nabla F(W_k)\|^2 + \frac{3G_0 J_0}{2}\mathbb{E}\sum_{n=1}^{N}\big\|\frac{1}{\Gamma_k^{(n)}}\sum_{j \in \mathcal{C}_k^{(n)}}\nabla F_j^{(n)}(W_k)\big\|^2$$

$$+ \frac{G_0 H_0}{6}\mathcal{A}_k^\dagger f_1^2(p_{g,k})G^2 + \frac{3G_0 I_0}{2}\mathcal{B}_k^\dagger f_2^2(p_{g,k})\sigma^2. \tag{10}$$

where $G_0 = \mathcal{T}\gamma$, $H_0 = L^2\mathcal{T}^2\gamma^2(3L\mathcal{T}\gamma + 1)$, $I_0 = L\gamma$, $J_0 = L\mathcal{T}\gamma$, $\mathcal{A}_k^\dagger = \sum_{g=1}^{G}|\mathcal{C}_g|\Gamma^*_{g,k}$, and $\mathcal{B}_k^\dagger = \sum_{g=1}^{G}|\mathcal{C}_g|/(\Gamma^*_{g,k})^2$. Full proof is provided in Appendix G.2.3.

**Interpretation.** Theorem 1 isolates the effect of mask-aware aggregation (MA) on a single optimization step. The second term, The second term $\mathbb{E}\sum_{n=1}^{N}\big\|\frac{1}{\Gamma}\Gamma_k^{(n)}\sum_{j \in \mathcal{C}_k^{(n)}}\nabla F_j^{(n)}(W_k)\big\|^2$ arises

because MA reconstructs the full-model gradient by normalizing each coordinate only over clients that retain it. This induces an amplification factor $|\mathcal{C}_g|/\Gamma^*_{g,k}$, improving gradient completeness but enlarging the sparse-drift and variance terms reflected in $\mathcal{A}^\dagger_k$ and $\mathcal{B}^\dagger_k$.

Compared with naive weight or gradient averaging (Appendix G.5), MA preserves consistent gradient directions and stable weight evolution. This stability is crucial for model restoration: newly restored coordinates integrate smoothly into optimization, avoiding oscillation and enabling progress beyond the capacity-limited plateau (empirically confirmed in Section 6.4).

**Theorem 2.** Under Assumptions 1–4, suppose all client masks satisfy $\Gamma^{(n)}_k \geq 1$, and choosing the local learning rate $\gamma < \frac{1}{6L\mathcal{T}}$ for IID data and $\gamma < \frac{1}{12L\mathcal{T}}$ for Non-IID data, MHFL converges to a small neighborhood of a stationary point of standard FL as follows: .

$$\frac{1}{K} \sum_{k=1}^{K} \mathbb{E}\|\nabla F(W_k)\|^2 \tag{11}$$

$$\leq \frac{4\mathbb{E}[F(W_1)]}{G_0 K} + \underbrace{\frac{3LH_0}{2K} \sum_{k=1}^{K} \mathcal{A}^\dagger_k f_1^2(p_{g,k}) G^2}_{\text{Model Drift Term}} + \underbrace{\frac{6I_0}{K} \sum_{k=1}^{K} \mathcal{B}^\dagger_k f_2^2(p_{g,k}) \sigma^2}_{\text{Variance Term}} \underbrace{\left( + \frac{J_0}{K} \sum_{k=1}^{K} \mathcal{A}^\dagger_k f_3^2(p_{g,k}) \zeta^2 \right)}_{\text{Bias Term (Non-IID only)}}.$$

**Interpretation.** Theorem 2 aggregates the per-round effects of Theorem 1 and yields a unified convergence bound: MA converges to a density-dependent neighborhood of a stationary point of full-model FL. The radius of this neighborhood is governed by terms of the form $\frac{1}{K} \sum_{k=1}^{K} \sum_{g=1}^{G} X f^2(p_{g,k})$, $X \in \{\mathcal{A}^\dagger_k, \mathcal{B}^\dagger_k\}$, which decrease as client densities increase. Thus, higher densities—as induced by GMR—improve coverage, shrink the amplification factors, and tighten the convergence neighborhood. The complete proof is provided in Appendix G.2.4.

From a subspace perspective, as shown in Appendix G.5, the aggregated update at round $k$ lies in the feasible span $\mathcal{S}^{(k)}_{\text{feas}} = \text{span}\{\nabla F_i(W_k) \odot m_{i,k}\}$. Model restoration ensures $m_{i,k} \leq m_{i,k+1}$, leading to a monotonically expanding subspace $\mathcal{S}^{(k)}_{\text{feas}} \subseteq \mathcal{S}^{(k+1)}_{\text{feas}}$, and increased coverage $\Gamma^{(n)}_k \uparrow$. Consequently, the optimization search domain enlarges over training and the coverage-induced noise diminishes, allowing the solution to approach the full-model optimum more closely.

# 6 EXPERIMENT

**Datasets and Tasks.** We evaluate FedGMR on three datasets that span small, medium, and large neural architectures, enabling a comprehensive assessment across different model capacities: (1) a Conv-2 network on FEMNIST (Caldas et al., 2018), (2) a VGG-11 network (Simonyan & Zisserman, 2014) on CIFAR-10 (Krizhevsky et al., 2009), and (3) a ResNet-18 network (He et al., 2016) on ImageNet-100 (Jiang et al., 2022). For the Non-IID setting, FEMNIST follows the LEAF partitioning protocol (Caldas et al., 2018); CIFAR-10 and ImageNet-100 use Dirichlet label partitioning with concentration parameter $\alpha = 0.6$. We simulate a federated environment with one server and ten clients whose communication bandwidths vary, thereby inducing High, Medium, and Low degrees of system heterogeneity depending on the number of BCCs. Full implementation details are provided in Appendix H.

## 6.1 FEASIBILITY STUDY: VERIFYING STAGE-DEPENDENT TRAINING SPEED

We hypothesize that the most effective sub-model density is stage-dependent, increasing as training deepens. To examine this, we train fixed-density sub-models independently and compute their accuracy–time slopes to measure learning speed at different stages. As detailed in Appendix I and illustrated in Fig. 2, lower densities (e.g., FEMNIST $\approx 0.3$–$0.4$, ImageNet-100 $\approx 0.6$) achieve the fastest early-stage improvement (FEMNIST 10–75%, ImageNet-100 3–10%), while higher densities (0.9–1.0) dominate in later stages as the optimization landscape becomes more complex. This shift in optimal density provides direct empirical evidence that training speed depends on the stage.

Consequently, these observations strongly support a gradual restoration strategy in which sub-model capacity is increased progressively throughout training.

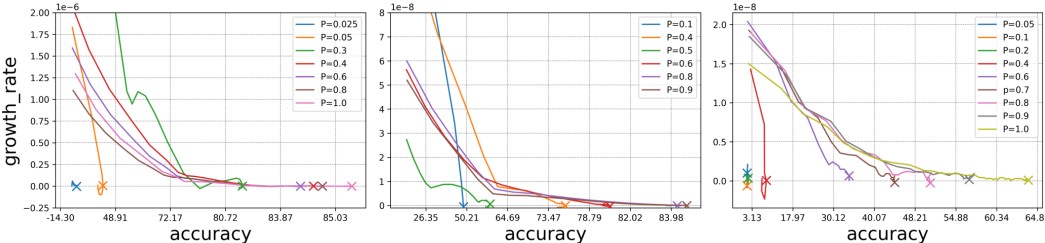

Figure 2: The accuracy growth rate with different model densities.

## 6.2 Performance vs. Baselines: Accuracy, Speed, and MRI

**Baselines.** We compare FedGMR with three full-model FL approaches— (1) **Syn-FL** (synchronous FedAvg (McMahan et al., 2017)), (2) **FedAsync** (Xie et al., 2019)—and four state-of-the-art MHFL methods: **HeteroFL** (Diao et al., 2021), **FjORD** (Horvath et al., 2021), **FedRolex** (Alam et al., 2022), **FIARSE** (Wu et al., 2024).

**Evaluation protocol.** To compare training speed, all methods are evaluated at a fixed wall-clock budget $t^\star$ chosen at the point where FedGMR stabilizes (e.g., $t^\star = 70\text{k s}$). The reported accuracy at $t^\star$ is a smoothed estimate—the mean±std of 11 test evaluations in a symmetric $\pm 5$ window around $t^\star$. To aggregate gains across settings, we report the *Mean Relative Improvement (MRI)* of method $M$ over baselines $\mathcal{B}$: $\text{MRI}(M) = \frac{1}{|\mathcal{B}|} \sum_{b \in \mathcal{B}} \frac{\text{Acc}(M) - \text{Acc}(b)}{\text{Acc}(b)}$, where $\text{Acc}(\cdot)$ is the smoothed accuracy at $t^\star$. Higher MRI indicates stronger overall gains.

Table 1: Test accuracy (%) of FedGMR and baselines across datasets and heterogeneity levels.

| Hetero. | Method | FEMNIST (70k) | | CIFAR-10 (180k) | | ImageNet-100 (250k) | |
| | | IID | Non-IID | IID | Non-IID | IID | Non-IID |
|---|---|---|---|---|---|---|---|
| High | **FedGMR** | **82.61**$_{\pm 0.29}$ | 81.55$_{\pm 0.20}$ | **84.88**$_{\pm 0.24}$ | **81.41**$_{\pm 0.36}$ | **62.32**$_{\pm 0.45}$ | **59.72**$_{\pm 0.44}$ |
| | FedAvg | 75.59$_{\pm 0.53}$ | 74.71$_{\pm 0.24}$ | 65.38$_{\pm 0.78}$ | 62.86$_{\pm 0.52}$ | 48.60$_{\pm 1.10}$ | 47.74$_{\pm 1.00}$ |
| | FedAsyn | 81.77$_{\pm 0.28}$ | 81.13$_{\pm 0.18}$ | 83.17$_{\pm 0.25}$ | 77.91$_{\pm 0.60}$ | 59.70$_{\pm 0.59}$ | 57.67$_{\pm 0.45}$ |
| | HeteroFL | 82.30$_{\pm 0.16}$ | 79.84$_{\pm 0.16}$ | 80.47$_{\pm 0.19}$ | 76.91$_{\pm 1.20}$ | 41.91$_{\pm 0.52}$ | 28.10$_{\pm 0.35}$ |
| | FedRolex | 81.16$_{\pm 0.15}$ | 79.21$_{\pm 0.26}$ | 79.70$_{\pm 0.22}$ | 78.36$_{\pm 0.29}$ | 41.20$_{\pm 0.70}$ | 29.78$_{\pm 0.75}$ |
| | Fjord | 82.58$_{\pm 0.17}$ | **81.88**$_{\pm 0.27}$ | 80.64$_{\pm 0.21}$ | 80.27$_{\pm 0.19}$ | 40.95$_{\pm 0.61}$ | 32.35$_{\pm 1.10}$ |
| | Fiarse | 81.28$_{\pm 0.16}$ | 78.97$_{\pm 0.21}$ | 69.73$_{\pm 0.41}$ | 35.14$_{\pm 0.55}$ | 54.41$_{\pm 0.78}$ | 49.73$_{\pm 0.61}$ |
| | **MRI** | **0.0236** | **0.0294** | **0.1181** | **0.2947** | **0.3322** | **0.5773** |
| Medium | **FedGMR** | 82.79$_{\pm 0.40}$ | **82.73**$_{\pm 0.29}$ | **84.80**$_{\pm 0.11}$ | **82.32**$_{\pm 0.17}$ | **63.57**$_{\pm 0.31}$ | **61.91**$_{\pm 0.61}$ |
| | FedAvg | 76.00$_{\pm 0.40}$ | 74.94$_{\pm 0.39}$ | 65.75$_{\pm 0.54}$ | 63.54$_{\pm 0.67}$ | 49.20$_{\pm 1.00}$ | 48.94$_{\pm 1.20}$ |
| | FedAsyn | 82.47$_{\pm 0.19}$ | 81.71$_{\pm 0.26}$ | 84.06$_{\pm 0.33}$ | 80.00$_{\pm 0.38}$ | 62.18$_{\pm 0.46}$ | 58.79$_{\pm 0.40}$ |
| | HeteroFL | 76.76$_{\pm 0.66}$ | 79.62$_{\pm 0.15}$ | 80.44$_{\pm 0.10}$ | 77.79$_{\pm 0.29}$ | 41.89$_{\pm 0.55}$ | 32.57$_{\pm 0.84}$ |
| | FedRolex | 81.90$_{\pm 0.23}$ | 79.51$_{\pm 0.15}$ | 79.90$_{\pm 0.14}$ | 72.20$_{\pm 0.41}$ | 43.12$_{\pm 0.75}$ | 36.57$_{\pm 0.78}$ |
| | Fjord | **82.82**$_{\pm 0.093}$ | 82.19$_{\pm 0.24}$ | 80.77$_{\pm 0.19}$ | 80.49$_{\pm 0.24}$ | 41.58$_{\pm 1.20}$ | 38.36$_{\pm 1.30}$ |
| | Fiarse | 81.73$_{\pm 0.10}$ | 79.93$_{\pm 0.30}$ | 69.72$_{\pm 0.33}$ | 34.25$_{\pm 0.56}$ | 56.48$_{\pm 0.44}$ | 54.38$_{\pm 0.69}$ |
| | **MRI** | **0.0325** | **0.0396** | **0.1134** | **0.3249** | **0.3268** | **0.4440** |
| Low | **FedGMR** | **83.78**$_{\pm 0.29}$ | **82.80**$_{\pm 0.28}$ | **85.60**$_{\pm 0.21}$ | **83.76**$_{\pm 0.26}$ | **64.86**$_{\pm 0.26}$ | **62.97**$_{\pm 0.42}$ |
| | FedAvg | 76.11$_{\pm 0.45}$ | 74.95$_{\pm 0.41}$ | 66.14$_{\pm 0.76}$ | 64.15$_{\pm 0.44}$ | 49.26$_{\pm 0.89}$ | 48.31$_{\pm 0.73}$ |
| | FedAsyn | 83.44$_{\pm 0.19}$ | 81.90$_{\pm 0.40}$ | 85.49$_{\pm 0.25}$ | 82.58$_{\pm 0.35}$ | 63.46$_{\pm 0.40}$ | 62.15$_{\pm 0.84}$ |
| | HeteroFL | 80.53$_{\pm 0.20}$ | 78.57$_{\pm 0.26}$ | 79.59$_{\pm 0.19}$ | 73.88$_{\pm 0.46}$ | 48.73$_{\pm 0.51}$ | 40.85$_{\pm 0.73}$ |
| | FedRolex | 80.64$_{\pm 0.21}$ | 77.70$_{\pm 0.23}$ | 79.36$_{\pm 0.20}$ | 73.82$_{\pm 0.38}$ | 49.05$_{\pm 1.10}$ | 40.11$_{\pm 1.10}$ |
| | Fjord | 82.26$_{\pm 0.19}$ | 81.07$_{\pm 0.23}$ | 80.13$_{\pm 0.26}$ | 79.40$_{\pm 0.24}$ | 41.84$_{\pm 0.88}$ | 41.24$_{\pm 0.88}$ |
| | Fiarse | 82.52$_{\pm 0.17}$ | 78.85$_{\pm 0.25}$ | 76.76$_{\pm 0.20}$ | 63.01$_{\pm 0.99}$ | 60.26$_{\pm 0.51}$ | 57.83$_{\pm 0.49}$ |
| | **MRI** | **0.0363** | **0.0511** | **0.1055** | **0.1621** | **0.2698** | **0.3406** |

**Summary of main results.** From Table 1, **FedGMR** achieves the best or near-best accuracy across all datasets and heterogeneity regimes. Its lead *widens* as tasks become harder and as data/system skew increases. For example, under *High* heterogeneity with non-IID splits, FedGMR's MRI rises with task difficulty—from **+0.029** on FEMNIST to **+0.577** on ImageNet-100—consistent with the growing model/data complexity (Conv2D → VGG-11 → ResNet-18). These trends confirm our

hypothesis: fixed-capacity MHFL struggles to exploit BCCs in late stages, whereas **FedGMR** keeps them active and *gradually restores* capacity, sustaining improvements. Consequently, FedGMR achieves the largest gains on challenging settings, while remaining competitive on easier tasks.

**Baseline comparison.** Synchronous FL converges slowly, as fast clients idle and heterogeneous participation cannot be exploited. HeteroFL's fixed structured pruning removes salient neurons, limiting final accuracy on complex models. FIARSE applies unstructured magnitude pruning but adopts uniform ratios across layers, which becomes unstable on deep networks (e.g., 26.6% on ImageNet-100 Non-IID vs. FedGMR 59.7%). FedRolex rotates training across different sub-model regions, but regions have unequal importance, this rotation introduces inconsistency and inefficiency, resulting in weaker convergence (e.g., 41.9% on ImageNet-100 IID vs. FedGMR 62.3%). FjORD inherits nested subnetworks and performs well on easier tasks, but its benefits collapse as task and system heterogeneity scale up (e.g., 32.4% on ImageNet-100 High Non-IID vs. 59.7% for FedGMR).

Overall, these baselines perform well only under restricted conditions. **FedGMR**, by contrast, consistently provides robust improvements across datasets and heterogeneity levels, demonstrating strong universality under realistic, highly skewed FL environments.

### 6.3 Ablation experiments with different parts of FedGMR

**Ablation setup.** We decompose FedGMR into its key components to isolate their contributions: (1) the core GMR mechanism, (2) asynchronous aggregation, and (3) the optional buffer and IMS modules, which improve practical efficiency but incur additional overhead. For fair comparison, all variants maintain the same restoration sensitivity so that differences arise solely from removing each module rather than modifying the restoration schedule or density trajectory.

**Result analysis.** **First**, asynchrony is essential: synchronous aggregation degrades accuracy (e.g., FEMNIST–High: $82.24\% \rightarrow 79.95\%$) as it slows progress after density restoration. **Second**, GMR is the main improvement source, scaling with task difficulty. Removing GMR causes substantial accuracy drops (e.g., ImageNet-100 (High IID): $61.64\% \rightarrow 54.33\%$), confirming that restoration resolves the fixed-density bottleneck. Moreover, Fig. 3 plots average density and accuracy over time, highlighting FedGMR's superior speed, specifically during model restoration. **Third**, the buffer and IMS are optional modules that improve practical efficiency but are not part of the core mechanism. Both add overhead: the buffer stores recent client models—though in large-scale settings this memory cost can be reduced by keeping only lagging clients' updates—while IMS transmits only incremental model segments after restoration, adding light processing. Despite these costs, both offer consistent gains on challenging tasks. For instance, removing the buffer drops accuracy from $61.64\% \rightarrow 56.24\%$ (High IID), showing its stabilizing effect under asynchrony. IMS mainly reduces server–to–client communication, which becomes increasingly beneficial at scale; see Appendix J.

Table 2: Ablation of FedGMR components across datasets and heterogeneity levels.

| Hetero. | Method | FEMNIST (70k) | | CIFAR-10 (180k) | | ImageNet-100 (250k) | |
| | | IID | Non-IID | IID | Non-IID | IID | Non-IID |
|---|---|---|---|---|---|---|---|
| High | FedGMR | 82.24 $_{\pm 0.14}$ | 81.50 $_{\pm 0.28}$ | **84.12** $_{\pm 0.18}$ | 80.38 $_{\pm 0.47}$ | **61.64** $_{\pm 0.37}$ | 58.67 $_{\pm 0.64}$ |
| | w/o Asyn | 79.95 $_{\pm 0.26}$ | 78.98 $_{\pm 0.65}$ | 81.64 $_{\pm 0.55}$ | 76.97 $_{\pm 0.93}$ | 58.02 $_{\pm 1.10}$ | 56.39 $_{\pm 1.00}$ |
| | w/o GMR | 81.92 $_{\pm 0.084}$ | 77.77 $_{\pm 0.17}$ | 80.93 $_{\pm 0.26}$ | 71.96 $_{\pm 0.30}$ | 54.33 $_{\pm 0.44}$ | 50.53 $_{\pm 0.86}$ |
| | w/o Buff | 82.07 $_{\pm 0.20}$ | 81.37 $_{\pm 0.33}$ | 83.12 $_{\pm 0.22}$ | 78.87 $_{\pm 0.27}$ | 56.24 $_{\pm 0.44}$ | 56.33 $_{\pm 0.48}$ |
| | w/o IMS | **82.29** $_{\pm 0.19}$ | **81.60** $_{\pm 0.26}$ | 82.67 $_{\pm 0.20}$ | 80.29 $_{\pm 0.23}$ | 59.59 $_{\pm 0.53}$ | 56.90 $_{\pm 0.34}$ |
| | w/o (Buff, IMS) | 81.30 $_{\pm 0.27}$ | 81.35 $_{\pm 0.26}$ | 82.80 $_{\pm 0.19}$ | 78.87 $_{\pm 0.46}$ | 59.16 $_{\pm 0.48}$ | 55.88 $_{\pm 0.55}$ |
| Medium | FedGMR | 82.75 $_{\pm 0.22}$ | **82.73** $_{\pm 0.29}$ | 84.34 $_{\pm 0.17}$ | 81.40 $_{\pm 0.42}$ | **62.55** $_{\pm 0.46}$ | **61.32** $_{\pm 0.47}$ |
| | w/o Asyn | 80.63 $_{\pm 0.52}$ | 81.31 $_{\pm 0.34}$ | 83.85 $_{\pm 0.40}$ | 79.89 $_{\pm 0.76}$ | 59.28 $_{\pm 0.59}$ | 57.45 $_{\pm 0.72}$ |
| | w/o GMR | 81.91 $_{\pm 0.10}$ | 79.99 $_{\pm 0.26}$ | 83.62 $_{\pm 0.14}$ | 78.60 $_{\pm 0.25}$ | 56.53 $_{\pm 0.46}$ | 55.76 $_{\pm 0.46}$ |
| | w/o Buff | **82.79** $_{\pm 0.15}$ | 81.87 $_{\pm 0.18}$ | 83.70 $_{\pm 0.24}$ | 80.75 $_{\pm 0.24}$ | 60.58 $_{\pm 0.44}$ | 57.37 $_{\pm 0.50}$ |
| | w/o IMS | 82.59 $_{\pm 0.27}$ | 82.03 $_{\pm 0.19}$ | 83.83 $_{\pm 0.25}$ | 81.40 $_{\pm 0.20}$ | 61.17 $_{\pm 0.26}$ | 59.13 $_{\pm 0.30}$ |
| | w/o (Buff, IMS) | 82.17 $_{\pm 0.16}$ | 82.01 $_{\pm 0.18}$ | 83.73 $_{\pm 0.24}$ | 80.40 $_{\pm 0.31}$ | 59.92 $_{\pm 0.31}$ | 58.60 $_{\pm 0.72}$ |
| Low | FedGMR | 83.16 $_{\pm 0.20}$ | 82.17 $_{\pm 0.30}$ | **85.57** $_{\pm 0.13}$ | **83.46** $_{\pm 0.15}$ | **64.10** $_{\pm 0.33}$ | **62.97** $_{\pm 0.42}$ |
| | w/o Asyn | 82.26 $_{\pm 0.23}$ | 81.62 $_{\pm 0.48}$ | 84.84 $_{\pm 0.17}$ | 81.23 $_{\pm 0.54}$ | 58.43 $_{\pm 0.79}$ | 56.70 $_{\pm 1.00}$ |
| | w/o GMR | 82.69 $_{\pm 0.10}$ | 79.61 $_{\pm 0.26}$ | 84.28 $_{\pm 0.18}$ | 80.56 $_{\pm 0.25}$ | 59.62 $_{\pm 0.67}$ | 56.69 $_{\pm 1.00}$ |
| | w/o Buff | **83.36** $_{\pm 0.17}$ | 82.78 $_{\pm 0.36}$ | 84.75 $_{\pm 0.13}$ | 82.76 $_{\pm 0.12}$ | 63.38 $_{\pm 0.43}$ | 61.35 $_{\pm 0.50}$ |
| | w/o IMS | 83.21 $_{\pm 0.18}$ | **82.86** $_{\pm 0.22}$ | 85.44 $_{\pm 0.17}$ | 83.03 $_{\pm 0.18}$ | 63.36 $_{\pm 0.36}$ | 61.40 $_{\pm 0.32}$ |
| | w/o (Buff, IMS) | 83.02 $_{\pm 0.16}$ | 82.45 $_{\pm 0.34}$ | 84.79 $_{\pm 0.22}$ | 83.14 $_{\pm 0.18}$ | 62.71 $_{\pm 0.28}$ | 60.73 $_{\pm 0.33}$ |

## 6.4 COMPARISON WITH THE DIFFERENT AGGREGATION METHODS

In our convergence analysis, we examine the effect of MaskFedAvg. For comparison, we additionally include two alternative aggregation rules—**gradient averaging**(GA) and **FedAvg**(FA)—as formally defined in Appendix G.1.

**Result analysis.** We summarize four key observations from Table 3, which jointly evaluate how different aggregation rules interact with GMR. **(1) MA achieves the strongest performance when combined with GMR.** Across all datasets and heterogeneity levels, GMR + MA consistently delivers the best or near-best accuracy. For instance, under High heterogeneity it reaches **82.24%** on FEMNIST, **84.12%** on CIFAR-10, and **61.64%** on ImageNet-100. This aligns with Theorem 1: as density grows, MA normalizes each coordinate independently, allowing newly restored parameters to be incorporated smoothly without destabilizing gradients. **(2) The benefit of GMR may be limited by the aggregation rule.** On CIFAR-10 and ImageNet-100, GMR may yield limited gains under GA/FA, and **GMR + GA** can even underperform **w/o GMR + GA**. The reason is that density restoration reactivates many previously pruned parameters, causing large shifts in update magnitudes that GA/FA cannot adequately normalize. As discussed in Appendix G.5, GA/FA are not well aligned with GMR's restoration dynamics. **(3) Without GMR, MA does not perform well all the time.** In the fixed-density regime, MA attempts to approximate the full gradient by rescaling sparse updates, but this rescaling also *amplifies estimation error and variance* when the mask remains static. Consequently, MA may be inferior to GA in the no-restoration setting, as GA avoids such amplification and accumulates smaller long-term bias. As discussed in Appendix G.4, this explains why w/o GMR the advantage of MA weakens or disappears. Notably, this effect is more pronounced on deeper architectures such as ResNet-18, where the enlarged variance issue becomes even more significant. Moreover, FA further illustrates the difficulty of operating with heterogeneous sub-model aggregation: averaging missing coordinates with zeros induces severe weight shrinkage, leading to consistent degradation across datasets (e.g. near-chance $10\%$ on CIFAR-10).

Table 3: Test accuracy (%) evaluating the joint effect of GMR with various aggregation methods.

| Hetero. | Method | FEMNIST (70k) | | CIFAR-10 (180k) | | ImageNet-100 (250k) | |
|---|---|---|---|---|---|---|---|
| | | IID | Non-IID | IID | Non-IID | IID | Non-IID |
| High | GMR + MA | **82.24** $_{\pm0.14}$ | **81.50** $_{\pm0.28}$ | **84.12** $_{\pm0.18}$ | 80.38 $_{\pm0.47}$ | **61.64** $_{\pm0.37}$ | **58.67** $_{\pm0.64}$ |
| | GMR + GA | 81.33 $_{\pm0.23}$ | 79.74 $_{\pm0.34}$ | 82.86 $_{\pm0.34}$ | 80.02 $_{\pm0.55}$ | 61.33 $_{\pm0.53}$ | 58.07 $_{\pm0.82}$ |
| | GMR + FA | 81.36 $_{\pm0.25}$ | 79.68 $_{\pm0.21}$ | 10.00 $_{\pm0.00}$ | 10.00 $_{\pm0.00}$ | 59.66 $_{\pm0.41}$ | 57.71 $_{\pm0.77}$ |
| | w/o GMR + MA | **81.92** $_{\pm0.084}$ | 77.77 $_{\pm0.17}$ | 80.93 $_{\pm0.26}$ | 71.96 $_{\pm0.30}$ | 54.33 $_{\pm0.44}$ | 50.53 $_{\pm0.86}$ |
| | w/o GMR + GA | 80.02 $_{\pm0.13}$ | **78.71** $_{\pm0.24}$ | **83.09** $_{\pm0.31}$ | **77.88** $_{\pm0.75}$ | **62.37** $_{\pm0.33}$ | **58.67** $_{\pm0.53}$ |
| | w/o GMR + FA | 69.33 $_{\pm0.26}$ | 69.98 $_{\pm0.39}$ | 10.00 $_{\pm0.00}$ | 10.00 $_{\pm0.00}$ | 61.07 $_{\pm0.50}$ | 58.10 $_{\pm0.56}$ |
| Medium | GMR + MA | **82.75** $_{\pm0.22}$ | **82.73** $_{\pm0.29}$ | **84.34** $_{\pm0.17}$ | **81.40** $_{\pm0.42}$ | 62.55 $_{\pm0.46}$ | 61.32 $_{\pm0.47}$ |
| | GMR + GA | 81.96 $_{\pm0.19}$ | 81.45 $_{\pm0.25}$ | 83.91 $_{\pm0.23}$ | 81.07 $_{\pm0.24}$ | **63.16** $_{\pm0.47}$ | **62.08** $_{\pm0.42}$ |
| | GMR + FA | 82.06 $_{\pm0.18}$ | 80.81 $_{\pm0.21}$ | 10.00 $_{\pm0.00}$ | 10.00 $_{\pm0.00}$ | 63.14 $_{\pm0.60}$ | 59.75 $_{\pm0.52}$ |
| | w/o GMR + MA | 81.91 $_{\pm0.10}$ | **79.99** $_{\pm0.26}$ | 83.62 $_{\pm0.14}$ | 78.60 $_{\pm0.25}$ | 56.53 $_{\pm0.46}$ | 55.76 $_{\pm0.86}$ |
| | w/o GMR + GA | 80.42 $_{\pm0.17}$ | 78.75 $_{\pm0.23}$ | **83.96** $_{\pm0.17}$ | **80.26** $_{\pm0.64}$ | **63.45** $_{\pm0.71}$ | **61.40** $_{\pm0.45}$ |
| | w/o GMR + FA | 70.63 $_{\pm0.23}$ | 70.09 $_{\pm0.22}$ | 10.00 $_{\pm0.00}$ | 10.00 $_{\pm0.00}$ | 61.34 $_{\pm0.68}$ | 59.11 $_{\pm0.60}$ |
| Low | GMR + MA | **83.16** $_{\pm0.20}$ | 82.17 $_{\pm0.30}$ | **85.57** $_{\pm0.13}$ | **83.46** $_{\pm0.15}$ | 64.10 $_{\pm0.33}$ | 62.97 $_{\pm0.42}$ |
| | GMR + GA | 82.96 $_{\pm0.19}$ | **82.20** $_{\pm0.27}$ | 85.14 $_{\pm0.17}$ | 82.51 $_{\pm0.24}$ | **64.20** $_{\pm0.27}$ | **63.28** $_{\pm0.45}$ |
| | GMR + FA | 81.39 $_{\pm0.14}$ | 81.29 $_{\pm0.36}$ | 10.00 $_{\pm0.00}$ | 10.00 $_{\pm0.00}$ | 63.38 $_{\pm0.44}$ | 62.10 $_{\pm0.46}$ |
| | w/o GMR + MA | 82.69 $_{\pm0.10}$ | **79.61** $_{\pm0.26}$ | 84.28 $_{\pm0.18}$ | 80.56 $_{\pm0.25}$ | 59.62 $_{\pm0.67}$ | 56.69 $_{\pm1.00}$ |
| | w/o GMR + GA | 81.76 $_{\pm0.24}$ | 79.17 $_{\pm0.14}$ | **84.82** $_{\pm0.24}$ | **83.04** $_{\pm0.23}$ | **64.29** $_{\pm0.42}$ | **61.86** $_{\pm0.71}$ |
| | w/o GMR + FA | 74.67 $_{\pm0.35}$ | 72.77 $_{\pm0.36}$ | 10.00 $_{\pm0.00}$ | 10.00 $_{\pm0.00}$ | 61.06 $_{\pm0.45}$ | 59.28 $_{\pm0.58}$ |

## 7 CONCLUSION

We addressed the challenge that bandwidth-constrained clients hinder convergence and bias optimization in heterogeneous FL. FedGMR tackles this by gradually restoring model capacity under asynchrony and model heterogeneity, with tailored transmission and aggregation. Theory and experiments on FEMNIST, CIFAR-10, and ImageNet-100 confirm faster convergence and higher accuracy, especially under non-IID and highly heterogeneous conditions.

**Reproducibility Statement.** We have taken several measures to ensure the reproducibility of our work. The training procedure, hyperparameter settings are described in detail in Appendix H. In addition, we provide source code and Jupyter notebooks as anonymous supplementary material, where the notebooks contain the main experimental results for direct verification.

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

## A  LLM Usage

Large Language Models (LLMs) were used to aid in the writing and polishing of the manuscript. Specifically, we used an LLM to assist in refining the language, improving readability, and ensuring clarity in various sections of the paper. The model helped with tasks such as sentence rephrasing, grammar checking, and enhancing the overall flow of the text.

It is important to note that the LLM was not involved in the ideation, research methodology, or experimental design. All research concepts, ideas, and analyses were developed and conducted by the authors. The contributions of the LLM were solely focused on improving the linguistic quality of the paper, with no involvement in the scientific content or data analysis.

The authors take full responsibility for the content of the manuscript, including any text generated or polished by the LLM. We have ensured that the LLM-generated text adheres to ethical guidelines and does not contribute to plagiarism or scientific misconduct.

## B  Valid domain of the time difference as approximate gradient for model density

We set the fastest client's training time as the standard $t_o$, and adjust the densities of other clients to match it, ensuring that all clients undergo a similar training process.

To valid the domain of time difference as an approximate gradient for model density, such that $\frac{t'_i - t_i}{\rho'_i - \rho_i} > 0$, we first analyze the composition of a single round of client training time. The total time can be divided into four main components:

1. **Upload time** $t_{u,i}$: The time required for the client to transmit the model to the server, which mainly depends on the model size and client upload speed.

2. **Server processing time**: The time spent by the server to either wait for other clients or aggregate client models.

3. **Download time**: The time required for the server to transmit the global model back to the client.

4. **Client local processing time**: The time spent by the client on local computation.

Based on prior research (Kairouz et al., 2021), the upload time ($t_{u,i}$) is identified as the primary contributor to the overall delay in client training time. Thus, the total training time for client $i$ can be approximated as:

$$t_i = t_{u,i} + t_{o,i} \tag{12}$$

where $t_{o,i}$ represents the combined time of other components.

Next, we model the client's upload process using a simplified approach. Let the original model size be $A$, the average upload speed of client $i$ be $B$, and the available download speed of the server be $C$. Pruning reduces the size of the model and the size of the pruned model can be approximated as $\rho_i A$, where $\rho_i$ is the current model density of the client $i$. The upload time for client $i$ can then be expressed as: $t_{u,i} = \frac{\rho_i A}{\min(B,C)}$.

Substituting $t_{u,i}$ into the total training time formula gives:

$$t_i = \frac{\rho_i A}{\min(B, C)} + t_{o,i}. \tag{13}$$

Now, we define the target density for client $i$ as $\rho'_i$, with updated variables $B', C', t'_{o,i}, t'_i$. The objective is to approach the new client $i$'s training time $t'_i$ to the standard client training time $t_s$.

$$\frac{\rho_i A}{\min(B, C)} + t_{o,i} = \frac{\rho'_i A}{\min(B', C')} + t'_{o,i}. \tag{14}$$

Rearranging for $\rho'_i$, the solution can be expressed as:

$$\rho'_i = \frac{\min(B', C')}{\min(B, C)} \cdot \frac{t'_i - t'_{o,i}}{t_i - t_{o,i}} \cdot \rho_i. \tag{15}$$

Since the upload time $t_{u,i}$ is identified as the primary contributor to the overall delay in one round of client training, we assume $t_i \geq \lambda t_{o,i} > 0$, which indicates that the upload time is at least $\lambda$-times larger than the other components. Furthermore use $t'_i \approx t_s$ for $\rho'_i$ can be bounded as:

$$\frac{\lambda - 1}{\lambda} \cdot \frac{\min(B', C')}{\min(B, C)} \cdot \frac{t_s}{t_i} \cdot \rho_i < \rho'_i < \frac{\lambda}{\lambda - 1} \cdot \frac{\min(B', C')}{\min(B, C)} \cdot \frac{t_s}{t_i} \cdot \rho_i. \tag{16}$$

The two boundaries of $p'_i$ are due to when $\frac{t_{u,i}}{t_{o,i}} = \lambda$ and $\frac{t_{u,i}}{t_{o,i}} \to \infty$. The randomness of $\frac{t_{u,i}}{t_{o,i}}$ and network conditions (e.g., variations in $B$ and $C$) can disrupt the linear relationship between training time and model density.

However,when $t_s - t_i > 0$, as long as:

$$|t_s - t_i| > \left( \frac{\lambda \cdot \min(B_s, C_s)}{(\lambda - 1) \cdot \min(B_i, C_i)} - 1 \right) \cdot t_i. \tag{17}$$

and when $t_s - t_i < 0$, as long as:

$$|t_s - t_i| > \left( 1 - \frac{(\lambda - 1) \cdot \min(B_s, C_s)}{\lambda \cdot \min(B_i, C_i)} \right) \cdot t_i \tag{18}$$

The time difference and the density difference have the same sign, and the time difference can seem as the approximate gradient for density.

Since $t_{u,i}$ is the dominant factor in training delay, the parameter $\lambda$ typically takes a relatively large value. And assume FL environment with stable network conditions, $\frac{\lambda \cdot \min(B_s, C_s)}{(\lambda - 1) \cdot \min(B_i, C_i)} - 1$ and $1 - \frac{(\lambda - 1) \cdot \min(B_s, C_s)}{\lambda \cdot \min(B_i, C_i)}$ approach to 0, which prove the effectiveness of the time difference seem as the approximate gradient for density. Moreover, by replacing single-sample variables with the average of multiple iterations, the randomness caused by $\frac{t_{u,i}}{t_{o,i}}$ and variations in $B$ and $C$ is significantly reduced, making the effectiveness more universally applicable across different conditions.

## C   JUSTIFICATION OF GMR IMPROVING TRAINING SPEED

To support our claim that Gradual Model Restoration (GMR) can accelerate training at certain stages, we present a simplified theoretical analysis.

**Lemma 1.**   *Restoring Model Capacity Improves Training Speed Post-Plateau*

Let $w_t^{(p)}$ denote the weight of a sub-model with density $p < 1$ at time $t$. Let $F(w)$ be the training loss and define the average convergence speed over interval $[t_1, t_2]$ as:

$$\mathcal{A}(t_1, t_2) := -\frac{F(w_{t_2}) - F(w_{t_1})}{t_2 - t_1}. \tag{19}$$

Under the assumption that sub-models have limited representation power, during the interval $[T_{k+n}, T_{k+n+m}]$, the restored model achieves a strictly higher training speed than the prior stagnated phase $[T_k, T_{k+n}]$:

$$\mathcal{A}(T_{k+n}, T_{k+n+m}) > \mathcal{A}(T_k, T_{k+n}). \tag{20}$$

**Assumptions:**

- (Smoothness) $F(\mathbf{w})$ is $L$-smooth, ensuring predictable behavior of gradient descent and enabling analysis via gradient norm.

- (Capacity Hierarchy) Since smaller sub-models are nested within larger ones, sub-models limit representation power: for densities $p_1 < p_2 < p_3$, the corresponding optima satisfies:
$$F(\mathbf{w}_{p_3}^*) < F(\mathbf{w}_{p_2}^*) < F(\mathbf{w}_{p_1}^*). \tag{21}$$

Then we introduce the time interval that restores the model will improve the training speed. At time $t = T_k$, the sub-model $\mathbf{w}_{T_k, p_1}$ reaches its local minimum:

$$F(\mathbf{w}_{T_k, p_1}) = F(\mathbf{w}_{p_1}^*). \tag{22}$$

After a time interval $\Delta t$, the training gain remains zero:

$$\mathcal{A}(T_k, T_{k+n}) \leq 0. \tag{23}$$

If we restore the model to a larger density $p_2 > p_1$ at $T_{k+n}$, the model continues improving and eventually reaches $F(\mathbf{w}_{p_2}^*) < F(\mathbf{w}_{p_1}^*)$ at $T_{k+n+m}$. Thus:

$$\mathcal{A}(T_{k+n}, T_{k+n+m}) = -\frac{F(\mathbf{w}_{p_2}^*) - F(\mathbf{w}_{T_{k+n},p_1})}{T_{k+n+m} - T_{k+n}} > 0. \tag{24}$$

**Conclusion:** During the interval $[T_{k+n}, T_{k+n+m}]$, the restored model exhibits strictly higher training speed than the previous stagnated phase $[T_k, T_{k+n}]$:

$$\mathcal{A}(T_{k+n}, T_{k+n+m}) > \mathcal{A}(T_k, T_{k+n}). \tag{25}$$

**Implication:** This lemma supports the core mechanism of GMR. When a sub-model has plateaued, restoring its capacity enables further progress. Since sub-models are nested within larger models, the restoration allows smooth transitions without retraining from scratch. Although this is an idealized case, it demonstrates that restoring model capacity can improve training speed after stagnation. In other words, the optimal model size in FL is dynamic and should gradually increase as training progresses. GMR thereby enhances training efficiency and convergence in heterogeneous FL settings such as FedGMR.

## D PSEUDOCODE FOR BUFFMASKFEDAVG

We adopt BuffMaskFedAvg to aggregate heterogeneous client models in asynchronous FL. Masks are used to track the pruning status of each neuron's presence across clients, enabling MA guided by a staleness-aware weighting mechanism. The pseudocode is provided in algorithm 4.

---

**Algorithm 3:** BuffMaskFedAvg Aggregation

---

**Input:** Previous model $\mathbf{W}_{k-1}$, buffer $\mathcal{B}$
**Output:** Aggregated model $\mathbf{W}_k$
**for** *each client update* $\mathbf{w}_{i,k} \in \mathcal{B}$ **do**
    compute staleness weight $\beta_{i,k}$;
    derive mask $\mathbf{m}_{i,k}$ from nonzero coordinates of $\mathbf{w}_{i,k}$;
    accumulate: $\mathbf{W}_{\text{cum}} \leftarrow \mathbf{W}_{\text{cum}} + \beta_{i,k}\mathbf{w}_{i,k}$,;
    $\mathbf{M}_{\text{cum}} \leftarrow \mathbf{M}_{\text{cum}} + \beta_{i,k}\mathbf{m}_{i,k}$;
**for** *each parameter* $n = 1, \ldots, N$ **do**
    **if** $\mathbf{M}_{cum}^{(n)} \neq 0$ **then**
        $\mathbf{W}_k^{(n)} \leftarrow \mathbf{W}_{\text{cum}}^{(n)}/\mathbf{M}_{\text{cum}}^{(n)}$;
    **else**
        $\mathbf{W}_k^{(n)} \leftarrow \mathbf{W}_{k-1}^{(n)}$;

---

## E DETAILS FOR EXTRACT THE SUB—MODELS

First, we introduce neuron importance scores for unstructured pruning. Given the client density vector $\mathbf{P}_k = \{\rho_{i,k}\}_{i=1}^C$, the corresponding binary masks $\mathbf{M}_k = \{m_{i,k}\}_{i=1}^C$ are generated accordingly. Following Molchanov et al. (2019), pruning is guided by importance scores, which quantify a neuron's contribution as the squared difference in prediction error between the full and pruned models.

$$\mathbf{E}_k = (F(\mathbf{W}_k) - F(\mathbf{W}_k \odot \mathbf{m}))^2 \tag{26}$$

where $\mathbf{E}_k$ represents the pruning error, $\odot$ denotes element-wise multiplication, and $\mathbf{m}$ is a binary mask indicating whether a neuron in $\mathbf{W}_k$ is pruned (0) or retained (1).

To simplify computation, a first-order Taylor expansion is used Molchanov et al. (2019); Liu et al. (2021), approximating neuron importance as:

$$\mathbf{I}_k \approx (\mathbf{G}_k \odot \mathbf{W}_k)^2 \tag{27}$$

where $\mathbf{I}_k$ is the importance score matrix for $\mathbf{W}_k$, computed as the squared product of each neuron's weight and its gradient.

In the FL scenario, transmitting full gradient data imposes a burden on RCCs. To mitigate this, gradients are approximated using weight changes. By storing the old global model weight $\mathbf{W}_{k-1}$, the gradient is approximated as $\mathbf{G}_k \approx \mathbf{W}_k - \mathbf{W}_{k-1}$, and neuron importance is calculated as:

$$\mathbf{I}_k \approx \left( (\mathbf{W}_k - \mathbf{W}_{k-1}) \odot \mathbf{W}_k \right)^2 \tag{28}$$

Then, based on Eq. 28, we compute neuron importance scores, sort neurons accordingly, and derive masks consistent with the target density. In asynchronous settings, however, frequent aggregation makes per-round sorting computationally expensive. To reduce overhead, masks are updated only every few rounds.

---

**Algorithm 4:** Extract the sub-models

---

**Input:** Global model $\mathbf{W}_k$, old model $\mathbf{W}_{k-1}$, client densities $\mathbf{P}_k = \{\rho_{i,k}\}_{i=1}^C$
**Output:** Client masks $\mathbf{M} = \{\mathbf{m}_{i,k}\}_{i=1}^C$
**def** $\mathbf{W}_k, \mathbf{W}_{k-1}, \mathbf{P}_k$**:**
    Compute importance scores $\mathbf{I}_k$ by Eq. 28;
    Sort neurons by $\mathbf{I}_k$ (descending);
    **for** *each client $i$* **do**
        assign top-$\rho_{i,k}$ fraction of neurons as $m_{i,k}$;
    return $\mathbf{M}$

---

During these intervals, the server utilizes the masks in $\mathbf{M}$ to prune the global model to the corresponding client models, thereby reducing the computational overhead.

The client densities list is represented as $\mathbf{P}$, calculated using the GMR strategy. The algorithm for Federated Model Pruning is detailed in Eq. 4, with the output being the corresponding masks $\mathbf{M}$. The neurons are then sorted based on their importance scores, the masks are filled with sorted neurons to meet the target density and saved.

For simplicity, we assume that $\mathbf{P}$ is sorted in descending order, meaning the client models satisfy the nested relation: $(\mathbf{w}^k \odot \mathbf{m}_1) \supseteq (\mathbf{w}^k \odot \mathbf{m}_2) \supseteq \cdots \supseteq (\mathbf{w}^k \odot \mathbf{m}_M)$. The bottleneck of the algorithm lies in the sorting operation, with an overall time complexity of $O(|\mathbf{w}^k| \log |\mathbf{w}^k|)$.

## F INCREMENTAL MODEL SPLITTING: TRANSMITTING VARYING-SIZE MODELS TO CLIENTS.

Moreover, unlike MHFL methods that transmit the full model and require clients to extract their sub-models locally, we avoid sending each sub-model separately. Instead, we employ unstructured pruning to partition the global model based on the nested relation among different densities. The pruned global model is decomposed into a set of non-overlapping *increments*, $\Delta W_k = \{\Delta w_{j,k}\}_{j=1}^M$, with corresponding index sets $\mathrm{Idx}_k = \{\mathrm{idx}_{i,k}\}_{i=1}^C$. Each client reconstructs its target sub-model by summing the required increments:

$$\mathbf{w}_{i,k_i'} = \sum_{j \in \mathrm{idx}_{i,k}} \Delta \mathbf{w}_{j,k}.$$

This Incremental Model Splitting (IMS) design removes redundancy and substantially reduces server-to-client bandwidth, at the cost of light splitting and merging operations performed once per round. The complete pseudocode is provided in algorithm 5.

Inside `IMSs`, the mask set $\mathbf{M}$ is obtained from densities $\mathbf{P}_k$ via `FMP`. In practice, $\mathbf{M}$ is not recomputed every round due to cost; it is refreshed only every $k_{\mathrm{rest}}$ rounds and reused in between:

$$\mathbf{M} \leftarrow \begin{cases} \mathtt{FMP}(\mathbf{W}_k, \mathbf{W}_{k-1}, \mathbf{P}_k), & \text{if } k \bmod k_{\mathrm{rest}} = 0, \\ \mathbf{M} \text{ (reuse previous)}, & \text{otherwise.} \end{cases}$$

`IMSs` then uses the current $\mathbf{M}$ to form increments and indices. We embed `FMP` inside `IMSs` only for brevity in the top-level pseudocode; implementation-wise, it can equivalently be invoked outside and passed as input.

---

**Algorithm 5:** Incremental Model Splitting

---

**Input:** $\mathbf{W}_k$ and client density set $\mathbf{P}_k$
**Output:** model increments set $\Delta\mathbf{W}_k$ and index set $\mathbf{Idx}_k$
**def** IMSs $(\mathbf{W}_k, \mathbf{P}_k)$:
    $\mathbf{M} \leftarrow \mathbf{FMP}(\mathbf{W}_k, \mathbf{W}_{k-1}, \mathbf{P}_k)$;
    $\mathbf{M}_s, I = \text{sort}(\text{density}(\mathbf{M}), \text{descending=False})$;
    **for** $j \in [1, len(\mathbf{M}_k)]$ **do**
      $\Delta\mathbf{W}_k[j] = \mathbf{W}_k \odot (\mathbf{M}_s[j] - \mathbf{M}_s[j-1])$;
      $\mathbf{Idx}_k[I[j]] = [1, j]$;
    **return** $\Delta\mathbf{W}_k, \mathbf{Idx}_k$
**def** IMSc $(\mathbf{idx}_{i,k})$:
    $\{\Delta\mathbf{w}_{i,k}\}_{i=1}^{\mathbf{idx}_{i,k}} = \text{Download}(\mathbf{idx}_{i,k})$;
    $\mathbf{w}_{i,k} = \sum_{i \in \mathbf{idx}_{i,k}} \Delta\mathbf{w}_{i,k}$;
    **return** $\mathbf{w}_i^k$

---

## G  CONVERGENCE ANALYSIS

We consider $C$ clients and a global model whose coordinates we index by neurons $n \in \{1, \ldots, N\}$ (with $N = d$). MHFL assigns different-sized sub-models to clients via binary masks. To isolate the effect of *model sparsity* (and avoid staleness confounds), we conduct the analysis in the *synchronous* setting, Client $i$ at round $k$ holds a binary mask $m_{i,k} \in \{0,1\}^N$. Let the active–client set and its size at coordinate $n$ be

$$C_k^{(n)} := \{\, i : m_{i,k}^{(n)} = 1 \,\}, \qquad \Gamma_k^{(n)} := |C_k^{(n)}| \, (\geq 1).$$

### G.1  DIFFERENT AGGREGATION METHODS IN MODEL-HETEROGENEOUS FL

Below we give the three aggregation rules used in our analysis.

**Mask-aware aggregation (MA).** Simplify BuffMaskFedAvg to a **mask-aware aggregation (MA)** operator. MA averages *only* over clients that retain each coordinate:

$$W_{k+1}^{(n)} = \frac{1}{\Gamma_k^{(n)}} \sum_{i \in C_k^{(n)}} w_{i,k}^{(n)} \quad \Longleftrightarrow \quad W_{k+1} = \Big(\sum_{i=1}^{C} w_{i,k} \odot m_{i,k}\Big) \odot \pi_k, \;\; \pi_k^{(n)} := \frac{1}{\Gamma_k^{(n)}}. \quad (29)$$

**Gradient-average aggregation (GA).** For comparison, GA applies masked *gradients* but still divides by the *total* number of clients:

$$W_{k+1} = W_k - \gamma \cdot \frac{1}{C} \sum_{i=1}^{C} \sum_{\tau=1}^{\mathcal{T}} \nabla F_i\big(w_{i,k,\tau-1}, \xi_{i,\tau-1}\big) \odot m_{i,k}, \quad (30)$$

so the per–coordinate step size scales with $\Gamma_k^{(n)}/C$.

**FedAvg (FA) with zero padding.** Standard FedAvg averages local models; with structural heterogeneity it is commonly implemented by zero–padding pruned coordinates:

$$W_{k+1} = \frac{1}{C} \sum_{i=1}^{C} \big(w_{i,k} \odot m_{i,k}\big), \qquad \text{equivalently} \qquad W_{k+1}^{(n)} = \frac{1}{C} \sum_{i=1}^{C} w_{i,k}^{(n)} \mathbf{1}\{n \in m_{i,k}\}. \quad (31)$$

When $\Gamma_k^{(n)} \ll C$, many zeros dilute the average, shrinking $W_{k+1}^{(n)}$ toward 0; as density changes (e.g., under restoration), the effective averaging scale $\Gamma_k^{(n)}/C$ *jumps*, which can destabilize training.

Unlike MA/GA, FA with zero padding does *not* correspond to a stochastic gradient step on any fixed smooth objective. Indeed,

$$W_{k+1}^{(n)} = \frac{1}{C} \sum_{i=1}^{C} w_{i,k}^{(n)} \mathbf{1}\{n \in m_{i,k}\} = \frac{\Gamma_k^{(n)}}{C} \underbrace{\left( \frac{1}{\Gamma_k^{(n)}} \sum_{i \in C_k^{(n)}} w_{i,k}^{(n)} \right)}_{\text{MA average at coord. } n}.$$

Thus FA equals the MA average multiplied by a *coverage–dependent* factor $\Gamma_k^{(n)}/C \in [0,1]$, i.e., a coordinate/round–dependent contraction that depends on participation patterns rather than objective geometry. Consequently, FA cannot be written as $W_{k+1} = W_k - \gamma \widehat{g}_k$ with $\mathbb{E}[\widehat{g}_k] = \nabla F(W_k)$ for a fixed $F$, and is therefore not amenable to a clean gradient–based convergence analysis in the mask–misaligned regime. This motivates focusing on MA/GA in our theory.

**Two stage aggregation** Similar to the analytical steps in Zhou et al. (2023), we examine convergence from the perspective of neuron-wise aggregation. However, we do not agree with the concept of the *minimum coverage index* proposed in Zhou et al. (2023), as it merely reflects the scaled minimum density among all clients. Instead, We argue that the heterogeneity of model densities across clients and communication rounds exerts an accumulative effect on convergence.

To validate this hypothesis, we leverage the nested structure induced by shared pruning strategies and redefine the model aggregation process as a two-stage procedure. Specifically, clients are first grouped into $\mathcal{G}$ structural groups $\{\mathcal{C}_g\}_{g=1}^{\mathcal{G}}$, where each group shares a common mask $m_{g,k}$ and density $p_g$. Let $\mathcal{G}_k^{(n)}$ denote the set of groups whose masks contain neuron $n$.

Then the two-stage aggregation is given by: (1) Within-group aggregation (using traditional FedAvg since structures are uniform), followed by (2) Parameter-wise aggregation across groups to obtain the global model:

$$\mathbf{w}_k^{(n)} = \frac{1}{\Gamma_k^{(n)}} \sum_{g \in \mathcal{G}_k^{(n)}} |\mathcal{C}_g| \cdot \mathbf{w}_{g,k}^{(n)}, \quad \text{where } \mathbf{w}_{g,k} := \frac{1}{|\mathcal{C}_g|} \sum_{i \in \mathcal{C}_g} \mathbf{w}_{i,k} \odot m_{g,k}. \tag{32}$$

This formulation is equivalent to Eq. 29, but recast to emphasize the structural grouping. For comparison, the corresponding GA becomes:

$$\mathbf{w}_k^{(n)} = \frac{1}{C} \sum_{g \in \mathcal{G}_k^{(n)}} |\mathcal{C}_g| \cdot \mathbf{w}_{g,k}^{(n)}, \quad \text{where } \mathbf{w}_{g,k} := \frac{1}{|\mathcal{C}_g|} \sum_{i \in \mathcal{C}_g} \mathbf{w}_{i,k} \odot m_{g,k}. \tag{33}$$

Thus MA *preserves per–coordinate scale* by normalizing with the actual coverage $\Gamma_k^{(n)}$ (maintaining gradient completeness), whereas GA divides by $C$ and therefore underweights partially covered coordinates.

## G.2 Convergence analysis based on Mask-aware aggregation

### G.2.1 Expansion toward the Initial Convergence Main Term

**Step 1 (L-smoothness).** Under Assumption 1, for any round $k$, we have:

$$F(\mathbf{W}_{k+1}) \leq F(\mathbf{W}_k) + \langle \nabla F(\mathbf{W}_k), \mathbf{W}_{k+1} - \mathbf{W}_k \rangle + \frac{L}{2} \|\mathbf{W}_{k+1} - \mathbf{W}_k\|^2. \tag{34}$$

Taking expectation on both sides:

$$\mathbb{E}[F(\mathbf{W}_{k+1})] - \mathbb{E}[F(\mathbf{W}_k)] \leq \mathbb{E}\langle \nabla F(\mathbf{W}_k), \mathbf{W}_{k+1} - \mathbf{W}_k \rangle + \frac{L}{2} \mathbb{E}\|\mathbf{W}_{k+1} - \mathbf{W}_k\|^2. \tag{35}$$

**Step 2** we then analyze the global update expression, for any region $n$, we have:

$$\mathbf{W}_k^{(n)} - \mathbf{W}_{k+1}^{(n)} = \frac{1}{\Gamma_k^{(n)}} \sum_{i \in C_k^{(n)}} \sum_{\tau=1}^{\mathcal{T}} \gamma \nabla F_i^{(n)}(\mathbf{w}_{i,k,\tau-1}, \xi_{i,\tau-1}). \tag{36}$$

**Step 3** We next analyze the inner product. $\mathbb{E} \langle \nabla F(\mathbf{W}_k), \mathbf{W}_{k+1} - \mathbf{W}_k \rangle$ by considering a sum of inner products over $n$ regions. We have:

$$\mathbb{E} \langle \nabla F(\mathbf{W}_k), \mathbf{W}_{k+1} - \mathbf{W}_k \rangle$$

$$= \sum_{n=1}^{N} \mathbb{E} \left\langle \nabla F^{(n)}(\mathbf{W}_k), \mathbf{W}_{k+1}^{(n)} - \mathbf{W}_k^{(n)} \right\rangle$$

(According to the properties of the inner product, split the inner product into $K$ structural regions)

$$= \sum_{n=1}^{N} \mathbb{E} \left\langle \nabla F^{(n)}(\mathbf{W}_k), -\frac{1}{\Gamma_k^{(n)}} \sum_{i \in C_k^{(n)}} \sum_{\tau=1}^{\mathcal{T}} \gamma \nabla F_i^{(n)}(\mathbf{w}_{i,k,\tau-1}, \xi_{i,\tau-1}) \right\rangle$$

$$= \sum_{n=1}^{N} \mathbb{E} \left\langle \nabla F^{(n)}(\mathbf{W}_k), -\frac{1}{\Gamma_k^{(n)}} \sum_{i \in C_k^{(n)}} \sum_{\tau=1}^{\mathcal{T}} \gamma \, \mathbb{E} \left[ \nabla F_i^{(n)}(\mathbf{w}_{i,k,\tau-1}, \xi_{i,\tau-1}) \mid \mathbf{W}_k \right] \right\rangle$$

(Use the tower rule of expectation over $\mathbf{W}_k$)

$$= \sum_{n=1}^{N} \mathbb{E} \left\langle \nabla F^{(n)}(\mathbf{W}_k), -\frac{1}{\Gamma_k^{(n)}} \sum_{i \in C_k^{(n)}} \sum_{\tau=1}^{\mathcal{T}} \gamma \nabla F_i^{(n)}(\mathbf{w}_{i,k,\tau-1}) \right\rangle$$

(Now gradients are conditional means)

$$= - \sum_{n=1}^{N} \mathbb{E} \left\langle \nabla F^{(n)}(\mathbf{W}_k), \frac{1}{\Gamma_k^{(n)}} \sum_{i \in C_k^{(n)}} \sum_{\tau=1}^{\mathcal{T}} \gamma \left( \nabla F^{(n)}(\mathbf{W}_k) + \nabla F_i^{(n)}(\mathbf{w}_{i,k,\tau-1}) - \nabla F^{(n)}(\mathbf{W}_k) \right) \right\rangle$$

(Insert $\nabla F^{(n)}(\mathbf{W}_k) - \nabla F^{(n)}(\mathbf{W}_k) = 0$ to isolate descent direction)

$$= - \sum_{n=1}^{N} \mathbb{E} \left\langle \nabla F^{(n)}(\mathbf{W}_k), \gamma \mathcal{T} \nabla F^{(n)}(\mathbf{W}_k) \right\rangle \quad \text{(Because } \frac{1}{\Gamma_k^{(n)}} \sum_{i \in C_k^{(n)}} \nabla F^{(n)}(\mathbf{W}_k) = \nabla F^{(n)}(\mathbf{W}_k))$$

$$- \sum_{n=1}^{N} \mathbb{E} \left\langle \nabla F^{(n)}(\mathbf{W}_k), \frac{1}{\Gamma_k^{(n)}} \sum_{i \in C_k^{(n)}} \sum_{\tau=1}^{\mathcal{T}} \gamma \left( \nabla F_i^{(n)}(\mathbf{w}_{i,k,\tau-1}) - \nabla F^{(n)}(\mathbf{W}_k) \right) \right\rangle \quad (37)$$

(Separate the true descent term and the deviation from it)

For the first term on the right-hand side of Eq. 37, it is easy to see that:

$$- \sum_{n=1}^{N} \mathbb{E} \left\langle \nabla F^{(n)}(\mathbf{W}_k), \gamma \mathcal{T} \nabla F^{(n)}(\mathbf{W}_k) \right\rangle$$

$$= - \gamma \mathcal{T} \sum_{n=1}^{N} \mathbb{E} \left\| \nabla F^{(n)}(\mathbf{W}_k) \right\|^2 = -\gamma \mathcal{T} \mathbb{E} \left\| \nabla F(\mathbf{W}_k) \right\|^2. \quad (38)$$

For the second term on the right-hand side of Eq. 37, we apply the Cauchy–Schwarz inequality: $\langle a, b \rangle \leq \frac{1}{2} \|a\|^2 + \frac{1}{2} \|b\|^2$ to obtain:

$$- \sum_{n=1}^{N} \mathbb{E} \left\langle \nabla F^{(n)}(\mathbf{W}_k), \frac{1}{\Gamma_k^{(n)}} \sum_{i \in C_k^{(n)}} \sum_{\tau=1}^{\mathcal{T}} \gamma \left( \nabla F_i^{(n)}(\mathbf{w}_{i,k,\tau-1}) - \nabla F^{(n)}(\mathbf{W}_k) \right) \right\rangle$$

$$= - \sum_{n=1}^{N} \gamma \mathcal{T} \cdot \mathbb{E} \left\langle \nabla F^{(n)}(\mathbf{W}_k), \frac{1}{\mathcal{T} \Gamma_k^{(n)}} \sum_{i \in C_k^{(n)}} \sum_{\tau=1}^{\mathcal{T}} \left( \nabla F_i^{(n)}(\mathbf{w}_{i,k,\tau-1}) - \nabla F^{(n)}(\mathbf{W}_k) \right) \right\rangle$$

$$\leq \frac{\gamma \mathcal{T}}{2} \sum_{n=1}^{N} \mathbb{E} \left\| \nabla F^{(n)}(\mathbf{W}_k) \right\|^2$$

$$+ \frac{\gamma \mathcal{T}}{2} \sum_{n=1}^{N} \mathbb{E} \left\| \frac{1}{\mathcal{T} \Gamma_k^{(n)}} \sum_{i \in C_k^{(n)}} \sum_{\tau=1}^{\mathcal{T}} \left( \nabla F_i^{(n)}(\mathbf{w}_{i,k,\tau-1}) - \nabla F^{(n)}(\mathbf{W}_k) \right) \right\|^2. \tag{39}$$

Combining Eq. 37, Eq. 38, Eq. 39 and , we obtain:

$$\mathbb{E} \left\langle \nabla F(\mathbf{W}_k), \mathbf{w}_{k+1} - \mathbf{w}_k \right\rangle$$

$$\leq - \gamma \mathcal{T} \mathbb{E} \left\| \nabla F(\mathbf{W}_k) \right\|^2 + \frac{\gamma \mathcal{T}}{2} \sum_{n=1}^{N} \mathbb{E} \left\| \nabla F^{(n)}(\mathbf{W}_k) \right\|^2$$

$$+ \frac{\gamma \mathcal{T}}{2} \sum_{n=1}^{N} \mathbb{E} \left\| \frac{1}{\mathcal{T} \Gamma_k^{(n)}} \sum_{i \in C_k^{(n)}} \sum_{\tau=1}^{\mathcal{T}} \left( \nabla F_i^{(n)}(\mathbf{w}_{i,k,\tau-1}) - \nabla F^{(n)}(\mathbf{W}_k) \right) \right\|^2$$

$$\leq - \frac{\gamma \mathcal{T}}{2} \sum_{n=1}^{N} \mathbb{E} \left\| \nabla F^{(n)}(\mathbf{W}_k) \right\|^2$$

$$+ \frac{\gamma \mathcal{T}}{2} \sum_{n=1}^{N} \mathbb{E} \left\| \frac{1}{\mathcal{T} \Gamma_k^{(n)}} \sum_{i \in C_k^{(n)}} \sum_{\tau=1}^{\mathcal{T}} \left( \nabla F_i^{(n)}(\mathbf{w}_{i,k,\tau-1}) - \nabla F^{(n)}(\mathbf{W}_k) \right) \right\|^2. \tag{40}$$

**Step 4:** Analysis of the upperbound for $\frac{L}{2} \mathbb{E} \|\mathbf{W}_{k+1} - \mathbf{W}_k\|^2$.

$$\frac{L}{2} \mathbb{E} \|\mathbf{w}_{k+1} - \mathbf{w}_k\|^2$$

$$= \frac{L}{2} \sum_{n=1}^{N} \mathbb{E} \left\| \frac{1}{\Gamma_k^{(n)}} \sum_{i \in C_k^{(n)}} \sum_{\tau=1}^{\mathcal{T}} \gamma \nabla F_i^{(n)}(\mathbf{w}_{i,k,\tau-1}, \xi_{i,\tau-1}) \right\|^2$$

$$\leq \frac{3L}{2} \sum_{n=1}^{N} \mathbb{E} \left\| \frac{1}{\Gamma_k^{(n)}} \sum_{i \in C_k^{(n)}} \sum_{\tau=1}^{\mathcal{T}} \gamma \left[ \nabla F_i^{(n)}(\mathbf{w}_{i,k,\tau-1}, \xi_{i,\tau-1}) - \nabla F_i^{(n)}(\mathbf{w}_{i,k,\tau-1}) \right] \right\|^2$$

$$+ \frac{3L}{2} \sum_{n=1}^{N} \mathbb{E} \left\| \frac{1}{\Gamma_k^{(n)}} \sum_{i \in C_k^{(n)}} \sum_{\tau=1}^{\mathcal{T}} \gamma \left[ \nabla F_i^{(n)}(\mathbf{w}_{i,k,\tau-1}) - \nabla F_i^{(n)}(\mathbf{W}_k) \right] \right\|^2$$

$$+ \frac{3L}{2} \sum_{n=1}^{N} \mathbb{E} \left\| \frac{1}{\Gamma_k^{(n)}} \sum_{i \in C_k^{(n)}} \sum_{\tau=1}^{\mathcal{T}} \gamma \nabla F_i^{(n)}(\mathbf{W}_k) \right\|^2.$$

$$\leq \frac{3L \mathcal{T}^2 \gamma^2}{2} \sum_{n=1}^{N} \mathbb{E} \left\| \frac{1}{\mathcal{T} \Gamma_k^{(n)}} \sum_{i \in C_k^{(n)}} \sum_{\tau=1}^{\mathcal{T}} \left[ \nabla F_i^{(n)}(\mathbf{w}_{i,k,\tau-1}, \xi_{i,\tau-1}) - \nabla F_i^{(n)}(\mathbf{w}_{i,k,\tau-1}) \right] \right\|^2$$

$$+ \frac{3L \mathcal{T}^2 \gamma^2}{2} \sum_{n=1}^{N} \mathbb{E} \left\| \frac{1}{\mathcal{T} \Gamma_k^{(n)}} \sum_{i \in C_k^{(n)}} \sum_{\tau=1}^{\mathcal{T}} \left[ \nabla F_i^{(n)}(\mathbf{w}_{i,k,\tau-1}) - \nabla F_i^{(n)}(\mathbf{W}_k) \right] \right\|^2$$

$$+ \frac{3L\mathcal{T}^2\gamma^2}{2} \sum_{n=1}^{N} \mathbb{E} \left\| \frac{1}{\Gamma_k^{(n)}} \sum_{i \in C_k^{(n)}} \nabla F_i^{(n)}(\mathbf{W}_k) \right\|^2 . \tag{41}$$

**Step 5** Combining the two upperbounds, we have:

$$\mathbb{E}[F(\mathbf{w}_{k+1})] - \mathbb{E}[F(\mathbf{W}_k)]$$

$$\leq \mathbb{E} \langle \nabla F(\mathbf{W}_k), \mathbf{w}_{k+1} - \mathbf{w}_k \rangle + \frac{L}{2} \mathbb{E} \|\mathbf{W}_{k+1} - \mathbf{W}_k\|^2$$

$$\leq - \frac{\gamma\mathcal{T}}{2} \sum_{n=1}^{N} \mathbb{E} \left\| \nabla F^{(n)}(\mathbf{W}_k) \right\|^2$$

$$+ \frac{\gamma\mathcal{T}}{2} \sum_{n=1}^{N} \mathbb{E} \left\| \frac{1}{\mathcal{T}\Gamma_k^{(n)}} \sum_{i \in C_k^{(n)}} \sum_{\tau=1}^{\mathcal{T}} \left( \nabla F_i^{(n)}(\mathbf{w}_{i,k,\tau-1}) - \nabla F^{(n)}(\mathbf{W}_k) \right) \right\|^2$$

$$+ \frac{3L\mathcal{T}^2\gamma^2}{2} \sum_{n=1}^{N} \mathbb{E} \left\| \frac{1}{\mathcal{T}\Gamma_k^{(n)}} \sum_{i \in C_k^{(n)}} \sum_{\tau=1}^{\mathcal{T}} \left[ \nabla F_i^{(n)}(\mathbf{w}_{i,k,\tau-1}, \xi_{i,\tau-1}) - \nabla F_i^{(n)}(\mathbf{w}_{i,k,\tau-1}) \right] \right\|^2$$

$$+ \frac{3L\mathcal{T}^2\gamma^2}{2} \sum_{n=1}^{N} \mathbb{E} \left\| \frac{1}{\mathcal{T}\Gamma_k^{(n)}} \sum_{i \in C_k^{(n)}} \sum_{\tau=1}^{\mathcal{T}} \left[ \nabla F_i^{(n)}(\mathbf{w}_{i,k,\tau-1}) - \nabla F_i^{(n)}(\mathbf{W}_k) \right] \right\|^2$$

$$+ \frac{3L\mathcal{T}^2\gamma^2}{2} \sum_{n=1}^{N} \mathbb{E} \left\| \frac{1}{\Gamma_k^{(n)}} \sum_{i \in C_k^{(n)}} \nabla F_i^{(n)}(\mathbf{W}_k) \right\|^2 . \tag{42}$$

G.2.2 PROOF OF THE PART OF CONVERGENCE MAIN TERM

Similar to Zhou et al. (2023), we leverage the fact that the $L_2$ norm of the gradient over the entire model can be decomposed as the sum of norms over different parameter regions (i.e., model blocks $n = 1, \ldots, N$). This allows us to analyze the full weight matrix rather than focusing on sub-vectors.

However, unlike Zhou et al. (2023), we do not decompose the model drift into the model reduction noise and the local training drift when characterizing the difference between the local and global models at any local epoch $\tau$.

Our rationale is that, under mask-aware aggregation, each sub-model only contributes to its corresponding sub-region of the global model. Hence, the reduction in model size during server splitting does not reflect actual model degradation. We therefore define model drift solely as the local training drift:

$$\|\mathbf{w}_{i,k,\tau} \odot \mathbf{m}_{i,k} - \mathbf{W}_k \odot \mathbf{m}_{i,k}\| . \tag{43}$$

In this formulation, the model reduction noise is excluded because sub-models do not affect the masked-out parameters. The convergence degradation arises from partial updates and limited gradient coverage, not from global model reduction.

**Lemma 2.** *Group-wise Client Model Deviation Bound*

We aim to bound the deviation within a group $g$ between the average client model $\mathbf{w}_{k,g,\tau}$ and the group-level model $\mathbf{W}_k$:

$$\frac{1}{\mathcal{T}} \sum_{\tau=1}^{\mathcal{T}} \mathbb{E} \left\| (\mathbf{w}_{g,k,\tau-1} - \mathbf{W}_k) \odot m_{g,k} \right\|^2 \leq \frac{2\mathcal{T}^2}{3} \gamma^2 f_1^2(p_g) G \tag{44}$$

Step 1: Expand as average over clients in group $g$.

$$\frac{1}{\mathcal{T}} \sum_{\tau=1}^{\mathcal{T}} \mathbb{E} \left\| (\mathbf{w}_{g,k,\tau-1} - \mathbf{W}_k) \odot m_{g,k} \right\|^2 = \frac{1}{\mathcal{T}} \sum_{\tau=1}^{\mathcal{T}} \mathbb{E} \left\| \frac{1}{|C_g|} \sum_{n \in C_g} [(\mathbf{w}_{i,k,\tau-1} - \mathbf{W}_k) \odot m_{g,k}] \right\|^2 \tag{45}$$

Step 2: Apply Jensen's inequality over clients.

$$\leq \frac{1}{|C_g|} \sum_{n \in C_g} \frac{1}{\mathcal{T}} \sum_{\tau=1}^{\mathcal{T}} \mathbb{E} \left\| (\mathbf{w}_{i,k,\tau-1} - \mathbf{W}_k) \odot m_{g,k} \right\|^2 \tag{46}$$

Step 3: Use SGD update rule for client $n$. Using SGD updates and bounded gradients:

$$(\mathbf{w}_{i,k,\tau-1} - \mathbf{W}_k) \odot m_{g,k} = -\gamma \sum_{s=0}^{\tau-1} \nabla F_i(\mathbf{w}_{i,k,s}, \xi_{i,s}) \odot m_{g,k} \tag{47}$$

Then apply it to the first term

$$\frac{1}{\mathcal{T}} \sum_{\tau=1}^{\mathcal{T}} \mathbb{E} \left\| (\mathbf{w}_{i,k,\tau-1} - \mathbf{W}_k) \odot m_{g,k} \right\|^2$$

$$= \frac{1}{\mathcal{T}} \sum_{\tau=1}^{\mathcal{T}} \mathbb{E} \left\| \gamma \sum_{s=0}^{\tau-1} \nabla F_i(\mathbf{w}_{i,k,s}, \xi_{i,s}) \odot m_{g,k} \right\|^2$$

$$\leq \frac{1}{\mathcal{T}} \sum_{n=1}^{N} (\tau - 1) \sum_{s=0}^{\tau-1} \mathbb{E} \left\| \gamma \nabla F_i(\mathbf{w}_{i,k,s}, \xi_{i,s}) \odot m_{g,k} \right\|^2$$

$$= \frac{1}{\mathcal{T}} \sum_{n=1}^{N} (\tau - 1) \sum_{s=0}^{\tau-1} \gamma^2 f_1^2(p_g) G^2$$

$$= \frac{\mathcal{T}^2 - 1}{3} \gamma^2 f_1^2(p_g) G$$

$$\leq \frac{\mathcal{T}^2 \gamma^2}{3} f_1^2(p_g) G \tag{48}$$

To simplify the expression, the structural influence of the mask $m_{g,k}$ is encapsulated by the density-aware function $f_1^2(p_g)$.

**Lemma 3.** *Group-aware Gradient Drift Bound under Parameter-wise Aggregation*

We aim to bound the squared gradient drift between the average local model and the global model:

$$\sum_{n=1}^{N} \mathbb{E} \left\| \frac{1}{\mathcal{T}} \sum_{\tau=1}^{\mathcal{T}} \frac{1}{\Gamma_k^{(n)}} \sum_{n \in C_k^{(n)}} \left[ \nabla F_i^{(n)}(\mathbf{w}_{i,k,\tau-1}) - \nabla F_i^{(n)}(\mathbf{W}_k) \right] \right\|^2 \leq \frac{L^2 \gamma^2 \mathcal{T}^2}{3} \sum_{g=1}^{\mathcal{G}} \frac{|C_g|}{\Gamma_{g,k}^*} f_1^2(p_g) G^2 \tag{49}$$

where $\sum_{g=1}^{\mathcal{G}} \frac{|\mathcal{C}_g|}{\Gamma_{g,k}^*}$ quantifies the aggregation discrepancy introduced by structural heterogeneity among client models, which amplifies gradient drift during aggregation. Moreover, $\sum_{g=1}^{\mathcal{G}} \frac{|\mathcal{C}_g|}{\Gamma_{g,k}^*} f^2(p_g)$ can be interpreted as the magnification of sparsity-induced sub-vector errors.

**Proof.**

Step 1: Jensen over local training epochs. By Jensen's inequality over the local training epochs:

$$\sum_{n=1}^{N} \mathbb{E} \left\| \frac{1}{\mathcal{T}} \sum_{\tau=1}^{\mathcal{T}} \frac{1}{\Gamma_k^{(n)}} \sum_{n \in C_k^{(n)}} \left[ \nabla F_i^{(n)}(\mathbf{w}_{i,k,\tau-1}) - \nabla F_i^{(n)}(\mathbf{W}_k) \right] \right\|^2$$

$$\leq \frac{1}{\mathcal{T}} \sum_{\tau=1}^{\mathcal{T}} \sum_{n=1}^{N} \mathbb{E} \left\| \frac{1}{\Gamma_k^{(n)}} \sum_{n \in C_k^{(n)}} \left[ \nabla F_i^{(n)}(\mathbf{w}_{i,k,\tau-1}) - \nabla F_i^{(n)}(\mathbf{W}_k) \right] \right\|^2 \tag{50}$$

Step 2: Apply Group wised aggregation Because clients in a group have the same structure,

$$= \frac{1}{\mathcal{T}} \sum_{\tau=1}^{\mathcal{T}} \sum_{n=1}^{N} \mathbb{E} \left\| \frac{1}{\Gamma_k^{(n)}} \sum_{g \in \mathcal{G}_k^{(n)}} |C_g| \left[ \nabla F_g^{(n)}(\mathbf{w}_{g,k,\tau-1}) - \nabla F_g^{(n)}(\mathbf{W}_k) \right] \right\|^2 \tag{51}$$

Step 3: Expand deviation and simplify using Jensen

Due to $\frac{1}{\Gamma_k^{(n)}} \sum_{g \in \mathcal{G}_k^{(n)}} |C_g| = 1$, apply Weighted Jensen's inequality:

$$\leq \frac{L^2}{\mathcal{T}} \sum_{\tau=1}^{\mathcal{T}} \sum_{n=1}^{N} \frac{1}{\Gamma_k^{(n)}} \sum_{g \in \mathcal{G}_k^{(n)}} |C_g| \cdot \mathbb{E} \left\| \nabla F_g^{(n)}(\mathbf{w}_{g,k,\tau-1}) - \nabla F_g^{(n)}(\mathbf{W}_k) \right\|^2 \tag{52}$$

Swap the sums of $g$ and $n$:

$$= \frac{L^2}{\mathcal{T}} \sum_{\tau=1}^{\mathcal{T}} \sum_{g=1}^{\mathcal{G}} \sum_{n \in S_{g,k}} \frac{|C_g|}{\Gamma_k^{(n)}} \cdot \mathbb{E} \left\| \nabla F_g^{(n)}(\mathbf{w}_{g,k,\tau-1}) - \nabla F_g^{(n)}(\mathbf{W}_k) \right\|^2 \tag{53}$$

Let $\Gamma_{g,k}^* := \min_{n \in S_{g,k}} \Gamma_k^{(n)}$ to relax the inequality, and due to clients' model in one group having the same structure:

$$\leq \frac{L^2}{\mathcal{T}} \sum_{\tau=1}^{\mathcal{T}} \sum_{g=1}^{\mathcal{G}} \frac{|C_g|}{\Gamma_{g,k}^*} \sum_{n \in S_{g,k}} \mathbb{E} \left\| \nabla F_g^{(n)}(\mathbf{w}_{g,k,\tau-1}) - \nabla F_g^{(n)}(\mathbf{W}_k) \right\|^2$$

$$= \frac{L^2}{\mathcal{T}} \sum_{\tau=1}^{\mathcal{T}} \sum_{g=1}^{\mathcal{G}} \frac{|C_g|}{\Gamma_{g,k}^*} \mathbb{E} \left\| (\nabla F_g(\mathbf{w}_{g,k,\tau-1}) - \nabla F_g(\mathbf{W}_k)) \odot m_{g,k} \right\|^2 \tag{54}$$

Step 4: Apply Lipschitz Smoothness. By the $L$-smoothness of each $F_i(\cdot)$, we have:

$$\leq \frac{L^2}{\mathcal{T}} \sum_{\tau=1}^{\mathcal{T}} \sum_{g=1}^{\mathcal{G}} \frac{|C_g|}{\Gamma_{g,k}^*} \mathbb{E} \left\| (\mathbf{w}_{g,k,\tau-1} - \mathbf{w}_k) \odot m_{g,k} \right\|^2 \tag{55}$$

Step 5: Final bound with group-wise deviation. Thus,

$$\sum_{n=1}^{N} \mathbb{E} \left\| \frac{1}{\mathcal{T}} \sum_{\tau=1}^{\mathcal{T}} \frac{1}{\Gamma_k^{(n)}} \sum_{n \in C_k^{(n)}} \left[ \nabla F_i^{(n)}(\mathbf{w}_{i,k,\tau-1}) - \nabla F_i^{(n)}(\mathbf{W}_k) \right] \right\|^2$$

$$\leq \frac{L^2}{\mathcal{T}} \sum_{\tau=1}^{\mathcal{T}} \sum_{g=1}^{\mathcal{G}} \frac{|C_g|}{\Gamma_{g,k}^*} \mathbb{E} \left\| (\mathbf{w}_{g,k,\tau-1} - \mathbf{w}_k) \odot m_{g,k} \right\|^2$$

$$= L^2 \sum_{g=1}^{\mathcal{G}} \frac{|C_g|}{\Gamma_{g,k}^*} \frac{1}{\mathcal{T}} \sum_{\tau=1}^{\mathcal{T}} \mathbb{E} \left\| (\mathbf{w}_{g,k,\tau-1} - \mathbf{w}_k) \odot m_{g,k} \right\|^2$$

$$\leq \frac{L^2 \gamma^2 \mathcal{T}^2}{3} \sum_{g=1}^{\mathcal{G}} \frac{|C_g|}{\Gamma_{g,k}^*} f_1^2(p_g) G^2 \tag{56}$$

**Lemma 4.** *Group-aware Gradient variance bound under Parameter-wise Aggregation.*

Under Assumption 3 (bounded stochastic gradient variance), for any round $k$, the deviation between the aggregated stochastic gradient and the full-batch gradient satisfies:

$$\sum_{n=1}^{N} \mathbb{E} \left\| \frac{1}{\mathcal{T}} \sum_{\tau=1}^{\mathcal{T}} \frac{1}{\Gamma_k^{(n)}} \sum_{n \in C_k^{(n)}} [\nabla F_i(\mathbf{w}_{k,i,\tau}, \xi_t) - \nabla F_i(\mathbf{W}_k)] \right\|^2 \leq \sum_{g=1}^{\mathcal{G}} \frac{|C_g|}{|\Gamma_{g,k}^*|^2} \cdot \frac{f_2^2(p_g) \cdot \sigma^2}{\mathcal{T}} \quad (57)$$

**Proof.**

We now define the stochastic gradient variance for client $n$ at time step $t$ as $\nabla F_i(\mathbf{w}_{i,k,\tau-1}, \xi_{i,\tau-1}) - \nabla F_i(\mathbf{w}_{i,k,\tau-1})$. We assume the following standard properties under Assumption 4:

- The stochastic gradient is unbiased with respect to the local full-batch gradient:
  $$\mathbb{E}_\xi \left[ [\nabla F_i(\mathbf{w}_{i,k,\tau-1}, \xi_{i,\tau-1}) - \nabla F_i(\mathbf{w}_{i,k,\tau-1})] \odot m^{(q,n)} \right] = 0.$$

- Clients sample independently from their local datasets, and thus the gradient noise terms $\delta_{i,\tau-1}$ are statistically independent across $n$.

- Each noise term has bounded second moment scaled by the structural sparsity:
  $$\mathbb{E} \| [\nabla F_i(\mathbf{w}_{i,k,\tau-1}, \xi_{i,\tau-1}) - \nabla F_i(\mathbf{w}_{i,k,\tau-1})] \odot m^{(q,n)} \|^2 \leq f_2^2(p_i)\sigma^2,$$
  where $f_2^2(p_i)$ encapsulats the influence of $m^{(q,n)}$ for variance .

These assumptions allow us to compute the variance of aggregated group gradients by summing client-local variances.

Then, Define the group-level gradient variance:

$$[\nabla F_g(\mathbf{w}_{g,k,\tau-1}, \xi_{g,\tau-1}) - \nabla F(\mathbf{w}_{g,k,\tau-1})] \odot m_{g,k} \quad (58)$$

$$:= \frac{1}{|C_g|} \sum_{n \in C_g} [\nabla F_i(\mathbf{w}_{i,k,\tau-1}, \xi_{i,\tau-1}) - \nabla F(\mathbf{w}_{i,k,\tau-1})] \odot m_{g,k} \quad (59)$$

We have:

$$\mathbb{E} \left[ [\nabla F_g(\mathbf{w}_{g,k,\tau-1}, \xi_{g,\tau-1}) - \nabla F_g(\mathbf{w}_{g,k,\tau-1})] \odot m_{g,k} \right]$$

$$= \mathbb{E} \left[ \frac{1}{|C_g|} \sum_{n \in C_g} [[\nabla F_i(\mathbf{w}_{i,k,\tau-1}, \xi_{i,\tau-1}) - \nabla F_i(\mathbf{w}_{i,k,\tau-1})] \odot m_{g,k}] \right]$$

$$= \frac{1}{|C_g|} \sum_{n \in C_g} \mathbb{E} \left[ [\nabla F_i(\mathbf{w}_{i,k,\tau-1}, \xi_{i,\tau-1}) - \nabla F_i(\mathbf{w}_{i,k,\tau-1})] \odot m_{g,k} \right]$$

$$= 0 \quad (60)$$

Then using the fact that $\mathbb{E} \left\| \sum_{i=1}^{n} x_i \right\|^2 = \sum_{i=1}^{n} \mathbb{E} \|x_i\|^2$ if $x_i$ are indenpendent and $\mathbb{E}[x_i] = 0$.

$$\mathbb{E} \left\| [\nabla F_g(\mathbf{w}_{g,k,\tau-1}, \xi_{g,\tau-1}) - \nabla F_g(\mathbf{w}_{g,k,\tau-1})] \odot m_{g,k} \right\|^2$$

$$= \mathbb{E} \left\| \frac{1}{|C_g|} \sum_{n \in C_g} [\nabla F_i(\mathbf{w}_{i,k,\tau-1}, \xi_{i,\tau-1}) - \nabla F_i(\mathbf{w}_{i,k,\tau-1})] \odot m_{g,k} \right\|^2$$

$$= \frac{1}{|C_g|^2} \mathbb{E} \left\| \sum_{n \in C_g} [\nabla F_i(\mathbf{w}_{i,k,\tau-1}, \xi_{i,\tau-1}) - \nabla F_i(\mathbf{w}_{i,k,\tau-1})] \odot m_{g,k} \right\|^2$$

$$= \frac{1}{|C_g|^2} \sum_{n \in C_g} \mathbb{E} \| [\nabla F_i(\mathbf{w}_{i,k,\tau-1}, \xi_{i,\tau-1}) - \nabla F_i(\mathbf{w}_{i,k,\tau-1})] \odot m_{g,k} \|^2$$

$$\leq \frac{|C_g| f_2^2(p_g)\sigma^2}{|C_g|^2}$$

$$= \frac{f_2^2(p_g)\sigma^2}{|C_g|}. \quad (61)$$

We begin with the definition of gradient variance:

$$\sum_{n=1}^{N} \mathbb{E} \left\| \frac{1}{\mathcal{T}} \sum_{\tau=1}^{\mathcal{T}} \frac{1}{\Gamma_k^{(n)}} \sum_{n \in C_k^{(n)}} \left[ \nabla F_i(\mathbf{w}_{k,i,\tau}, \xi_t) - \nabla F_i(\mathbf{W}_k) \right] \right\|^2$$

$$= \sum_{n=1}^{N} \mathbb{E} \left\| \frac{1}{\mathcal{T}} \sum_{\tau=1}^{\mathcal{T}} \frac{1}{\Gamma_k^{(n)}} \sum_{g \in \mathcal{G}_k^{(n)}} |C_g| \left[ \nabla F_g^{(n)}(\mathbf{w}_{g,k,\tau-1}, \xi_{g,\tau-1}) - \nabla F_g^{(n)}(\mathbf{w}_{g,k,\tau-1}) \right] \right\|^2 \quad (62)$$

Since all the sample $\xi_{i,\tau-1}$ are independent from each other for different $n$ and $\tau-1$, the difference between gradient and stochasic gradient $\nabla F_i^{(n)}(\mathbf{w}_{i,k,\tau-1}, \xi_{i,\tau-1}) - \nabla F_i^{(n)}(\mathbf{w}_{i,k,\tau-1})$ are independent gradient noise. Also, based on Eq. 60, the difference between group gradient and group stochasic gradient $\mathbb{E}[\nabla F_g(\mathbf{w}_{g,k,\tau-1}, \xi_{g,\tau-1}) - \nabla F_g(\mathbf{w}_{g,k,\tau-1})]$ are independent gradient noise has zero mean. Using the fact of $\mathbb{E} \left\| \sum_{i=1}^{n} x_i \right\|^2 = \sum_{i=1}^{n} \mathbb{E} \|x_i\|^2$:

$$\leq \sum_{n=1}^{N} \frac{1}{\mathcal{T}^2} \sum_{n=1}^{N} \sum_{g \in \mathcal{G}_k^{(n)}} \frac{|C_g|^2}{|\Gamma_k^{(n)}|^2} \mathbb{E} \left\| \nabla F_g^{(n)}(\mathbf{w}_{g,k,\tau-1}, \xi_{g,\tau-1}) - \nabla F_g^{(n)}(\mathbf{w}_{g,k,\tau-1}) \right\|^2 \quad (63)$$

Swap the sums of $g$ and $i$:

$$= \frac{1}{\mathcal{T}^2} \sum_{\tau=1}^{\mathcal{T}} \sum_{g=1}^{\mathcal{G}} \sum_{n \in S_{g,k}} \frac{|C_g|^2}{|\Gamma_k^{(n)}|^2} \mathbb{E} \left\| \nabla F_g^{(n)}(\mathbf{w}_{g,k,\tau-1}, \xi_{g,\tau-1}) - \nabla F_g^{(n)}(\mathbf{w}_{g,k,\tau-1}) \right\|^2 \quad (64)$$

Using $\Gamma_{g,k}^*$ to relax the inequality, and due to clients' model in one group having the same structure:

$$\leq \frac{1}{\mathcal{T}^2} \sum_{\tau=1}^{\mathcal{T}} \sum_{g=1}^{\mathcal{G}} \frac{|C_g|^2}{|\Gamma_{g,k}^*|^2} \sum_{n \in S_{g,k}} \mathbb{E} \left\| \nabla F_g^{(n)}(\mathbf{w}_{g,k,\tau-1}, \xi_{g,\tau-1}) - \nabla F_g^{(n)}(\mathbf{w}_{g,k,\tau-1}) \right\|^2$$

$$= \frac{1}{\mathcal{T}^2} \sum_{\tau=1}^{\mathcal{T}} \sum_{g=1}^{\mathcal{G}} \frac{|C_g|^2}{|\Gamma_{g,k}^*|^2} \mathbb{E} \left\| [\nabla F_g(\mathbf{w}_{g,k,\tau-1}, \xi_{g,\tau-1}) - \nabla F_g(\mathbf{w}_{g,k,\tau-1})] \odot m_{g,k} \right\|^2 \quad (65)$$

Therefore:

$$\mathbb{E} \left\| \frac{1}{\mathcal{T}} \sum_{\tau=1}^{\mathcal{T}} \nabla F(\mathbf{w}_{k,\tau}, \xi_t) - \nabla F(\mathbf{W}_k) \right\|^2$$

$$\leq \frac{1}{\mathcal{T}^2} \sum_{\tau=1}^{\mathcal{T}} \sum_{g=1}^{\mathcal{G}} \frac{|C_g|^2}{|\Gamma_{g,k}^*|^2} \mathbb{E} \left\| [\nabla F_g(\mathbf{w}_{g,k,\tau-1}, \xi_{g,\tau-1}) - \nabla F_g(\mathbf{w}_{g,k,\tau-1})] \odot m_{g,k} \right\|^2$$

$$\leq \frac{1}{\mathcal{T}^2} \sum_{\tau=1}^{\mathcal{T}} \sum_{g=1}^{\mathcal{G}} \frac{|C_g|^2}{|\Gamma_{g,k}^*|^2} \frac{f_2^2(p_g)\sigma^2}{|C_g|}.$$

$$\leq \sum_{g=1}^{\mathcal{G}} \frac{|C_g|}{|\Gamma_{g,k}^*|^2} \frac{f_2^2(p_g)\sigma^2}{\mathcal{T}} \quad (66)$$

**Lemma 5.** *Group-aware Gradient bias bound under Parameter-wise Aggregation.*

We first define the average gradient of the global model as

$$\nabla F^{(n)}(\mathbf{w}) := \frac{1}{\Gamma_k^{(n)}} \sum_{n \in C_k^{(n)}} \nabla F_i^{(n)}(\mathbf{w}) := \frac{1}{\Gamma_k^{(n)}} \sum_{g \in \mathcal{G}_k^{(n)}} |C_g| \nabla F_g^{(n)}(\mathbf{w}),$$

$$\text{where } \bar{\nabla} F_g := \frac{1}{|C_g|} \sum_{n \in \mathcal{C}_g} \nabla F_i^{(n)}(\mathbf{w}) \odot m_{g,k}$$

Then, the deviation between the structure-weighted group average gradient and the global gradient is bounded as:

$$\sum_{n=1}^{N} \mathbb{E} \left\| \frac{1}{\mathcal{T}} \sum_{\tau=1}^{\mathcal{T}} \frac{1}{\Gamma_k^{(n)}} \sum_{n \in C_k^{(n)}} \left[ \nabla F_i^{(n)}(\mathbf{w}_k) - \nabla F^{(n)}(\mathbf{W}_k) \right] \right\|^2 \leq \sum_{g=1}^{\mathcal{G}} \frac{|C_g|}{(\Gamma_{g,k}^*)} f_3^2(q_g) \zeta_g^2, \quad (67)$$

where $\zeta^2 := \|\nabla F_i(\mathbf{w}) - \nabla F(\mathbf{w})\|^2$ measures the statistical heterogeneity between cleint $n$ and the global gradient,.

**Proof.**

We first define the group gradient bias as :

$$[\nabla F_g(\mathbf{W}_k) - \nabla F(\mathbf{w}_k)] \odot m_{g,k} := \frac{1}{|C_g|} \sum_{n \in C_g} [\nabla F_g(\mathbf{W}_k) - \nabla F(\mathbf{w}_k)] \odot m_{g,k} \quad (68)$$

Then:

$$\mathbf{E} \left[ [\nabla F_g(\mathbf{W}_k) - \nabla F(\mathbf{w}_k)] \odot m_{g,k} \right]$$

$$= \mathbf{E} \left[ \frac{1}{|C_g|} \sum_{n \in C_g} [\nabla F_g(\mathbf{W}_k) - \nabla F(\mathbf{w}_k)] \odot m_{g,k} \right]$$

$$\leq \frac{1}{|C_g|} \sum_{n \in C_g} \mathbf{E} \left[ [\nabla F_g(\mathbf{W}_k) - \nabla F(\mathbf{w}_k)] \odot m_{g,k} \right]$$

$$\leq f_3^2(q_g) \zeta_g^2 \quad (69)$$

We start from the following group-aware gradient:

$$\sum_{n=1}^{N} \mathbb{E} \left\| \frac{1}{\mathcal{T}} \sum_{\tau=1}^{\mathcal{T}} \frac{1}{\Gamma_k^{(n)}} \sum_{n \in C_k^{(n)}} \left[ \nabla F_i^{(n)}(\mathbf{w}_k) - \nabla F^{(n)}(\mathbf{W}_k) \right] \right\|^2$$

$$= \sum_{n=1}^{N} \mathbb{E} \left\| \frac{1}{\mathcal{T}} \sum_{\tau=1}^{\mathcal{T}} \frac{1}{\Gamma_k^{(n)}} \sum_{g \in \mathcal{G}_k^{(n)}} |C_g| \left[ \nabla F_g^{(n)}(\mathbf{w}_k) - \nabla F^{(n)}(\mathbf{W}_k) \right] \right\|^2$$

$$\leq \frac{1}{\mathcal{T}} \sum_{\tau=1}^{\mathcal{T}} \sum_{i=1}^{K} \frac{1}{\Gamma_k^{(n)}} \sum_{g \in \mathcal{G}_k^{(n)}} |C_g| \cdot \mathbb{E} \left\| \nabla F_g^{(n)}(\mathbf{w}_k) - \nabla F^{(n)}(\mathbf{W}_k) \right\|^2$$

$$= \frac{1}{\mathcal{T}} \sum_{\tau=1}^{\mathcal{T}} \sum_{g=1}^{\mathcal{G}} \sum_{n \in S_{g,k}} \frac{|C_g|}{\Gamma_k^{(n)}} \cdot \mathbb{E} \left\| \nabla F_g^{(n)}(\mathbf{w}_k) - \nabla F^{(n)}(\mathbf{W}_k) \right\|^2$$

$$\leq \frac{1}{\mathcal{T}} \sum_{\tau=1}^{\mathcal{T}} \sum_{g=1}^{\mathcal{G}} \frac{|C_g|}{\Gamma_{g,k}^*} \sum_{n \in S_{g,k}} \mathbb{E} \left\| \nabla F_g^{(n)}(\mathbf{w}_k) - \nabla F^{(n)}(\mathbf{W}_k) \right\|^2$$

$$= \frac{1}{\mathcal{T}} \sum_{\tau=1}^{\mathcal{T}} \sum_{g=1}^{\mathcal{G}} \frac{|C_g|}{\Gamma_{g,k}^*} \mathbb{E} \left\| [\nabla F_g(\mathbf{w}_k) - \nabla F(\mathbf{W}_k)] \odot m_{g,k} \right\|^2$$

$$\leq \sum_{g=1}^{\mathcal{G}} \frac{|C_g|}{(\Gamma_{g,k}^*)} f_3^2(q_g) \zeta_g^2 \quad (70)$$

### G.2.3 SUBSTITUTE LAMMA INTO THE CONVERGENCE MAIN TERM

The third term on the right hand side of Eq. 41 can be changed to $\nabla F(\mathbf{W}_k)$ related. If the data are IID, then for all $n \in \mathcal{N}_k^{(n)}$, we have $\nabla F_i(\mathbf{W}_k) = \nabla F(\mathbf{W}_k)$. So the third term in Eq. (33) simplifies as:

$$\frac{3L\mathcal{T}^2\gamma^2}{2} \sum_{n=1}^{N} \mathbb{E} \left\| \frac{1}{\Gamma_k^{(n)}} \sum_{i \in C_k^{(n)}} \nabla F_i^{(n)}(\mathbf{W}_k) \right\|^2 = \frac{3L\mathcal{T}^2\gamma^2}{2} \mathbb{E} \|\nabla F(\mathbf{W}_k)\|^2 \tag{71}$$

If the data are non-IID, by inserting and subtracting the reference point $\nabla F^{(n)}(\mathbf{W}_k)$ and using bias decomposition:

$$\frac{3L\mathcal{T}^2\gamma^2}{2} \sum_{n=1}^{N} \mathbb{E} \left\| \frac{1}{\Gamma_k^{(n)}} \sum_{i \in C_k^{(n)}} \nabla F_i^{(n)}(\mathbf{W}_k) \right\|^2 .$$

$$= \frac{3L\mathcal{T}^2\gamma^2}{2} \sum_{n=1}^{N} \mathbb{E} \left\| \nabla F^{(n)}(\mathbf{W}_k) + \sum_{i \in C_k^{(n)}} \frac{1}{\Gamma_k^{(n)}} \left( \nabla F_i^{(n)}(\mathbf{W}_k) - \nabla F^{(n)}(\mathbf{W}_k) \right) \right\|^2$$

$$\leq 3L\mathcal{T}^2\gamma^2 \sum_{n=1}^{N} \mathbb{E} \left\| \nabla F^{(n)}(\mathbf{W}_k) \right\|^2 + 3L\mathcal{T}^2\gamma^2 \sum_{n=1}^{N} \mathbb{E} \left\| \sum_{i \in C_k^{(n)}} \frac{1}{\Gamma_k^{(n)}} \left( \nabla F_i^{(n)}(\mathbf{W}_k) - \nabla F^{(n)}(\mathbf{W}_k) \right) \right\|^2$$

$$\leq 3L\mathcal{T}^2\gamma^2 \cdot \mathbb{E} \|\nabla F(\mathbf{W}_k)\|^2 + 3L\mathcal{T}^2\gamma^2 \cdot \sum_{g=1}^{G} \frac{|C_g|}{\Gamma_{g,k}^{*2}} f_3^2(p_g)\zeta^2. \tag{72}$$

Therefore, under IID data, combining Eq. 71 and Eq. 42 with Lemmas 2–4, we obtain:

$$\mathbb{E}[F(\mathbf{w}_{k+1})] - \mathbb{E}[F(\mathbf{W}_k)]$$

$$\leq \underbrace{-\frac{\gamma\mathcal{T}}{2} \cdot \mathbb{E}\|\nabla F(\mathbf{W}_k)\|^2}_{\text{descent term}} + \underbrace{\frac{L^2\gamma^3\mathcal{T}^3}{6} \sum_{g=1}^{\mathcal{G}} \frac{|C_g|}{\Gamma_{g,k}^*} f_1^2(p_g)G^2}_{\text{inner product drift}} + \underbrace{\frac{3L\mathcal{T}\gamma^2}{2} \sum_{g=1}^{\mathcal{G}} \frac{|C_g|}{\Gamma_{g,k}^{*2}} f_2^2(p_g)\sigma^2}_{\text{variance term}}$$

$$+ \underbrace{\frac{L^3\mathcal{T}^4\gamma^4}{2} \sum_{g=1}^{\mathcal{G}} \frac{|C_g|}{\Gamma_{g,k}^*} f_1^2(p_g)G^2}_{\text{model drift}} + \underbrace{\frac{3L\mathcal{T}^2\gamma^2}{2} \cdot \mathbb{E}\|\nabla F(\mathbf{W}_k)\|^2}_{\text{gradient noise term}} .$$

$$= \left( \frac{\gamma\mathcal{T}(3L\mathcal{T}\gamma - 1)}{2} \right) \cdot \mathbb{E}\|\nabla F(\mathbf{W}_k)\|^2 + \frac{L^2\gamma^3\mathcal{T}^3(3L\mathcal{T}\gamma + 1)}{6} \cdot \sum_{g=1}^{\mathcal{G}} \frac{|C_g|}{\Gamma_{g,k}^*} f_1^2(p_g)G^2$$

$$+ \frac{3L\mathcal{T}\gamma^2}{2} \sum_{g=1}^{\mathcal{G}} \frac{|C_g|}{\Gamma_{g,k}^{*2}} f_2^2(p_g)\sigma^2. \tag{73}$$

If the data are non-IID, combining Eq. 72 and Eq. 42 with Lemmas 2–5, we obtain:

$$\mathbb{E}[F(\mathbf{w}_{k+1})] - \mathbb{E}[F(\mathbf{W}_k)]$$

$$\leq \underbrace{-\frac{\gamma\mathcal{T}}{2} \cdot \mathbb{E}\|\nabla F(\mathbf{W}_k)\|^2}_{\text{descent term}} + \underbrace{\frac{L^2\gamma^3\mathcal{T}^3}{6} \sum_{g=1}^{\mathcal{G}} \frac{|C_g|}{\Gamma_{g,k}^*} f_1^2(p_g)G^2}_{\text{inner product drift}} + \underbrace{\frac{3L\mathcal{T}\gamma^2}{2} \sum_{g=1}^{\mathcal{G}} \frac{|C_g|}{\Gamma_{g,k}^{*2}} f_2^2(p_g)\sigma^2}_{\text{variance}}$$

$$+ \frac{L^3 \mathcal{T}^4 \gamma^4}{2} \underbrace{\sum_{g=1}^{\mathcal{G}} \frac{|C_g|}{\Gamma_{g,k}^*} f_1^2(p_g) G^2}_{\text{model drift}} + \underbrace{3L\mathcal{T}^2 \gamma^2 \mathbb{E}\|\nabla F(\mathbf{W}_k)\|^2}_{\text{gradient noise}} + \underbrace{3L\mathcal{T}^2 \gamma^2 \sum_{g=1}^{\mathcal{G}} \frac{|C_g|}{\Gamma_{g,k}^{*2}} f_3^2(p_g) \zeta^2}_{\text{bias term}} \cdot$$

$$= \left( \frac{\gamma \mathcal{T}(6L\mathcal{T}\gamma - 1)}{2} \right) \cdot \mathbb{E}\|\nabla F(\mathbf{W}_k)\|^2 + \frac{L^2 \gamma^3 \mathcal{T}^3 (3L\mathcal{T}\gamma + 1)}{6} \cdot \sum_{g=1}^{\mathcal{G}} \frac{|C_g|}{\Gamma_{g,k}^*} f_1^2(p_g) G^2$$

$$+ \frac{3L\mathcal{T}\gamma^2}{2} \sum_{g=1}^{\mathcal{G}} \frac{|C_g|}{\Gamma_{g,k}^{*2}} f_2^2(p_g) \sigma^2 + 3L\mathcal{T}^2 \gamma^2 \sum_{g=1}^{\mathcal{G}} \frac{|C_g|}{\Gamma_{g,k}^{*2}} f_3^2(p_g) \zeta^2. \tag{74}$$

### G.2.4 FINAL BOUND BASED ON MASK-AWARE AGGREGATION

**Telescoping sum.** We now apply the upper bound for $\mathbb{E}\langle \nabla F(\mathbf{W}_k), \mathbf{W}_{k+1} - \mathbf{W}_k \rangle$ and the bound for $\frac{L}{2}\mathbb{E}\|\mathbf{W}_{k+1} - \mathbf{W}_k\|^2$, and plug them into the smoothness inequality. Summing both sides of the inequality over $k = 1, \ldots, K$ yields:

$$\mathbb{E}[F(\mathbf{W}_{K+1})] - \mathbb{E}[F(\mathbf{W}_1)] = \sum_{k=1}^{K} \left( \mathbb{E}[F(\mathbf{W}_{k+1})] - \mathbb{E}[F(\mathbf{W}_k)] \right) \tag{75}$$

Based on Eq. 73, substituting the detailed expressions of each term under the IID setting, we obtain:

$$\mathbb{E}[F(\mathbf{W}_{k+1})] - \mathbb{E}[F(\mathbf{W}_1)]$$

$$\leq \left( \frac{\gamma \mathcal{T}(3L\mathcal{T}\gamma - 1)}{2} \right) \sum_{k=1}^{K} \mathbb{E}\|\nabla F(\mathbf{W}_k)\|^2 + \sum_{k=1}^{K} \left[ \frac{L^2 \gamma^3 \mathcal{T}^3 (3L\mathcal{T}\gamma + 1)}{6} \sum_{g=1}^{\mathcal{G}} \frac{|C_g|}{\Gamma_{g,k}^*} f_1^2(p_{g,k}) G^2 \right]$$

$$+ \sum_{k=1}^{K} \left[ \frac{3L\mathcal{T}\gamma^2}{2} \sum_{g=1}^{\mathcal{G}} \frac{|C_g|}{\Gamma_{g,k}^{*2}} f_2^2(p_{g,k}) \sigma^2 \right]. \tag{76}$$

We choose learning rate $\gamma \leq \frac{1}{6L\mathcal{T}}$ (which satisfies $\gamma \leq \frac{1}{3L\mathcal{T}}$) to ensure descent. With this choice, the descent coefficient becomes negative, and we can move the descent term to the left-hand side:

$$\frac{\mathcal{T}\gamma}{4} \sum_{k=1}^{K} \mathbb{E}\|\nabla F(\mathbf{W}_k)\|^2 \leq \mathbb{E}[F(\mathbf{W}_1)] - \mathbb{E}[F(\mathbf{W}_{k+1})]$$

$$+ \sum_{k=1}^{K} \left[ \frac{L^2 \gamma^3 \mathcal{T}^3 (3L\mathcal{T}\gamma + 1)}{6} \sum_{g=1}^{\mathcal{G}} \frac{|C_g|}{\Gamma_{g,k}^*} f_1^2(p_{g,k}) G^2 \right]$$

$$+ \sum_{k=1}^{K} \left[ \frac{3L\mathcal{T}\gamma^2}{2} \sum_{g=1}^{\mathcal{G}} \frac{|C_g|}{\Gamma_{g,k}^{*2}} f_2^2(p_{g,k}) \sigma^2 \right]. \tag{77}$$

Using the fact that $\mathbb{E}[F(\mathbf{W}_{k+1})]$ is non-negative and dividing both sides above by $\frac{\mathcal{T}\gamma K}{4}$, we have:

$$\frac{1}{K} \sum_{k=1}^{K} \mathbb{E}\|\nabla F(\mathbf{W}_k)\|^2 \leq \frac{4}{\mathcal{T}\gamma K} \mathbb{E}[F(\mathbf{W}_1)]$$

$$+ \frac{3L^2 \gamma^2 \mathcal{T}^2 (3L\mathcal{T}\gamma + 1)}{2K} \sum_{k=1}^{K} \sum_{g=1}^{\mathcal{G}} \frac{|C_g|}{\Gamma_{g,k}^*} f_1^2(p_{g,k}) G^2$$

$$+ \frac{6L\gamma}{K} \sum_{k=1}^{K} \sum_{g=1}^{\mathcal{G}} \frac{|C_g|}{\Gamma_{g,k}^{*2}} f_2^2(p_{g,k}) \sigma^2 \tag{78}$$

For non-IID data distribution, based on Eq. 74, similar to the process of the IID setting, by choosing learning rate $\gamma \leq \frac{1}{12L\mathcal{T}}$ (which satisfies $\gamma \leq \frac{1}{6L\mathcal{T}}$) to ensure descent and using the fact that $\mathbb{E}[F(\mathbf{W}_{K+1})]$ is non-negative, we have:

$$\frac{1}{K} \sum_{k=1}^{K} \mathbb{E}\|\nabla F(\mathbf{W}_k)\|^2 \leq \frac{4}{\mathcal{T}\gamma K} \mathbb{E}[F(\mathbf{W}_1)]$$

$$+ \frac{3L^2\gamma^2\mathcal{T}^2(3L\mathcal{T}\gamma + 1)}{2K} \sum_{k=1}^{K} \sum_{g=1}^{\mathcal{G}} \frac{|C_g|}{\Gamma_{g,k}^*} f_1^2(p_{g,k}) G^2$$

$$+ \frac{6L\gamma}{K} \sum_{k=1}^{K} \sum_{g=1}^{\mathcal{G}} \frac{|C_g|}{\Gamma_{g,k}^{*2}} f_2^2(p_{g,k})\sigma^2$$

$$+ \frac{12L\mathcal{T}\gamma}{K} \sum_{k=1}^{K} \sum_{g=1}^{\mathcal{G}} \frac{|C_g|}{\Gamma_{g,k}^{*2}} f_3^2(p_g)\zeta^2. \tag{79}$$

### G.3 CONVERGENCE ANALYSIS BASED ON GRADIENT AVERAGE AGGREGATION

Similar to the process of MA:

$$\mathbb{E}[F(\mathbf{W}_{k+1})] - \mathbb{E}[F(\mathbf{W}_k)] \leq \mathbb{E}\langle \nabla F(\mathbf{W}_k), \mathbf{w}_{k+1} - \mathbf{w}_k \rangle + \frac{L}{2}\mathbb{E}\|\mathbf{W}_{k+1} - \mathbf{W}_k\|^2.$$

But for gradient average(GA), we have:

$$\mathbf{W}_k - \mathbf{W}_{k+1} = \frac{1}{C} \sum_{i=1}^{C} \sum_{\tau=1}^{\mathcal{T}} \gamma \nabla F_i(\mathbf{w}_{i,k,\tau-1}, \xi_{i,\tau-1}) \odot m_{i,k}.$$

Then, first analyze the inner product, we have:

$$\mathbb{E}\langle \nabla F(\mathbf{W}_k), \mathbf{W}_{k+1} - \mathbf{W}_k \rangle$$

$$= \mathbb{E}\left\langle \nabla F(\mathbf{W}_k), -\frac{1}{C} \sum_{i=1}^{C} \sum_{\tau=1}^{\mathcal{T}} \gamma \nabla F_i(\mathbf{w}_{i,k,\tau-1}, \xi_{i,\tau-1}) \odot m_{i,k} \right\rangle$$

$$= \mathbb{E}\left\langle \nabla F(\mathbf{W}_k), -\frac{1}{C} \sum_{i=1}^{C} \sum_{\tau=1}^{\mathcal{T}} \gamma \mathbb{E}\left[\nabla F_i(\mathbf{w}_{i,k,\tau-1}, \xi_{i,\tau-1}) \mid \mathbf{w}_k\right] \odot m_{i,k} \right\rangle$$

$$= \mathbb{E}\left\langle \nabla F(\mathbf{W}_k), -\frac{1}{C} \sum_{i=1}^{C} \sum_{\tau=1}^{\mathcal{T}} \gamma \nabla F_i(\mathbf{w}_{i,k,\tau-1}) \odot m_{i,k} \right\rangle$$

$$= -\mathbb{E}\left\langle \nabla(\mathbf{W}_k), \frac{1}{C} \sum_{i=1}^{C} \sum_{\tau=1}^{\mathcal{T}} \gamma \left((\nabla F_i(\mathbf{w}_{i,k,\tau-1}) - \nabla F(\mathbf{W}_k) + \nabla F(\mathbf{W}_k)) \odot m_{i,k}\right) \right\rangle$$

$$= -\mathbb{E}\left\langle \nabla F(\mathbf{W}_k), \gamma\mathcal{T}\frac{1}{C} \sum_{i=1}^{C} \nabla F(\mathbf{W}_k) \odot m_{i,k} \right\rangle$$

$$\quad - \mathbb{E}\left\langle \nabla F(\mathbf{W}_k), \frac{1}{C} \sum_{i=1}^{C} \sum_{\tau=1}^{\mathcal{T}} \gamma \left(\nabla F_i(\mathbf{w}_{i,k,\tau-1}) - \nabla F(\mathbf{W}_k)\right) \odot m_{i,k} \right\rangle$$

$$\leq -\gamma\mathcal{T}\mathbb{E}\left\|\frac{1}{C} \sum_{i=1}^{C} \nabla F(\mathbf{W}_k) \odot m_{i,k}\right\|^2 + \frac{\gamma\mathcal{T}}{2}\mathbb{E}\left\|\frac{1}{C} \sum_{i=1}^{C} \nabla F(\mathbf{W}_k) \odot m_{i,k}\right\|^2$$

$$\quad + \frac{\gamma\mathcal{T}}{2}\mathbb{E}\left\|\frac{1}{C} \sum_{i=1}^{C} \sum_{\tau=1}^{\mathcal{T}} \left(\nabla F_i(\mathbf{w}_{i,k,\tau-1}) - \nabla F(\mathbf{W}_k)\right) \odot m_{i,k}\right\|^2$$

$$\leq -\frac{\gamma\mathcal{T}}{2}\mathbb{E}\left\|\frac{1}{C} \sum_{i=1}^{C} \nabla F(\mathbf{W}_k) \odot m_{i,k}\right\|^2$$

$$+ \frac{\gamma \mathcal{T}}{2} \mathbb{E} \left\| \frac{1}{C} \sum_{i=1}^{C} \sum_{\tau=1}^{\mathcal{T}} (\nabla F_i(\mathbf{w}_{i,k,\tau-1}) - \nabla F(\mathbf{W}_k)) \odot m_{i,k} \right\|^2. \tag{80}$$

Upperbound for $\frac{L}{2} \mathbb{E} \|\mathbf{w}_{k+1} - \mathbf{w}_k\|^2$

$$\frac{L}{2} \mathbb{E} \|\mathbf{w}_{k+1} - \mathbf{w}_k\|^2 = \frac{L}{2} \mathbb{E} \left\| \frac{1}{C} \frac{1}{\mathcal{T}} \sum_{i=1}^{C} \sum_{\tau=1}^{\mathcal{T}} \gamma \nabla F_i(\mathbf{w}_{i,k,\tau-1}, \xi_{i,\tau-1}) \right\|^2$$

$$\leq \frac{3L\mathcal{T}^2 \gamma^2}{2} \mathbb{E} \left\| \frac{1}{C} \frac{1}{\mathcal{T}} \sum_{i=1}^{C} \sum_{\tau=1}^{\mathcal{T}} [\nabla F_i(\mathbf{w}_{i,k,\tau-1}, \xi_{i,\tau-1}) - \nabla F_i(\mathbf{w}_{i,k,\tau-1})] \odot m_{i,k} \right\|^2$$

$$+ \frac{3L\mathcal{T}^2 \gamma^2}{2} \mathbb{E} \left\| \frac{1}{C} \frac{1}{\mathcal{T}} \sum_{i=1}^{C} \sum_{\tau=1}^{\mathcal{T}} [\nabla F_i(\mathbf{w}_{i,k,\tau-1}) - \nabla F_i(\mathbf{W}_k)] \odot m_{i,k} \right\|^2$$

$$+ \frac{3L\mathcal{T}^2 \gamma^2}{2} \mathbb{E} \left\| \frac{1}{C} \sum_{i=1}^{C} \nabla F_i(\mathbf{W}_k) \odot m_{i,k} \right\|^2. \tag{81}$$

Then, combining the two upperbounds, we have:

$$\mathbb{E}[F(\mathbf{w}_{k+1})] - \mathbb{E}[F(\mathbf{W}_k)] \leq \mathbb{E} \langle \nabla F(\mathbf{W}_k), \mathbf{w}_{k+1} - \mathbf{w}_k \rangle + \frac{L}{2} \mathbb{E} \|\mathbf{W}_{k+1} - \mathbf{W}_k\|^2$$

$$\leq - \frac{\gamma \mathcal{T}}{2} \mathbb{E} \| \frac{1}{C} \sum_{i=1}^{\mathcal{C}} \nabla F(W_k) \odot m_{i,k} \|^2$$

$$+ \frac{\gamma \mathcal{T}}{2} \mathbb{E} \left\| \frac{1}{\mathcal{T} C} \sum_{\tau=1}^{C} \sum_{\tau=1}^{\mathcal{T}} (\nabla F_i(\mathbf{w}_{i,k,\tau-1}) - \nabla F(\mathbf{W}_k)) \odot m_{i,k} \right\|^2$$

$$+ \frac{3L\mathcal{T}^2 \gamma^2}{2} \mathbb{E} \left\| \frac{1}{C} \frac{1}{\mathcal{T}} \sum_{n=1}^{C} \sum_{\tau=1}^{\mathcal{T}} [\nabla F_i(\mathbf{w}_{i,k,\tau-1}, \xi_{i,\tau-1}) - \nabla F_i(\mathbf{w}_{i,k,\tau-1})] \odot m_{i,k} \right\|^2$$

$$+ \frac{3L\mathcal{T}^2 \gamma^2}{2} \mathbb{E} \left\| \frac{1}{C} \frac{1}{\mathcal{T}} \sum_{n=1}^{C} \sum_{\tau=1}^{\mathcal{T}} [\nabla F_i(\mathbf{w}_{i,k,\tau-1}) - \nabla F_i(\mathbf{W}_k)] \odot m_{i,k} \right\|^2$$

$$+ \frac{3L\mathcal{T}^2 \gamma^2}{2} \mathbb{E} \left\| \frac{1}{C} \sum_{n=1}^{C} \nabla F_i(\mathbf{W}_k) \odot m_{i,k} \right\|^2. \tag{82}$$

Then, similar to the MA process, by choosing local learning rate $\gamma < \frac{1}{6L\mathcal{T}}$ for IID data and $\gamma < \frac{1}{12L\mathcal{T}}$ for Non-IID data have :

$$\frac{1}{K} \sum_{k=1}^{K} \mathbb{E} \| \frac{1}{C} \sum_{i=1}^{\mathcal{C}} \nabla F(W_k) \odot m_{i,k} \|^2 \leq \frac{4\mathbb{E}[F(W_1)]}{\mathcal{T}\gamma K} + \underbrace{\frac{3L^2 \gamma^2 \mathcal{T}^2 (3L\mathcal{T}\gamma + 1)}{2KC} \sum_{k=1}^{K} \sum_{i=1}^{C} f_1^2(p_{i,k}) G^2}_{\text{Model Drift Term}}$$

$$+ \underbrace{\frac{6L\mathcal{T}\gamma^2}{KC^2} \sum_{k=1}^{K} \sum_{i=1}^{C} f_2^2(p_{i,k}) \sigma^2}_{\text{Variance Term}} + \underbrace{\frac{12L\mathcal{T}^2 \gamma^2}{KC} \sum_{k=1}^{K} \sum_{i=1}^{C} f_3^2(p_{i,k}) \zeta^2}_{\text{Bias Term (Non-IID only)}}. \tag{83}$$

## G.4 COMPARISON AND FINAL ANALYSIS OF THE CONVERGENCE OF TWO AGGREGATION RULES

Based on Eq. 81, Eq. 82, Eq. 78 and Eq. 79, we complete the proof of **Theorem 1** (Sec 5.2).

**Interpretation.** In GA, the server aggregates sparse gradients as $\frac{1}{C} \sum_{i=1}^{C} \nabla F_i(W_k) \odot m_{i,k}$, where both the gradients and the associated error terms (model drift and variance) are reduced in magnitude because they are sparsified by the masks, with the scaling governed by $f(p_{i,k})$. By contrast, MA simulates global gradients by aggregating partial client gradients, where each retained coordinate is effectively *amplified* by a factor $\frac{|\mathcal{C}_g|}{\Gamma_{g,k}^*}$. This factor arises because MA normalizes only over covering clients, rather than over all $C$ clients as GA does. Since $\Gamma_{g,k}^* \leq C$, we obtain $\sum_{g=1}^{G} \frac{|\mathcal{C}_g|}{\Gamma_{g,k}^*} f^2(p_{g,k}) \geq \sum_{g=1}^{G} \frac{|\mathcal{C}_g|}{C} f^2(p_{g,k}) = \frac{1}{C} \sum_{i=1}^{C} f^2(p_{i,k})$. Thus, while amplification improves gradient completeness and accelerates early training, it also enlarges the sparse drift and variance terms, ultimately hindering convergence.

Based on Eq. 83 and Eq. 42, we get the update gradient of GA and MA. and combined with the effective optimization subspace, we complete the proof of **Theorem 2** (Sec 5.2).

**Interpretation.** Both GA and MA converge to a small neighborhood of a stationary point of standard FL. The distance is better captured by the average sub-model density across clients and time. $\frac{1}{K} \sum_{k=1}^{K} \sum_{g=1}^{G} \sum_{X \in \{\mathcal{A}_k^\dagger, \mathcal{B}_k^\dagger\}} X f^2(p_{g,k})$, and higher densities (e.g., under GMR) increase coverage, decrease these functionals, and tighten the neighborhood. Although GA and MA operate over the same model search space in theory, their aggregation strategies differ fundamentally. GA aggregates gradients on all rigions, effectively computing:

$$\nabla F(W_k) \approx \frac{1}{C} \sum_{i=1}^{C} \nabla F_i(W_k) \odot m_{i,k}, \tag{84}$$

where each mask $m_{i,k}$ restricts the gradient to the pruned sub-model. This leads to a *shrunk and overly averaged* update direction, which does not reflect the true full-model gradient.

By contrast, MA reconstructs the full gradient more faithfully through partial participation and per-parameter aggregation:

$$\nabla F(W_k^{(n)}) \approx \frac{1}{|\Gamma_k^{(n)}|} \sum_{i \in C_k^{(n)}} \nabla F_i^{(n)}(W_k), \tag{85}$$

which preserves more diverse update signals across the parameter space.

As a result, gradient FedAvg may converge more slowly or to inferior optima compared to MA, especially in heterogeneous settings where model overlaps are limited and sub-models vary significantly.

## G.5 CONNECTION WITH THE GMR

**Optimization subspace view.** Building on Eq. 84 and Eq. 85, we derive the client-specific gradient expressions. This allows us to formally relate the attainable solutions of full-model training and model-heterogeneous training to their respective effective search subspaces. Let

$$\mathcal{S}_{\text{mask}} = \mathcal{S}_{\text{gradient}} := \text{span}\Big\{ \nabla F_i(W_k) \odot m_{i,k} \ : \ i = 1, \dots, C \Big\}, \qquad \mathcal{S}_{\text{full}} := \mathbb{R}^d.$$

Thus both submodel-based strategies (GA and MA) update within the same restricted space, $\mathcal{S}_{\text{gradient}} = \mathcal{S}_{\text{mask}} \subseteq \mathbb{R}^d$, whereas full-model FL searches in $\mathbb{R}^d$. By standard convex-optimization arguments, optimizing over a strict subspace yields a weakly worse optimum than optimizing over the full space; hence

$$F(W_{\text{full}}^*) \leq F(W_{\text{gradient}}^*), \qquad F(W_{\text{full}}^*) \leq F(W_{\text{mask}}^*),$$

with strict inequality whenever the full-model minimizer does not lie in the affine set reachable by masked updates. Under GMR's nested masks, the span $\mathcal{S}_{\text{mask}}$ expands over rounds, enlarging the feasible search region. Therefore, when submodels plateau, restoring capacity increases $\mathcal{S}_{\text{mask}}$, enabling further descent and tighter proximity to the full-model optimum.

**Effect of Different Aggregation Methods on GMR.** When *model restoration* (density increase) occurs, GA/FA lack *per-coordinate scale normalization*, so update magnitudes jump up or down and can dilute or offset GMR's gains.

**FedAvg (FA).** (zeros for missing neurons) The $n$-th coordinate at round $k$ is

$$W_{k+1}^{(n)} = \frac{1}{C} \sum_{i=1}^{C} w_{i,k}^{(n)}, \qquad w_{i,k}^{(n)} = 0 \text{ if pruned.}$$

Before restoration, only $\Gamma_k^{(n)}$ clients are active on coordinate $n$, so averaging over zeros effectively *shrinks* the corresponding weight. Once restoration occurs, $\Gamma_k^{(n)}$ increases and the effective averaging scale jumps from $\Gamma_k^{(n)}/C$ to $\Gamma_{k+1}^{(n)}/C$, leading to step-size discontinuities and instability. More critically, some neurons may collapse to zero during early aggregation; once removed, they cannot be effectively retrained even after restoration.

Our ablation results corroborate this effect. On FEMNIST with Conv2D, FA remains trainable but shows degraded accuracy, since only a fraction of neurons consistently contribute to similar tasks. On CIFAR-10 with VGG11, FA collapses entirely: the loss falls rapidly to a trivial value within a few rounds and fails to recover, even when neurons are later restored. In contrast, on ImageNet100 with ResNet, the residual connections allow restored neurons to re-integrate, preventing collapse and enabling recovery. These results explain the divergent behavior of FA across datasets and architectures.

**Gradient-Average (GA).** (Sum only active gradients but still divide by $C$)

$$\Delta W_k^{(n)} \approx \frac{\gamma}{C} \sum_{i=1}^{C} g_{i,k}^{(n)} \mathbf{1}\{n \in m_{i,k}\} \quad \Rightarrow \quad \mathbb{E}\big|\Delta W_k^{(n)}\big| \propto \gamma \frac{\Gamma_k^{(n)}}{C}.$$

Compared to FA, GA only rescales the magnitude of gradients without altering neuron weight, which explains why GA achieves comparable or even superior convergence once models are restored, in contrast to MA. However, under GMR, restoration not only changes model capacity but also increases $\Gamma_k^{(n)}$, thereby *linearly amplifying* the per-coordinate update step. This amplification introduces higher variance and jitter, ultimately diminishing the benefit of larger models.

**Mask-aware (MA).** MA normalizes per coordinate by $\Gamma_k^{(n)}$ (average over active clients only), keeping the update scale *invariant* across restorations. It avoids FA's weight shrinkage and GA's step amplification, hence combines stably and effectively with GMR.

Importantly, once all clients are fully restored ($\rho_i = 1$ for all $i$), every coordinate is active on all clients ($\Gamma_k^{(n)} = C$). In this regime, FA, GA, and MA all reduce to the standard FedAvg update

$$W_{k+1}^{(n)} = \frac{1}{C} \sum_{i=1}^{C} w_{i,k}^{(n)},$$

and are therefore equivalent in both formulation and convergence behavior.

**GA/FA benefit less from restoration**. Though model restoration enlarges the representational capacity and enables better convergence of the global model, FA and GA benefit less from restoration because they lack proper normalization on weights or gradients. In contrast, MA explicitly normalizes per-coordinate updates, making it more compatible with GMR.

FA aggregates with zero-padding,

$$W_{k+1}^{(n)} = \frac{1}{C} \sum_{i=1}^{C} w_{i,k}^{(n)}.$$

When $\Gamma_k^{(n)} \ll C$, zeros dominate and weights shrink; as GMR raises density ($\Gamma_k^{(n)} \uparrow$), shrinkage abates but the effective averaging scale jumps from $\Gamma_k^{(n)}/C$ to $\Gamma_{k+1}^{(n)}/C$, inducing abrupt step-size changes.

GA updates

$$\Delta W_k^{(n)} = \frac{\gamma}{C} \sum_{i=1}^{C} g_{i,k}^{(n)} \mathbf{1}\{n \in m_{i,k}\},$$

so restoration linearly enlarges the per-coordinate step ($\propto \Gamma_k^{(n)}/C$), increasing variance. In contrast, MA normalizes by the active coverage and effectively averages over $\Gamma_k^{(n)}$ rather than $C$, keeping the update scale stable across restorations and preserving GMR's gains.

## H    EXPERIMENTS SETTINGS

The FL simulation consists of one server and ten clients, where heterogeneity is introduced by varying client bandwidth. The server's average upload speed is fixed at 20 MB/s, while the download bandwidth is assumed to be sufficiently large. The client bandwidth allocation is summarized in Table 4, where each entry denotes the pair of (download/upload) speeds in MB/s.

Table 4: Bandwidth allocation for clients. Each cell shows the number of clients assigned to the corresponding bandwidth setting. The preset client model density is determined according to the bandwidth for the comparison methods.

| Bandwidth (Download / Upload, MB/s) | 20/5 | 10/2.5 | 4/1 | 2/0.5 | 1/0.25 |
|---|---|---|---|---|---|
| Preset model density | 1.0 | 0.5 | 0.20 | 0.10 | 0.05 |
| Low | 2 | 2 | 2 | 2 | 2 |
| Medium | 1 | 1 | 2 | 3 | 3 |
| High | 1 | 1 | 1 | 1 | 6 |

To simulate bandwidth variability, we introduce additional randomness using a log-exponential distribution, which better reflects real-world conditions. Consequently, BCCs require more time to complete each round of FL, consistent with their slower communication capabilities.

Training setup. The training hyperparameters are summarized in Table 5. For FedGMR, we further fine-tune the patience parameter under different settings.

Model architectures. The architectural details of the employed models are provided in Table 6.

## I    FEASIBILITY STUDY: OPTIMAL MODEL DENSITY AT DIFFERENT TRAINING STAGES

In Section 6.1, we showed that the optimal model size grows with training. The detailed experimental results are provided here for completeness.

Table 7 and Fig 2 report the accuracy growth rates at different accuracy intervals across datasets. Consistent with our discussion, smaller densities (e.g., 0.3 or 0.4) yield faster improvements in the early phase, while larger densities (e.g., 0.9 or 1.0) dominate in later stages, where higher accuracy requires greater capacity.

## J    EFFECT OF IMS.

We evaluate the efficiency of IMS on the FEMNIST dataset under a synchronous setting for 1000 communication rounds, and the total training time (in seconds) was recorded. Based on the advantages of IMS become more pronounced as the number of clients increases. Without IMS, training time grows rapidly with more clients—for example, from 8934s (5 clients) to 24104s (30 clients) under high heterogeneity—due to repeated transmission of similar sub-models.

In contrast, with IMS, the increase in training time remains nearly linear and much slower—for example, from 8375s to 8883s under the same setting—highlighting IMS's effectiveness in reusing shared parameters and mitigating redundancy during server-to-client communication.

## K    OTHER DISCUSSION

**Client density change during Training**    We report the client training numbers and density evolution of FedGMR under IID and low heterogeneity, as shown in Figure 3, where the horizontal axis denotes the training time under a semi-asynchronous setting. The top row illustrates the cumulative local training steps of individual clients, while the bottom row shows the corresponding evolution of client model densities. The top row (a–c) shows the cumulative local training steps of individual clients on FEMNIST, CIFAR-10, and ImageNet-100. In conventional asynchronous FL, the fastest client may exceed 10,000 updates while the slowest completes only a few hundred, leaving BCCs

Table 5: Evaluation configurations. $k$ denotes the index of the fastest-client rounds in the semi-asynchronous setting.

|  | FEMNIST | CIFAR-10 | ImageNet-100 |
|---|---|---|---|
| Learning rate at round $k$ | 0.25 | 0.25 | $0.05 \cdot 0.5^{\frac{k}{10000}}$ |
| Mini-batch size / local iterations | 20 / 5 | 20 / 5 | 20 / 5 |
| Restoration interval $k_{\text{rest}}$ (rounds) | 25 | 50 | 50 |
| Patience in ablation experiments / interval | 5 / 25 | 5 / 50 | 5 / 50 |
| Total training time (sec) | 80,000 | 310,000 | 320,000 |

Table 6: Model architectures and parameter sizes.

| Architecture | Conv-2 | VGG-11 | ResNet-18 |
|---|---|---|---|
| Convolutional | 32, pool; 64, pool | 64, pool; 128, pool; $2 \times 256$, pool; $2 \times 512$, pool; $2 \times 512$, pool | 64, pool; $2\times[64, 64]$; $2\times[128, 128]$; $2 \times [256, 256]$; $2 \times [512, 512]$ |
| Fully-connected | 2048, 62 (input: 3136) | 512, 512, 10 (input: 512) | avgpool, 100 (input: 512) |
| Conv/FC/all params | 52.1K / 6.6M / 6.6M | 9.2M / 530.4K / 9.8M | 11.2M / 102.6K / 11.3M |

almost irrelevant. The bottom row (d–f) depicts the evolution of client densities under GMR. By progressively restoring capacity, even the weakest clients reach 2,000–3,000 updates and contribute meaningfully throughout training.

Dataset-specific patterns also emerge. On FEMNIST, restoration is slow and clients remain at low densities for long periods. On CIFAR-10, small models sustain training briefly, but restoration accelerates and clients quickly reach full models. On ImageNet-100, very small models are ineffective and restore almost immediately, yet once densities surpass 0.5, clients can train at that level for an extended period before further restoration is required.

Overall, the figure illustrates how GMR balances efficiency with capacity: clients train sub-models to saturation, then restore and continue, ensuring more sustained and equitable contributions across heterogeneous clients.

**Restore Sensitivity in GMR.** Table 9 summarizes the optimal patience values under different datasets, heterogeneity levels, and data distributions. In general, the optimal patience tends to increase as the task becomes more difficult. From the accuracy growth curves, however, we observe that larger patience leads to slower progress in the early training phase, as the restricted model capacity limits the efficiency of learning. Nevertheless, higher patience allows the global model to better incorporate knowledge from BCCs, which in turn results in better convergence at later stages. On the other hand, if the patience is set excessively high, the same capacity limitation may hinder further convergence, leading to diminishing returns.

Table 7: Optimal model density and corresponding accuracy interval where the highest accuracy growth rate is observed.

| FEMNIST | | CIFAR-10 | | ImageNet-100 | |
|---|---|---|---|---|---|
| Density | Interval | Density | Interval | Density | Interval |
| 0.3 | 9.35–75.38 | 0.1 | 12.75–35.98 | 0.6 | 2.66–10.18 |
| 0.4 | 75.55–83.35 | 0.4 | 36.84–61.21 | 0.8 | 10.69–18.85 |
| 0.6 | 83.54–84.33 | 0.6 | 61.64–72.80 | 0.9 | 19.30–38.33 |
| 0.8 | 84.35–84.58 | 0.8 | 73.04–83.49 | 0.8 | 38.65–41.71 |
| 1.0 | 84.65–85.20 | 1.0 | 83.55–84.51 | 1.0 | 42.01–63.89 |

| Method | High Heterogeneity | | | Low Heterogeneity | | |
|--------|-----------|------------|------------|-----------|------------|------------|
| | 5 Clients | 10 Clients | 30 Clients | 5 Clients | 10 Clients | 30 Clients |
| IMS | 8375 | 8651 | 8883 | 8551 | 8738 | 8978 |
| w/o IMS | 8934 | 11999 | 24104 | 10966 | 15999 | 36136 |

Table 8: Communication cost comparison (seconds) with and without IMS under different heterogeneity levels and client counts.

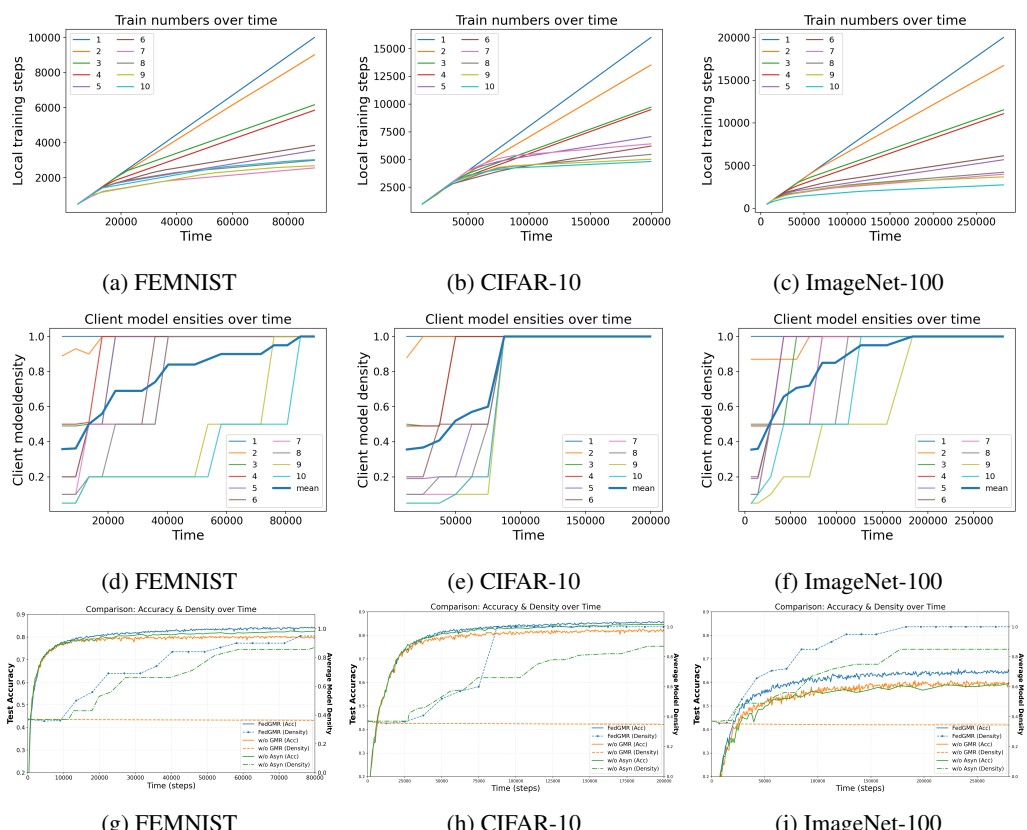

(a) FEMNIST  (b) CIFAR-10  (c) ImageNet-100

(d) FEMNIST  (e) CIFAR-10  (f) ImageNet-100

(g) FEMNIST  (h) CIFAR-10  (i) ImageNet-100

Figure 3: Training dynamics under heterogeneous bandwidth: top row shows training steps over time, bottom row shows accuracy over time, for FEMNIST, CIFAR-10, and ImageNet-100.

**GMR Makes the Global Model More Compact.**    Training with sub-models enables **GMR** to progressively reinforce well-trained subnetworks, yielding a more compact yet effective global model. To validate this, we directly pruned the final global model to the target density and compared its accuracy, as shown in Table 10. The results confirm that **GMR** preserves performance while producing compact models. This also explains our design choice: restoration follows the natural training trajectory of sub-models rather than arbitrary resizing, since repeatedly optimizing these nested structures strengthens their representations and improves adaptability of the restored model, which is consistent with the observations in Once-for-All training (Cai et al., 2020).

| Dataset | IID | | | Non-IID | | |
|---------|------|--------|-----|------|--------|-----|
| | high | medium | low | high | medium | low |
| FEMNIST | 10/25 | 1/25 | 1/25 | 5/25 | 5/25 | 2/25 |
| CIFAR10 | 2/50 | 1/50 | 1/50 | 2/50 | 2/50 | 2/50 |
| IMAGENET100 | 15/50 | 15/50 | 7/50 | 10/50 | 15/50 | 5/50 |

Table 9: Optimal patience values (first number) and evaluation intervals (second number) for different datasets under varying heterogeneity levels and data distributions.

Table 10: Models Pruning performance (%) trained by FedGMR and FedAsyn.

| Task | Model | 0.05 | 0.10 | 0.20 | 0.40 | 0.60 | 0.80 | 1.00 |
|------|-------|------|------|------|------|------|------|------|
| FEMNIST | FedGMR | 52.58 | 69.15 | 78.54 | 81.99 | 82.75 | 83.18 | 83.04 |
| | FedAsyn | 25.23 | 61.78 | 75.38 | 81.46 | 82.66 | 82.99 | 83.04 |
| CIFAR-10 | FedGMR | 15.52 | 48.80 | 80.24 | 83.54 | 83.84 | 83.93 | 84.04 |
| | FedAsyn | 10.00 | 10.00 | 21.06 | 81.15 | 83.65 | 83.97 | 84.00 |
| ImageNet-100 | FedGMR | 6.06 | 22.26 | 50.42 | 59.28 | 60.98 | 61.16 | 61.00 |
| | FedAsyn | 1.00 | 2.04 | 9.92 | 46.64 | 57.42 | 60.32 | 61.04 |

