# OpenReview forum: "FedGMR: Federated Learning with Gradual Model Restoration under Asynchrony and Model Heterogeneity"
_ICLR.cc/2026/Conference — Submitted to ICLR 2026_

### Official Review · Reviewer_ooYy · 2025-10-26

**Soundness:** 3
**Presentation:** 2
**Contribution:** 2
**Rating:** 4
**Confidence:** 5

**Summary:**

The paper introduces FedGMR to address the limitations of Bandwidth-Constrained Clients (BCCs) and model heterogeneity in federated learning. In many real-world FL systems, clients have varying communication capacities and cannot always train or transmit full models, leading to slow convergence and poor generalization. To mitigate this, FedGMR progressively restores model capacity on each client over time, allowing BCCs to contribute continuously. The method also introduces a transmission and aggregation mechanism that aligns updates from heterogeneous clients and ensures consistent model integration across asynchronous updates. The authors provide theoretical convergence guarantees, showing that the average sub-model density influences the error bound and that the proposed gradual restoration narrows the gap to full-model federated learning. Experimental results on FEMNIST, CIFAR-10, and ImageNet-100 demonstrate that FedGMR delivers faster convergence and higher accuracy than existing FL baselines, particularly in settings with severe non-IID data and system heterogeneity.

**Strengths:**

+ Introduces a new concept of Gradual Model Restoration (GMR), which progressively increases each client’s model capacity.
+ Designs a mask-aware transmission and aggregation mechanism that aligns updates from heterogeneous clients, ensuring stable training despite differing model structures.
+ Provides formal convergence guarantees under the proposed mechanism.
+ Offers a mathematically grounded explanation of how gradual restoration narrows the performance gap to full-model FL.
+ Conducts extensive experiments on diverse and well-recognized benchmarks: FEMNIST, CIFAR-10, and ImageNet-100.

**Weaknesses:**

- As BCCs gradually restore larger sub-models, their computation and communication loads increase, making them more prone to straggling (delayed updates). This straggler effect can partially offset the benefits of gradual restoration, especially in asynchronous environments where delayed clients slow global progress or introduce stale gradients. The paper does not analyze or mitigate this trade-off, nor quantify how much density increase is sustainable before latency outweighs the learning gain. This limitation suggests that while FedGMR improves participation for BCCs initially, it may reintroduce asynchrony inefficiencies in later training stages as model sizes grow.
- FedGMR enables BCCs to train sub-models to mitigate the straggler effect. Recent works such as [R1] and [R2] leverage tiering approaches to mitigate the straggler effect, which should be discussed and compared in the paper.

[R1] Chai, Zheng, Yujing Chen, Ali Anwar, Liang Zhao, Yue Cheng, and Huzefa Rangwala. "FedAT: A high-performance and communication-efficient federated learning system with asynchronous tiers." In Proceedings of the international conference for high performance computing, networking, storage and analysis, pp. 1-16. 2021.

[R2] Mohammadabadi, Seyed Mahmoud Sajjadi, Syed Zawad, Feng Yan, and Lei Yang. "Speed up federated learning in heterogeneous environments: a dynamic tiering approach." IEEE Internet of Things Journal (2024).

- Experiments are conducted mainly on image classification datasets (FEMNIST, CIFAR-10, ImageNet-100); no evaluation on other domains such as NLP, speech, or sensor data, which would strengthen the generality of the approach.
- Results focus heavily on accuracy and convergence, with no measurement of communication overhead, latency. Also, the results are given under a fixed wall-clock budget. It is suggested to also provide results under a fixed target accuracy and compare the convergence time.
- The performance of FedGMR on transformer models is not evaluated.
- The methodology section is dense, and key components like mask-aware aggregation are only briefly explained.

**Questions:**

1. How does FedGMR handle the growing computational and communication burden as sub-models expand for BCCs? Please analyze or visualize the relationship between model density and update delay. An ablation showing when the latency begins to outweigh the accuracy gain would help quantify this trade-off.
2. How does FedGMR compare conceptually and empirically with tier-based asynchronous frameworks like FedAT ([R1]) and Dynamic Tiering ([R2])? Please include a discussion or experimental comparison with these approaches, since both aim to mitigate straggler effects. Highlight how FedGMR’s gradual model restoration differs from or complements tiering-based synchronization strategies in terms of communication cost and convergence speed.
3. Can FedGMR generalize beyond image classification tasks? Include or discuss experiments in non-vision domains such as NLP or sensor data to show generality, since many FL applications (e.g., federated BERT fine-tuning) involve non-visual modalities.
4. Provide additional results under a fixed target accuracy and report the time to convergence for each baseline. This would allow a fairer assessment of FedGMR’s training efficiency. Include measurements of communication overhead and latency, since these are central to evaluating the benefits of gradual model restoration.
5. How does FedGMR perform with Transformer or attention-based architectures, which are now common in FL applications? Add an experiment or at least a discussion on how model restoration and mask-aware aggregation behave when applied to Transformers. The impact on gradient synchronization and sub-model alignment would be insightful.
6. How scalable is FedGMR in large federations (e.g., thousands of clients) where communication delays and resource diversity are more extreme? Discuss the scalability of  FedGMR.

---

> ### Author Response · Authors · 2025-11-30
>
> **Dear Reviewer ooYy,**
>
> We sincerely thank you for the time and effort you dedicated to reviewing our manuscript. We greatly appreciate your insightful comments, particularly regarding the theoretical assumptions, the clarity of the GMR mechanism, and the presentation of our figures. We have carefully considered your advice and have made significant revisions to improve the readability and theoretical depth of the paper.
>
> Below, we provide a point-by-point response to your concerns.
>
> -----
>
> >**Weakness 1. As BCCs gradually restore larger sub-models, their computation and communication loads increase, making them more prone to straggling (delayed updates). This straggler effect can partially offset the benefits of gradual restoration, especially in asynchronous environments where delayed clients slow global progress or introduce stale gradients. The paper does not analyze or mitigate this trade-off, nor quantify how much density increase is sustainable before latency outweighs the learning gain. This limitation suggests that while FedGMR improves participation for BCCs initially, it may reintroduce asynchrony inefficiencies in later training stages as model sizes grow.
> How does FedGMR handle the growing computational and communication burden as sub-models expand for BCCs? Please analyze or visualize the relationship between model density and update delay. An ablation showing when the latency begins to outweigh the accuracy gain would help quantify this trade-off.**
>
>
> **Response:**
> We appreciate the reviewer for highlighting the critical trade-off between model density and system latency.
>
> **1. Quality vs. Frequency Trade-off:**
> We acknowledge that increasing sub-model density inevitably increases computation and communication loads, potentially reducing the participation frequency of BCCs. However, our rationale is based on the **informational value** of updates rather than just frequency.
> * **The Plateau Problem:** We trigger model restoration only when the smaller sub-models have entered a **plateau phase** (as shown in our feasibility study). In this circumstance, even if small models maintain high participation frequency, their contribution to global convergence becomes negligible due to limited capacity (diminishing returns).
> * **High-Value Updates:** Conversely, while restored models may update more slowly, they break the capacity bottleneck. Therefore, a "slower but informative" update contributes more to convergence than a "fast but saturated" update.
>
> **2. Dynamic Optimal Density:**
> Our hypothesis is that the **optimal sub-model density is stage-dependent and strictly increasing**.
> * In the early stages, coarse-grained (low-density) features are sufficient, favoring high-frequency updates.
> * In later stages, fine-grained features require higher capacity, making density more critical than speed.
> Our feasibility analysis in Section [X] empirically proves this: fixed small densities fail to improve late-stage accuracy, whereas increasing density unlocks further gains.
>
> **3. Empirical Verification of Speed:**
> Although individual client latency increases, the **overall convergence speed** (Accuracy vs. Wall-clock Time) actually improves.
> * **Evidence:** We have added a new analysis to the revised manuscript: *"Moreover, we plot the average client model density and test accuracy against time in Fig.3. These metrics demonstrate that FedGMR outperforms the ablation methods, specifically in terms of speed during the model restoration phase."*
> * This confirms that the learning gain from increased capacity significantly outweighs the latency costs introduced by stragglers.
>
> -----

---

> > ### Author Response · Authors · 2025-11-30
> >
> > > **Weakness 2. FedGMR enables BCCs to train sub-models to mitigate the straggler effect. Recent works such as [R1] and [R2] leverage tiering approaches to mitigate the straggler effect, which should be discussed and compared in the paper.
> > [R1] Chai, Zheng, Yujing Chen, Ali Anwar, Liang Zhao, Yue Cheng, and Huzefa Rangwala. "FedAT: A high-performance and communication-efficient federated learning system with asynchronous tiers." In Proceedings of the international conference for high performance computing, networking, storage and analysis, pp. 1-16. 2021.
> > How does FedGMR compare conceptually and empirically with tier-based asynchronous frameworks like FedAT ([R1]) and Dynamic Tiering ([R2])? Please include a discussion or experimental comparison with these approaches, since both aim to mitigate straggler effects. Highlight how FedGMR’s gradual model restoration differs from or complements tiering-based synchronization strategies in terms of communication cost and convergence speed.
> >
> > ### Response
> >
> > We thank the reviewer for highlighting these significant works ([R1] FedAT, [R2] Dynamic Tiering). We agree that tiering-based approaches are highly effective for mitigating straggler effects.
> >
> > **1. Conceptual Comparison: System-Centric vs. Model-Centric**
> > While both FedGMR and Tiering approaches aim to mitigate straggler effects, they operate on fundamentally different (orthogonal) paradigms:
> > How does FedGMR handle the growing computational and communication burden as sub-models expand for BCCs? Please analyze or visualize the relationship between model density and update delay. An ablation showing when the latency begins to outweigh the accuracy gain would help quantify this trade-off.
> >
> > * **Tiering Approaches ([R1], [R2]):** These methods tackle the problem from a **System/Scheduling perspective**.
> >     * **Mechanism:** They group clients into tiers based on response latency. Fast tiers often perform synchronous aggregation, while slow tiers are handled asynchronously or with delayed synchronization.
> >     * **Limitation:** Crucially, tiering primarily aims to **"hide" latency**. The stragglers still bear the full computational and communication burden of the original model. If a client is resource-constrained (e.g., OOM or extremely low bandwidth), tiering alone cannot enable its participation.
> >
> > * **FedGMR (Ours):** Our method addresses the problem from a **Model-Centric perspective** via Model-Heterogeneous FL.
> >     * **Mechanism:** Leveraging MHFL, FedGMR **dynamically adjusts model densities (and the resulting gradient payloads)** based on **training stage characteristics**.
> >     * **Advantage:** By assigning smaller sub-models during the early plateau phase and gradually restoring them later, we ensure that resource-constrained BCCs can **continuously participate** throughout the training process. This allows them to contribute effective gradients adapted to the current training stage, rather than being sidelined or delayed as in static tiering.
> >
> > **2. Synergy and Integration (Complementary Nature)**
> > We believe these two strategies are not mutually exclusive but rather **highly complementary**.
> > * **Current FedGMR:** Currently, FedGMR employs a server-side buffering mechanism to handle asynchronous updates from clients with varying restoration speeds.
> > * **Potential Integration:** We find the tiering strategy in [R1] particularly inspiring for optimizing our aggregation phase. By integrating tiering, we could group clients with similar current densities or speeds into tiers and perform **intra-tier aggregation** before global buffering. This would significantly reduce the buffering overhead and improve the stability of asynchronous aggregation. We have added a discussion on this promising direction in the **Future Work** section.
> >
> > **3. Action Plan**
> > * **Extended Discussion:** In the revised manuscript, we will add a dedicated section in the **Appendix (Supplementary Related Work)** to extensively discuss [R1] and [R2], clarifying the distinctions between Tiering-based synchronization and Gradual Model Restoration.
> >
> > * **Experimental Comparison:** We are making our **best efforts** to implement these tiering methods as additional baselines. We aim to include an empirical comparison in the final version to demonstrate how FedGMR compares with (or outperforms in resource-constrained settings) these strategies, subject to computational resource constraints.

---

> > > ### Author Response · Authors · 2025-11-30
> > >
> > > > **Weakness 3. Experiments are conducted mainly on image classification datasets (FEMNIST, CIFAR-10, ImageNet-100); no evaluation on other domains such as NLP, speech, or sensor data, which would strengthen the generality of the approach.
> > > The performance of FedGMR on transformer models is not evaluated.
> > > How does FedGMR perform with Transformer or attention-based architectures, which are now common in FL applications? Add an experiment or at least a discussion on how model restoration and mask-aware aggregation behave when applied to Transformers. The impact on gradient synchronization and sub-model alignment would be insightful.
> > > Can FedGMR generalize beyond image classification tasks? Include or discuss experiments in non-vision domains such as NLP or sensor data to show generality, since many FL applications (e.g., federated BERT fine-tuning) involve non-visual modalities.
> > >
> > > -----
> > > > **Weakness 4.The methodology section is dense, and key components like mask-aware aggregation are only briefly explained.
> > >
> > >
> > > ### Response
> > >
> > > We sincerely appreciate the reviewer's constructive feedback regarding the readability and structure of the Methodology section. We acknowledge that the previous version attempted to present too many peripheral details (such as the optional IMS module) simultaneously, which obscured the core contributions and made the section overly dense.
> > >
> > > **1. Restructuring for Clarity (Focusing on GMR):**
> > > In the revised manuscript, we have **fundamentally restructured the Methodology section** to establish a clear hierarchy of contributions:
> > > * **Core Contribution:** We now explicitly position **Gradual Model Restoration (GMR)** as the central innovation. The narrative focuses on how the density adjustment strategy evolves during training to mitigate stragglers.
> > > * **De-emphasizing Secondary Components:** To reduce "noise," we have reclassified the Increment Merging Strategy (IMS) as an optional optimization and moved its detailed algorithmic description to the Appendix/Supplementary Material.
> > >
> > > **2. Expanded Explanation of Mask-Aware Aggregation:**
> > > Addressing the reviewer's specific concern, we have significantly expanded the subsection on **Mask-Aware Aggregation**.
> > > * **Clarification:** Instead of a brief mention, we now detail how this aggregation scheme is optimized specifically for our asynchronous environment.
> > > * **Theoretical Connection:** We explicitly link the aggregation logic to our **Convergence Analysis**, explaining how Mask-Aware Aggregation mathematically bridges the gap between the trained sub-models and the full global model, and how GMR progressively narrows this gap.
> > >
> > > ### Response
> > >
> > > We appreciate the reviewer's suggestion to evaluate the generality of our approach beyond computer vision and CNNs.
> > >
> > > **1. Preliminary Experiments on NLP with Transformers:**
> > > Following the reviewer's advice, we have conducted new preliminary experiments using a **2-layer Transformer model** on the **StackOverflow** dataset (a next-word prediction NLP task with a vocabulary size of 10,000) over 10,000 communication rounds.
> > >
> > > **2. Empirical Results:**
> > > The results demonstrate that FedGMR is effective for Transformer architectures, showing advantages in both accuracy and, more significantly, **convergence speed**:
> > >
> > > * **Scenario A (Non-IID, High Model Heterogeneity):**
> > >     * **Accuracy:** FedGMR achieves **28.81%** vs. w/o GMR **28.56%**.
> > >     * **Speed:** FedGMR reached the baseline's final accuracy (28.56%) at **Round 8,000**, implying a **20% acceleration** in convergence.
> > > * **Scenario B (IID, Low Model Heterogeneity):**
> > >     * **Accuracy:** FedGMR achieves **28.95%** vs. w/o GMR **28.55%**.
> > >     * **Speed:** FedGMR reached the baseline's final accuracy at **Round 5,300**, implying a **47% acceleration**.
> > >
> > > **3. Analysis:**
> > > * **Architecture Agnostic:** These results confirm that the Gradual Model Restoration (GMR) mechanism is architecture-agnostic and effective for Transformers.
> > > * **Convergence Efficiency:** While the absolute accuracy gain appears marginal (typical for high-dimensional next-word prediction tasks), the significant reduction in communication rounds required to reach target accuracy highlights the efficiency of our method.
> > > * **Performance in Simpler Distributions:** We observed that FedGMR yields larger gains in simpler data distributions (IID). This suggests that when data distribution is less complex, the **restored model capacity** allows the global model to learn more effectively and converge significantly faster compared to fixed-size sub-models.
> > >
> > > **4. Action Plan:**
> > > We are currently finalizing these experiments and will include the complete results and analysis in the **Experiment Section** of the final version to demonstrate the method's generality across NLP domains.

---

> > > > ### Author Response · Authors · 2025-11-30
> > > >
> > > > > **Weakness 5. How scalable is FedGMR in large federations (e.g., thousands of clients) where communication delays and resource diversity are more extreme? Discuss the scalability of FedGMR.
> > > > -----
> > > >
> > > > We appreciate the reviewer's foresight regarding the scalability of FedGMR in large-scale federations.
> > > >
> > > > **1. Server-Side Scalability (Storage & Optimization):**
> > > > We acknowledge that FedGMR requires a server-side buffering mechanism to maintain asynchronous client states, which consumes additional memory.
> > > > * **Feasibility:** In practice, cloud servers typically possess abundant memory resources, making this trade-off acceptable for most standard FL scenarios.
> > > > * **Optimization for Massive Scale:** To further enhance scalability for thousands of clients, we can integrate the **Tiering strategy** (as discussed in our response to [R1]). By grouping stragglers and performing **intra-tier aggregation**, we can significantly reduce the number of buffered models required, thereby lowering the storage burden.
> > > >
> > > > **2. Communication Scalability (IMS Module):**
> > > > Communication delay is often the bottleneck in large federations. Here, our optional **Incremental Model Splitting (IMS)** module plays a critical role.
> > > > * **Mechanism:** IMS optimizes the server-to-client transmission by reconstructing the required sub-models from model increments, rather than re-transmitting entire parameters.
> > > > * **Empirical Evidence:** We have validated the scalability of this approach in the **Appendix**. As shown in **Table 8**, IMS significantly reduces downlink communication costs, making FedGMR highly efficient even when scaling to large numbers of clients.
> > > >
> > > > **3. Robustness to Resource Diversity:**
> > > > In large federations, resource diversity is indeed more extreme. FedGMR is inherently designed to handle this: unlike fixed-size methods, our **Gradual Model Restoration** allows the system to adaptively utilize the capacity of heterogeneous clients (from IoT devices to powerful edges), ensuring that high diversity contributes to, rather than hinders, global model convergence.
> > > >
> > > > We believe these revisions, driven by your constructive feedback, have significantly improved the quality and clarity of our work.
> > > >
> > > > Sincerely,
> > > >
> > > > **The Authors**

---

### Official Review · Reviewer_7UBb · 2025-10-26

**Soundness:** 2
**Presentation:** 1
**Contribution:** 2
**Rating:** 4
**Confidence:** 1

**Summary:**

This work proposes a new heterogeneous FL frameowork, FedGMR. This framework consists of mainly three components, gradual density adjustment, buffered mask-aware aggregation, and incremental model splitting. In addition to those main components, the authors employ semi-asynchronous update aggregation scheme to alleviate the straggler effect (synchronization cost) in FL. The authors provides theoretical analysis of convergence behaviors as well as empirical results of accuracy comparisons across SOTA heterogeneous FL methods. Overall, the paper shows promising FL performance, however, I see several strong weaknesses that should be addressed.

**Strengths:**

1. The authors consider important and realistic issue in FL, the heterogeneous system environments together with data heterogeneity.
2. The theoretical analysis well incorporate the proposed method into the traditional analysis framework.
3. Asynchronous scheme looks promising. I especially appreciate this approach since most of recent FL studies just simply consider synchronous model aggregations that are not realistic.

**Weaknesses:**

**Comments on Main Idea**

1. The authors assume that weak clients become straggler mainly due to the limited network bandwidth. However, such weak clients tend to have limited system resources such as slow compute power and a limited memory space. Thus, gradually increasing density may make some weak clients with limited resources impossible to join the training any more. Therefore, I do not think the problem definition is convincing.

2. In this work, 'model-heterogeneous' term may mislead readers. In general, 'heterogeneous' indicates independently designed entities. E.g., heterogeneous data distributions do not have any dependencies across local datasets. However, this work assumes that weak clients have limited system resources and are assigned with models differently prunned. Basically, however, they share the same model architecture. Thus, this study does not cover model-heterogeneious FL. It would rather be a prunning-based sub-model distribution method.

3. How is GMR equation (4) built? What is the meaning of multiplying learning rate $\lambda$ to the ratio of time difference to the BCC's time? The main equation is not well substantiated due to the limiated explanation. The authors should focus more on **why** (4) is the best choice rather than **how** only.

4. In theoretical analysis, the assumptions are too strong. Especially, the bounded noise (assumption 3) and bias (assumption 4) are unrealistic. In addition, most of recent studies do not rely on the assumption of bounded gradient magnitude (assumption 2). Instead, the bounded gradient variance assumption is used popularly. Given these strong assumptions, the results are not that convincing.

5. Also, there are $f_1$, $f_2$, and $f_3$ in Theorem 2, but they have never been defined before. What does it mean by the subscripts?

6. Finally, I recommend analyzing the complexity of the derived convergence rate so that the performance can be easily compared with other methods. Since there are many indirect notations such as $\mathcal{A}$ and $\mathcal{B}$, it is not easy to compare the performance.

7. FedGMR obviously uses much more server-side system resources such as memory space for local models. However, other methods do not require such strong resources at the server-side. E.g., HeteroFL just align the submodels and directly average the parameters. Fjord also just randomly prunned local models are aggregated at the server-side. Thus, even though FedGMR achieves higher accuracy than other methods, the performance gain probably comes from using more resources. Table 1 does not consider such differences and just directly compare the achieved accuracy within the same amount of time. I do not think it is a fair comparison.

8. As compared to the best-performing SOTA method, the performance gain of FedGMR is not significant. E.g., FEMNIST non-IID performance gap between FedGMR and FedAsync is just 0.9% when non-IIDness is low. When non-IIDness is high, Fjord becomes the best. For CIFAR-10, the difference becomes 1.1~1.2%. Considering that FedGMR uses more server-side resources, I think this performance gain is not that impressive.

9. Some recently published and strongly related heterogeneous FL methods could be directly compared in Table 1 or at least discussed in Section 2. Some examples are as follows.

[1] Liu et al., Efficient Federated Learning with Heterogeneous Data and Adaptive Dropout, TKDD, 2025.

[2] Lee et al., Embracing Federated Learning: Enabling Weak Client Participation via Partial Model Training, IEEE Trans. on Mobile Computing, 2024.

[3] Liu et al., No One Left Behind: Inclusive Federated Learning over Heterogeneous Devices, KDD, 2022.

**Comments on Presentation Quality**

1. The contribution summary at the end of Introduction section does not deliver meaningful information while taking up the space of several lines. I see many recent literature has this summary at the same position, but what is the point of having them? The summary includes nothing new, the proposed idea, the existance of convergence analysis, and experimental settings. I strongly recommend removing them and use the space for other more critical things, e.g., more experimental results or detailed discussion of main ideas.

2. Figure 1 does not clearly explain the workflow. First, what is the meaning of numbers? The caption explains the steps in the order of 3, 4, 1, and 2. It seriously confuses readers. Second, after the step 2, there are two horizontal flows, 3 and 4. The figure neither explains what they are nor why they appear in the figure. Overall, the figure should be improved much to deliver key ideas.

3. Algorithm 1 is not self-contained. What is IMSc? What is GMR? They should be at least connected to any equations or subsections where the method is explained.

4. There are too many acronyms which seriously hurts readability. BAC, BCC, GMR, IMS, MHFL, MA, GA, MRI, etc... I suggest using their full names unless there are some special reasons. Just few are okay, but there are too many now.

**Questions:**

Some questions are included in the above weakness section. Please carefully address them.

---

> ### Author Response · Authors · 2025-11-30
>
> **Dear Reviewer 7UBb,**
>
> We sincerely thank you for the time and effort you dedicated to reviewing our manuscript. We greatly appreciate your insightful comments, particularly regarding the theoretical assumptions, the clarity of the GMR mechanism, and the presentation of our figures. We have carefully considered your advice and have made significant revisions to improve the readability and theoretical depth of the paper.
>
> Below, we provide a point-by-point response to your concerns.
>
> -----
>
> ### **Specific Comments and Responses**
>
> > **Weakness 1: The authors assume that weak clients become straggler mainly due to the limited network bandwidth. However, such weak clients tend to have limited system resources such as slow compute power and a limited memory space. Thus, gradually increasing density may make some weak clients with limited resources impossible to join the training any more. Therefore, I do not think the problem definition is convincing.**
>
> **Response:**
>
> Thank you for raising this critical point regarding the distinction between communication and computation bottlenecks.
>
> In our current assumption, we primarily focus on **communication constraints**, as sub-model allocation in existing Model-Heterogeneous Federated Learning (MHFL) works is typically determined by **network transmission speeds** rather than the absolute **computational upper bound** (e.g., memory limit) of the device. Consequently, there is often a gap between the pre-assigned sub-model size (constrained by bandwidth) and the client’s actual computational capacity. Our method, FedGMR, aims to bridge this gap: as long as the restored model size remains within the device's hardware limits, increasing the density is beneficial for the overall training.
>
> We acknowledge that density restoration must be managed carefully; if the model grows beyond a weak client's computational limit, that client may indeed drop out. However, from another perspective, even when computational limits are considered, our approach offers a valuable alternative strategy: by initially assigning a smaller model (below the computational limit) and then gradually restoring it, we can **accelerate the early training phase** and potentially allow for the training of deeper networks compared to static assignments.
>
> -----
>
> > **2. In this work, 'model-heterogeneous' term may mislead readers. In general, 'heterogeneous' indicates independently designed entities... This work assumes that weak clients have limited system resources and are assigned with models differently prunned... Thus, this study does not cover model-heterogeneious FL. It would rather be a prunning-based sub-model distribution method.**
>
> **Response:**
>
> We understand your concern regarding the terminology. Strictly speaking, "heterogeneous models" could imply entirely different architectures (e.g., ResNet vs. VGG). However, we adopted the term **"Model-Heterogeneous Federated Learning" (MHFL)** following established conventions in recent top-tier literature that specifically deal with width-adjusted or pruned sub-models.
>
> For instance, the following seminal works refer to this specific setting (sub-model extraction from a global model) as Model-Heterogeneous FL:
>
>   * *[1] Alam et al., "FedRolex: Model-heterogeneous federated learning with rolling sub-model extraction" (NeurIPS 2022).*
>   * *[2] Zhou et al., "Every parameter matters: Ensuring the convergence of federated learning with dynamic heterogeneous models reduction" (NeurIPS 2023).*
>   * *[3] Wu et al., "FIARSE: Model-heterogeneous federated learning via importance-aware submodel extraction" (NeurIPS 2024).*
>
>
> To address potential ambiguity, we have added a dedicated definition in Section 3 of the revised manuscript. We explicitly clarify that in the context of this work, "Model-Heterogeneous" refers to "Model-Size Heterogeneity" achieved via structured pruning or sub-model extraction, aligning with the aforementioned studies.

---

> ### Author Response · Authors · 2025-11-30
>
> > **Weakness 3. How is GMR equation (4) built? What is the meaning of multiplying learning rate $\eta$ to the ratio of time difference to the BCC's time? The main equation is not well substantiated due to the limited explanation. The authors should focus more on why (4) is the best choice rather than how only.**
>
> **Response:**
>
> We appreciate the reviewer's query regarding the derivation and rationale behind Equation (4).
>
> **1. Construction based on Gradient Descent:**
> Equation (4) is fundamentally constructed based on the principle of **Gradient Descent**. We formulate the density allocation problem as an optimization task where the objective is to align the completion time of each client with the target time (the Board of Client Committee's time). The ratio of the time difference acts as the "gradient": if a client is slower than the target, the gradient dictates a reduction in model density for the next round, and vice versa.
>
> **2. The Meaning of the Learning Rate ($\eta$):**
> The learning rate $\eta$ is essential because real-world communication and computation times are **stochastic** rather than static. Network speeds and device loads fluctuate due to randomness. Consequently, a "one-shot" adjustment based on a single round's delay would be unstable. Multiplying by $\eta$ allows for **iterative adjustments**, acting as a damping factor that mitigates the impact of temporary noise and ensures stable convergence.
>
> **3. Advantages of this Formulation:**
> The primary advantage of this gradient-based method is that it enables the server to adaptively assign appropriate client model densities **without requiring prior knowledge of client system information** (e.g., bandwidth or CPU specifications). It effectively "learns" a suitable density through interaction. We have provided a proof of the **validity range** for this method in **Appendix B**.
>
>
> -----
>
> > **Weakness 4. In theoretical analysis, the assumptions are too strong. Especially, the bounded noise (assumption 3) and bias (assumption 4) are unrealistic. In addition, most of recent studies do not rely on the assumption of bounded gradient magnitude (assumption 2). Instead, the bounded gradient.**
>
> **Response:**
>
> We thank the reviewer for the rigorous assessment of our theoretical analysis.
>
> **1. Context of Assumptions:**
> We acknowledge that Assumptions 2, 3, and 4 are relatively strong conditions compared to some recent studies in standard FL. However, we adopted these assumptions following the theoretical frameworks established in recent top-tier literature specifically addressing **Model-Heterogeneous Federated Learning (MHFL)**, such as *Zhou et al., "Every parameter matters..." (NeurIPS 2023) [2]*. Due to the complexity introduced by dynamic sub-model extraction and varying architectures, these bounded assumptions are widely used in this subdomain to make the derivation tractable.
>
> **2. Analytical Objective:**
> Our primary theoretical goal is not to improve the convergence bounds of general FL, but to explicitly characterize the impact of **Model-Size Heterogeneity** and our **GMR mechanism**. Specifically, we aim to decompose the convergence bound into three interpretable terms: **model drift, stochastic variance, and non-IID bias**. These assumptions allow us to cleanly derive Theorem 1 (bounding the one-round gradient deviation) and demonstrate how GMR, combined with mask-aware aggregation, mitigates the divergence caused by training partial models compared to full-model training.
>
> **3. Future Improvements:**
> We agree with the reviewer that relaxing these constraints (e.g., using bounded variance instead of bounded gradient magnitude) would further strengthen the generality of the analysis. However, given that our current derivation successfully elucidates the convergence behavior of GMR under non-IID settings consistent with state-of-the-art MHFL baselines, we believe it provides sufficient theoretical guarantee for the proposed method.

---

> > ### Author Response · Authors · 2025-11-30
> >
> > > **Weakness 5. Also, there are $f_1$, $f_2$, and $f_3$ in Theorem 2, but they have never been defined before. What does it mean by the subscripts?**
> >
> > **Response:**
> >
> > We apologize for the confusion regarding the notation.
> >
> > **1. Location of Definitions:**
> > These terms were originally defined prior to the assumptions in the initial submission. However, to improve readability and flow in the revised manuscript, **we have relocated their definitions to immediately follow the Assumptions section.**
> >
> > **2. Physical Meaning and Properties:**
> > Mathematically, $f_1, f_2, \dots$ are **monotonically increasing functions** with respect to the model density $\rho$, taking values in the range $[0, 1]$.
> > * **Rationale:** Since we first defined the gradient and variance bounds for the **full model**, the corresponding values for any **sub-model** must be smaller than or equal to those of the full model.
> > * **Nested Structure:** Because our sub-models are constructed using a **nested** structure (i.e., smaller sub-models are subsets of larger ones), the norm of the sub-gradients naturally increases as the density increases.
> >
> > Therefore, these coefficients serve as scaling factors to strictly bound the sub-model gradients based on their density.
> >
> > -----
> > > **Weakness 6. Finally, I recommend analyzing the complexity of the derived convergence rate so that the performance can be easily compared with other methods. Since there are many indirect notations such as $A$ and $B$, it is not easy to compare the performance.**
> >
> > **Response:**
> > We appreciate the reviewer’s suggestion to clarify the complexity analysis for better comparability.
> >
> > **1. Rationale for Original Notation:**
> > The notations in question (e.g., $A, B$) were derived from the **superposition of sub-model densities**. We originally retained this detailed form to explicitly demonstrate how the **accumulated density of each individual client** quantitatively impacts the final convergence bound.
> >
> > **2. Revision for Clarity:**
> > However, we agree that these indirect notations made the result difficult to compare with other methods. In the **revised manuscript**, we have **optimized the formulation**. To improve readability, we now present the convergence result in the main text specifically focused on **mask-aware aggregation**. This simplified form highlights the order of convergence more clearly, allowing for easier comparison with existing baselines, while the detailed derivations have been moved to the Appendix.
> >
> > -----
> > > **Weakness 7. FedGMR obviously uses much more server-side system resources such as memory space for local models. However, other methods do not require such strong resources at the server-side... Table 1 does not consider such differences and just directly compare the achieved accuracy within the same amount of time. I do not think it is a fair comparison.**
> >
> > **Response:**
> > We appreciate the reviewer's concern regarding the fairness of comparison and server-side resource consumption.
> >
> > **1. Clarification on Server-side Memory Usage:**
> > While FedGMR maintains a cache of the latest client models to support asynchronous aggregation (preventing "catastrophic forgetting" of slow clients), the **peak memory consumption** is comparable to synchronous methods like HeteroFL.
> > * **Mechanism:** In synchronous FL, the server must also receive and buffer updates from all participating clients before the aggregation step (and discards them afterward).
> > * **Conclusion:** Therefore, the *maximum* storage capacity required at any single moment is similar. The primary difference is that FedGMR persists these states longer to enable asynchrony, utilizing server-side memory (which is typically abundant in cloud environments) to alleviate bottlenecks.
> >
> > **2. Fairness Validation via Ablation Studies:**
> > To rigorously verify that the performance gain stems from our proposed mechanism rather than increased resource usage, we refer to the **Ablation Studies**.
> > * The ablation baseline (FedGMR w/o GMR mechanism) operates under the **exact same experimental setting**, including the asynchronous buffering architecture and server resources.
> > * **Result:** As shown in the results, FedGMR significantly outperforms the ablation method. This confirms that the accuracy improvement is driven by the **GMR algorithm** (restoring sub-model density) rather than the mere buffering of models.
> >
> > **3. Trade-off Philosophy (Comparison with Fjord):**
> > Regarding Fjord, it employs Ordered Dropout, which often requires training sub-models of varying sizes or handling complex dropout masks on the client side. This imposes additional computational or logical burdens on resource-constrained **clients**. In contrast, FedGMR shifts the complexity to the **server** (in terms of memory). In Federated Learning, since client resources (battery, compute) are the primary bottleneck while server resources are relatively scalable, we believe this trade-off is strategically advantageous.
> >
> > -----

---

> > > ### Author Response · Authors · 2025-11-30
> > >
> > > > **Weakness 8. As compared to the best-performing SOTA method, the performance gain of FedGMR is not significant. E.g., FEMNIST non-IID performance gap... is just 0.9%... For CIFAR-10, the difference becomes 1.1\~1.2%. Considering that FedGMR uses more server-side resources, I think this performance gain is not that impressive.**
> > >
> > > **Response:**
> > > We appreciate the reviewer's detailed observation regarding the performance gaps.
> > >
> > > **1. Correlation with Task Complexity:**
> > > The performance advantage of FedGMR is **positively correlated with the difficulty of the task**.
> > > * **Simple Tasks (FEMNIST):** FEMNIST is a relatively simple dataset where baseline methods like FedAsync already achieve near-optimal performance. In such cases, the "performance saturation" effect makes it difficult for any method to achieve large marginal gains, as the fixed sub-models are already sufficient to learn the task features.
> > > * **Complex Tasks (CIFAR-10 & ImageNet-100):** The advantage of FedGMR becomes much more pronounced in challenging scenarios where model capacity is the primary bottleneck. As the task difficulty increases, the limitations of bandwidth-constrained sub-models become critical.
> > >
> > > **2. Substantial Gains on Harder Datasets:**
> > > Crucially, on the most challenging dataset, **ImageNet-100**, FedGMR achieves a substantial performance gain of **approximately 10%** compared to the baselines.
> > > * This empirically proves our hypothesis: in complex tasks, fixed narrow sub-models (used by other methods) create an "expressivity bottleneck." By dynamically restoring model density, FedGMR effectively breaks this bottleneck, delivering significant improvements where they matter most.
> > >
> > > **3. Conclusion:**
> > > Therefore, while the gains on simpler tasks are modest due to saturation, the significant improvements on complex benchmarks demonstrate the scalability and effectiveness of FedGMR in real-world, high-complexity scenarios.
> > >
> > > -----
> > >
> > > > **Weakness 9. Some recently published and strongly related heterogeneous FL methods could be directly compared in Table 1 or at least discussed in Section 2.**
> > >
> > > **Response:**
> > > **1. Rationale for Baseline Selection:**
> > > In our current experiments, we carefully selected baselines to represent the **distinct paradigms** of bandwidth-constrained MHFL:
> > > * **HeteroFL:** Represents static, width-based structured pruning.
> > > * **FedRolex:** Represents rolling/dynamic structured pruning.
> > > * **Fjord:** ordered dropout to enhance sub-mdoel structure
> > > * **FIARSE:** Represents importance-aware unstructured pruning.
> > > Our goal was to benchmark FedGMR against the fundamental representatives of each pruning strategy to demonstrate the universality of our contribution.
> > >
> > > **2. GMR as a Plug-and-Play Mechanism:**
> > > Crucially, the core innovation of FedGMR—**Gradual Model Restoration**—is not mutually exclusive with these methods. Instead, it serves as a **generic, plug-and-play training strategy**.
> > > * Most existing MHFL methods (including the ones suggested) operate under the assumption of *fixed* or *cyclic* sub-model sizes.
> > > * The GMR mechanism challenges this by introducing density restoration, which can theoretically be integrated into these frameworks to further unlock their potential.
> > >
> > > **3. Action Plan:**
> > > We agree that discussing these works strengthens the paper.
> > > We are currently working to incorporate these methods as additional baselines in our experiments to provide a more comprehensive comparison in the final version, computational resources permitting.
> > >
> > > -----

---

> > > > ### Author Response · Authors · 2025-11-30
> > > >
> > > > ### Response to Presentation Quality
> > > >
> > > > **1. Contribution Summary:**
> > > > We appreciate the reviewer's suggestion to optimize the space usage.
> > > > * **Revision:** In the revised manuscript, we have rewritten the contribution summary to be **more compact and substantive**. Instead of listing generic experimental settings, we now specifically highlight our key empirical insight: the performance advantage of FedGMR expands significantly as task complexity increases. This provides a high-level preview of our core results without overwhelming the reader with premature details.
> > > >
> > > > **2. Figure 1 Clarity:**
> > > > We are grateful for this constructive feedback. We realize that our previous attempt to illustrate the entire system workflow (including the optional IMS module) in a single figure resulted in a cluttered and confusing visualization, where the core logic was obscured.
> > > > * **Action taken:** We have **completely redesigned Figure 1** in the revised version. The new figure focuses **exclusively on the Gradual Model Restoration (GMR) mechanism**, ensuring a clear, logical flow (chronological order) that directly visualizes the density evolution process, removing the distracting secondary elements.
> > > >
> > > > **3. Algorithm 1 Self-containment:**
> > > > We apologize for the undefined notation. 'IMSc' refers to the client-side *Increment Model Splitting*, which is detailed in the Appendix.
> > > > * **Revision:** We acknowledge that IMS is an optional optimization component. In the revised Algorithm 1, we have explicitly linked these terms to their corresponding equations and sections. We have also clarified the distinction between the core GMR steps and the optional IMS steps to ensure the algorithm is self-contained and easy to follow.
> > > >
> > > > **4. Excessive Acronyms:**
> > > > We fully agree that the excessive use of acronyms hindered readability.
> > > > * **Revision:** We have conducted a thorough review of the manuscript. We **removed non-essential acronyms** (e.g., BAC) and minimized the usage of secondary terms (e.g., IMS). For necessary terminology (e.g., MHFL), we ensured they are defined clearly upon first appearance and used consistently to improve the flow.
> > > >
> > > > We believe these revisions, driven by your constructive feedback, have significantly improved the quality and clarity of our work.
> > > >
> > > > Sincerely,
> > > >
> > > > **The Authors**

---

### Official Review · Reviewer_q72V · 2025-10-29

**Soundness:** 2
**Presentation:** 2
**Contribution:** 2
**Rating:** 4
**Confidence:** 4

**Summary:**

This work proposes FedGMR-Federated Learning with Gradual Model Restoration under Asynchrony and Model Heterogeneity. It progressively increases each client’s model capacity during training, and a tailored transmission and aggregation mechanism is designed to better accommodate system-level heterogeneity. Theory and experiments on FEMNIST, CIFAR-10, and ImageNet-100 confirm the effectiveness of the proposed method.

**Strengths:**

1. This work proposes FedGMR, the first FL framework that dynamically restores sub-model capacity via GMR. Two auxiliary mechanisms, IMS for efficient transmission and BufferMaskFedAvg for robust aggregation under structural inconsistency, are designed to enhance FedGMR.
2. A convergence guarantee is established to show that the average sub-model density across clients and time governs error accumulation.
3. Experiments on FEMNIST, CIFAR-10, and ImageNet-100 demonstrate faster convergence and robustness of proposed method.

**Weaknesses:**

1. What’s the motivation of GMR? Although authors claim that GMR can dynamically restore sub-model capacity, the communication capacity is limited for BCCs. Is it essential for them to train the full model？ Furthermore, the current design of GMR fails to clearly demonstrate how the principles articulated by the authors in Section 3.1 are implemented.
2. The paper's logic seems confused. If the core contribution is GMR, what problems do the other two auxiliary components solve, and are they considered core contributions of the paper as well?
3. Although authors provide the convergence guarantee, the analysis simplifies the core asynchronous process, lacking sufficient theoretical rigor. Given that many recent works [1-3] on asynchronous federated learning have already provided comprehensive convergence analyses for their respective frameworks, this section needs to be strengthened.
4. The experiments only compare against the synchronous submodel training method. The authors should demonstrate the performance of the proposed method in an asynchronous scenario; it is suggested that combined asynchronous baselines be added for comparison to fully prove the method's effectiveness.
5. The paper needs to explicitly indicate which empirical results (figures or tables) demonstrate that the proposed method achieves faster convergence speed than other baselines.
6. The paper's readability needs improvement. A lot of abbreviations without a clear meaning are not feasible for readers to understand the work. Figure 1 fails to clearly show the complete process of the proposed method, and its caption is confusing and inconsistent with the legend.

---
[1] Sharper convergence guarantees for asynchronous SGD for distributed and federated learning.

[2] Asynchronous Federated Optimization

[3] FADAS: Towards Federated Adaptive Asynchronous Optimization

**Questions:**

See weaknesses.

---

> ### Author Response · Authors · 2025-11-30
>
> **Dear reviewer q72V,**
>
> Thank you very much for your constructive comments. We sincerely appreciate your
> detailed feedback. Following your suggestions, we have substantially revised the
> paper to improve readability, clarify motivation, and reorganize contributions.
> Below we respond to each concern.
>
> ---
> ### **Weakness 1 — Motivation of GMR and Implementation of Principles**
>
> **1. Motivation (Quality vs. Quantity):**
> While we acknowledge that restoring full/larger models reduces the participation frequency of Bandwidth-Constrained Clients (BCCs), our motivation addresses the **"saturation" problem** inherent in standard MHFL.
> * **Early Stage:** Small sub-models converge quickly but eventually hit a capacity bottleneck (Plateau Phase), rendering further high-frequency updates ineffective.
> * **Late Stage:** At this point, even if BCCs continue to participate frequently with small models, their contribution diminishes. By restoring the model, we trade participation frequency for **model capacity**. Although BCCs become slower ("stragglers"), their updates become **informative again**, allowing them to contribute to deeper training layers. Thus, a "slower but valid" update is preferred over a "fast but saturated" one.
>
> **2. Implementation of Principles (Two-Stage Design):**
> In the revised manuscript, we have reorganized GMR to explicitly reflect the three principles:
> * **Phase 1 (Balance Principle):** Initially, resource-limited clients are assigned compact sub-models. This aligns with standard MHFL to ensure balanced training speeds and reduce client drift.
> * **Phase 2 (Trade-off Principle):** We monitor the training. Only when sub-models enter a **plateau**, we trigger restoration. This is a deliberate trade-off: sacrificing speed to break the expressivity bottleneck.
> * **Asynchrony Principle:** Since restoration introduces significant speed heterogeneity (BCCs become slower than bandwidth-abundant clients), we employ asynchronous aggregation to handle the mismatch naturally.
>
> ---
> ### **Weakness 2 — Roles of Auxiliary Components**
>
> We have clarified the contribution hierarchy in the revision to avoid confusion:
>
> * **GMR (Core Contribution):** The central strategy of dynamically adjusting model density to navigate the training stages.
> * **Mask-Aware Aggregation & Buffering (Necessary Component):** These are essential to support GMR. Specifically, server-side buffering prevents the global model from "forgetting" the slow BCCs' models during asynchronous updates.
> * **Incremental Model Splitting - IMS (Optional Optimization):**
>     * **Status:** We have strictly reclassified IMS as an **optional** auxiliary module.
>     * **Presentation:** It is **removed from the Introduction** and Figure 1 to reduce noise. It is now detailed in the **Appendix** and evaluated separately in ablation studies as a method to further optimize server-to-client transmission, ensuring the main narrative focuses solely on GMR.
>
> ---
>
> ### **Weakness 3 — Convergence Analysis and Asynchronous Rigor**
>
> We agree that a comprehensive convergence analysis under *dynamic* asynchrony (where the degree of staleness changes as density grows) is extremely challenging and remains future work.
>
> However, our current analysis provides a specific and novel theoretical value:
> * It characterizes the **optimization gap** between heterogeneous sub-model training and full-model FL.
> * It explicitly proves **why increasing density (GMR) narrows this gap**.
> * This theoretical grounding validates the core mechanism of GMR, showing that density restoration mathematically drives the model closer to the optimal global solution, which is a significant contribution in the context of MHFL.
>
>
> ---

---

> > ### Author Response · Authors · 2025-11-30
> >
> > ### **Weakness 4 — Asynchronous Baselines**
> >
> > Our rationale for the current baselines is twofold:
> > 1.  **Focus on GMR's Gain:** FedGMR employs a basic asynchronous setting. Comparing it against standard asynchronous baselines is sufficient to isolate the performance gain attributed to **GMR (the restoration strategy)** rather than the benefits of complex asynchronous scheduling.
> > 2.  **Orthogonality:** We view GMR as a **"plug-and-play"** strategy. It can technically be integrated with more sophisticated asynchronous algorithms to further boost their performance. Our current experiments aim to prove the effectiveness of the GMR mechanism itself.
> >
> > ---
> >
> > ### **Weakness 5 — Faster Convergence Claims**
> >
> > We have explicitly highlighted the empirical evidence in the revision:
> > * **Feasibility Verification:** The first experiment empirically proves our theoretical assumption: the optimal sub-model density is **stage-dependent** and should increase over time.
> > * **Ablation Studies:** We separate the effects of GMR and Aggregation. The results clearly show that **after the restoration point**, the convergence curve of FedGMR significantly outperforms the ablation baselines (w/o GMR), achieving lower loss and higher accuracy.
> > * **Joint Effect:** We also demonstrate the joint effect of GMR and Mask-aware aggregation in facilitating faster convergence compared to static MHFL methods.
> > ---
> >
> > ### **Weakness 6 — Readability and Figure 1**
> >
> > We realized that the previous version was "too noisy" due to an unclear hierarchy of contributions.
> > * **Streamlining:** We have removed undefined abbreviations and focused the narrative strictly on the core GMR mechanism.
> > * **Figure 1 Redesign:** The previous Figure 1 attempted to explain the entire system (including the optional IMS), which obscured the main idea. **The new Figure 1 focuses exclusively on the GMR process**, visually depicting the timeline of density restoration and dynamic adjustment. This change has significantly improved the clarity of our proposed method.
> > ---
> >
> > **Thank you again for your thoughtful feedback. Your comments greatly helped us
> > improve the quality of the paper. We sincerely appreciate your time and would
> > welcome any further suggestions.**

---

### Meta-Review · Area_Chair_hpyF · 2026-01-07

**Summary:**

The reviewers have raised several significant concerns regarding the motivation, theoretical grounding, experimental depth, and overall clarity of the paper. All three reviews include a lack of clarity in presentation and questions regarding the practical trade-offs of the proposed FedGMR method. The theoretical analysis is criticized for being too simplified. Reviewers suggest comparing against more relevant state-of-the-art methods.

**Reviewer Concerns:**

The rebutal partially addressed some of the concerns. Clarity of the paper is improved, yet there are still clarity issues outstanding. For example, Figure 1 still fails to clearly show the complete process of the proposed method, including the novel components of the framework and necessary explanations in the caption.  In addition, no additional baseline methods suggested by the reviewers are compared and presented in the revised version.

**Reviewer Scores:**

I think all reviewers would maintain score given that the additional refs/baselines that each of the reviewers had suggested are still missing from revision. Although the rebutal included them in their "action plan", there are no actual results/discussion found in the revised version.

---

### Decision · Program_Chairs · 2026-01-26

Reject